



# A Modeling Study of Global Distribution and Formation Pathways of Highly Oxygenated Organic Molecules (HOMs) from Monoterpenes

Xinyue Shao[1,2], Yaman Liu[1,3], Xinyi Dong[1,2], Minghuai Wang[1,2], Ruochong Xu[4,5], Joel A. Thornton[5], Duseong S. Jo[6], Man Yue[3], Wenxiang Shen[1,2], Manish Shrivastava[7], Stephen R. Arnold[8], and Ken S. Carslaw[8]

[1]School of Atmospheric Science, Nanjing University, Nanjing, China
[2]Joint International Research Laboratory of Atmospheric and Earth System Sciences & Institute for Climate and Global Change Research, Nanjing University, China
[3]Zhejiang Institute of Meteorological Sciences, Hangzhou, China
[4]Department of Earth System Science, Ministry of Education Key Laboratory for Earth System Modeling, Institute for Global Change Studies, Tsinghua University, Beijing, China
[5]Department of Atmospheric Sciences, University of Washington, Seattle, WA, USA
[6]Department of Earth Science Education, Seoul National University, Seoul, 08826, South Korea
[7]Pacific Northwest National Laboratory, Richland, Washington, USA
[8]School of Earth and Environment, University of Leeds, Leeds, LS2 9JT, UK

*Correspondence to*: Xinyi Dong (dongxy@nju.edu.cn), Minghuai Wang (minghuai.wang@nju.edu.cn)

**Abstract.** Highly oxygenated organic molecules (HOMs) derived from monoterpenes are key precursors of secondary organic aerosols (SOA), yet their global-scale formation pathways and climate impacts remain poorly quantified due to uncertainties in autoxidation kinetics and branching ratios of peroxy radicals. Here, we integrate a comprehensive HOMs chemical mechanism into a global climate model, enabling a systematic evaluation of HOMs-derived SOA (HOMs-SOA) contributions and their sensitivity to key chemical parameters. The improved model shows reasonable agreement in the diurnal cycle and average HOM concentrations (normalized mean biases of 69% and 121% at the two sites). Sensitivity experiments identify the branching ratio of autoxidation-capable peroxy radicals (MT-bRO$_2$) as the dominant uncertainty source. While the MT-bRO$_2$ branching ratio has limited impact on C$_{10}$-HOMs concentrations (~60% formed via NO-terminated autoxidation), it strongly regulates C$_{15}$/C$_{20}$-HOM concentrations produced through cross-reactions of biogenic peroxy radicals. The contribution of HOMs-SOA to total monoterpene-derived SOA ranges from 19% to 41%, depending on the MT-bRO$_2$ branching ratio used in chamber experiments. C$_{15}$ and C$_{20}$ accretion products dominate in pristine regions (e.g., the Amazon, contributing ~50% of HOMs-SOA), whereas anthropogenic-influenced areas (e.g., southeastern China and India) exhibit higher contributions from NO-mediated formation of C$_{10}$-ON (nitrate HOMs). Our findings advance the representation of organic aerosols in climate models and provide critical insights to bridge gaps between chamber experiments and global-scale simulations.



## 1 Introduction

Monoterpenes are one of the most significant classes of biogenic volatile organic compounds (BVOCs) (Guenther et al., 2012). Monoterpene-derived organic peroxy radicals (MT-RO$_2$) can undergo complex photochemical oxidation processes in the atmosphere. These processes can rapidly generate extremely low-volatility organic compounds (ELVOCs) through intramolecular autoxidation reactions that involve hydrogen-atom shifts and molecular oxygen attachment to form peroxy radicals (Ehn et al., 2014; Crounse et al., 2013; Praske et al., 2018; Bianchi et al., 2019). As the number of oxygen atoms in the functional group increases, the volatility of the organics gradually decreases. Organic compounds generated through rapid autoxidation reactions that contain six or more oxygen atoms are referred to as highly oxygenated organic molecules (HOMs), and can contribute to secondary organic aerosols (SOA) formation (Bianchi et al., 2019). The SOA formed by HOMs are referred to as HOMs-SOA.

Studies have shown that monoterpene-derived HOMs promote new particle formation (NPF) due to their low volatility, affecting the concentrations of the cloud condensation nuclei (CCN), ultimately influencing radiative forcing. Ehn et al. (2014) found that HOMs made important contributions to the growth of particles with diameters between 5 and 50 nm in northern forests. Jokinen et al. (2015) combined chamber experiments with global model simulations and found that monoterpene-derived HOMs promote NPF in continental regions, especially under high supersaturation conditions, thereby increasing CCN concentrations. HOMs account for 27~47% of SOA produced from oxygenated monoterpenes emitted by sage plants in California (Mehra et al., 2020). Airborne measurements above the Finnish boreal forest indicated that HOMs are distributed at the top of the boundary layer during the daytime (Beck et al., 2022). Moreover, a regional model also demonstrated that HOMs dominated the NPF at an altitude of 13 km in the Amazon region where human activities have less impact, significantly contributing to CCN formation (Zhao et al., 2020). Accordingly, Gordon et al. (2016) found through global model simulations that new particles formed by monoterpene-derived HOMs result in a 27% reduction in radiative forcing from -0.28 W/m$^2$ to -0.06 W/m$^2$ due to cloud albedo variation from the preindustrial to the present condition. Similarly, Zhu et al. (2019) found that simulated new particles formed by monoterpene-derived HOMs reduced direct and indirect radiative forcing by 12.5% since the Industrial Revolution.

Despite these past studies, the formation mechanism of monoterpene-derived HOMs remains uncertain in several aspects, including the reaction rate, yields of reactions (including autoxidation reactions, self-reactions and cross-reactions), and the impact of nitrogen dioxide (NO$_x$) on their generation process. Monoterpenes are emitted into the atmosphere and rapidly oxidized by OH radicals or O$_3$ to generate peroxy radicals (RO$_2$), but the proportion of RO$_2$ radicals that can further undergo autoxidation reactions is not yet clear (Berndt et al., 2016; Kurten et al., 2015; Richters et al., 2016; Roldin et al., 2019). In addition, the reaction rates of autoxidation reactions remain highly uncertain, with different measurements in different chamber experiments ranging from 0.6 to 21 /s, differing by 1 to 2 orders of magnitude (Lee et al., 2023; Berndt et al., 2016; Moller et al., 2020). After autoxidation reactions, RO$_2$ radicals can undergo self-reactions and cross-reactions, generating accretion



products with more carbon atoms and lower volatility. The yields and reaction rates of the accretion products also vary by one to two orders of magnitude in different experimental measurements (Berndt et al., 2018; Zhao et al., 2018). Moreover, $NO_x$ has a dual effect on HOMs formation. On the one hand, $NO_x$ can promote HOMs formation by enhancing atmospheric
oxidation and promoting alkoxyl radicals and subsequent $RO_2$ radicals, or even organic nitrate formation. On the other hand, $NO_x$ can terminate multi-generational oxidation reactions that generate HOMs. Nevertheless, due to insufficient experimental data, modeling calculations are needed to constrain the reaction kinetic parameters of these uncertain processes.

Several modeling studies have been conducted to simulate HOMs formation, and most of them focus on theoretical simulation. Pye et al. (2019) represented the chemical formation reactions of HOMs through the yield of important organic peroxy radicals
from chamber experiments. Roldin et al. (2019) developed a one-dimensional column model with a near-explicit mechanism of HOMs, though it has not been applied on a global scale. Weber et al. (2021) and Xu et al. (2022) developed and summarized these explicit formation mechanisms of monoterpenes-derived HOMs in global models but the models still lack fully understand the uncertainties. Recently, Zhao et al. (2024) advanced global modeling capabilities by comprehensively integrating 11 NPF mechanisms, revealing the critical role of organic-driven pathways in aerosol formation across diverse
regions.

In this study, the Community Atmosphere Model version 6 with comprehensive tropospheric and stratospheric chemistry (CAM6-Chem) has been revised with the chemical mechanism and gas-particle partitioning processes of HOMs, aiming to better understand their formation and spatiotemporal distribution. Section 2 introduces the model with the revised mechanism of HOMs and sensitivity experiments used in this study. Section 3 validates the revised model with field campaigns,
demonstrates the spatiotemporal characteristics of HOMs, explores the dominant formation pathways of HOMs, and discusses the uncertainties in HOM chemistry. Results are summarized and discussed in Section 4.

## 2 Data and methods

### 2.1 Model configuration

The Community Atmosphere Model version 6 with comprehensive tropospheric and stratospheric chemistry (CAM6-Chem)
from the Community Earth System Model version 2.1.0 (CESM2.1.0) is used in this study. The default configuration of CAM6-Chem employs the four-mode version of the Modal Aerosol Module (MAM4) (Liu et al., 2016) and applies the Volatility Basis Set (VBS) approach (Donahue et al., 2006; Hodzic et al., 2016; Jo et al., 2021; Robinson et al., 2007) to represent the formation of SOA from all volatile organic compounds (VOCs). All simulations are configured with a horizontal resolution of 0.95° in latitude and 1.25° in longitude and a vertical resolution of 32 layers up to approximately 40 km (Emmons
et al., 2020). Meteorological fields, including temperature, winds, and surface fluxes, from the Modern-Era Retrospective analysis for Research and Applications (MERRA2) reanalysis data set (Gelaro et al., 2017) are used for offline nudging to minimize uncertainties in meteorology simulation (Jo et al., 2021; Tilmes et al., 2019; Liu et al., 2021). Anthropogenic and



biomass burning emissions are from the standard Coupled Model Intercomparison Project 6 (CMIP6) (Eyring et al., 2016).
The biogenic emissions are simulated online using the Model of Emissions of Gases and Aerosol from Nature version2.1
(MEGAN2.1) (Guenther et al., 2012).

This study uses an updated version of the model to better represent the heterogeneous production and photolytic depletion of
SOA (Liu et al., 2023). One update involves coupling the model with the Model for Simulating Aerosol Interactions and
Chemistry (MOSAIC) (Zaveri et al., 2021), enabling explicit representation of the heterogeneous uptake of IEPOX onto sulfate
aerosols (Jo et al., 2019; Jo et al., 2021). Another update is the incorporation of a faster photolysis rate for monoterpene-
derived SOA (MTSOA) according to recent chamber measurements (Epstein et al., 2010; Zawadowicz et al., 2020; Henry and
Donahue, 2012) and modeling studies (Liu et al., 2021; Liu et al., 2023). A modest photolysis rate of MTSOA (2.0% of the
$NO_2$ photolysis frequency) replaces the original rate (0.04% of the $NO_2$ photolysis frequency) in the default CAM6-Chem
model (Hodzic et al., 2016), despite evidence suggesting that some MTSOA may resist degradation (Zawadowicz et al., 2020).
Except for MTSOA, the photolysis rates of other SOA are unchanged (kept at 0.04% of $NO_2$ photolysis frequency) in the
simulations due to a lack of chamber reports. HOM chemistry is also incorporated, including autoxidation reactions and self-
and cross-reactions for accretion products, as described in Section 2.2.2.

## 2.2 HOMs formation mechanisms

### 2.2.1 Extension of volatility basis set (VBS)

SOA are formed when emitted volatile organic compounds (VOCs) are oxidized in the atmosphere with subsequent gas-
particle partitioning processes or new particle formation. However, the physical and chemical properties of SOA are
complicated due to the variety of VOCs, oxidants, and formation mechanisms. To simplify and represent the formation
processes in the models, SOA and their gas-phase precursors (SOAG) are lumped based on their volatilities, following the
volatility basis set (VBS) approach (Donahue et al., 2006; Robinson et al., 2007; Shrivastava et al., 2015). In CAM6-Chem,
the volatilities of SOA and SOAG are categorized into five bins based on their saturation concentrations (C*) of 0.01, 0.1, 1.0,
10.0, and 100.0 μg/m³ at 298 K as shown in Table 1. The yield of SOAG from various VOCs, including isoprene, glyoxal,
monoterpenes (α-pinene, β-pinene, limonene, and myrcene), the β-caryophyllene surrogate sesquiterpene, benzene, toluene,
lumped xylenes, intermediate VOC (IVOC), and semi-VOC (SVOC), are based on Tilmes et al. (2019) and Jo et al. (2021)
and shown in Table S11. The SOAG in different volatility bin (SOAG0~4 in Table 1) condenses on the preexisting aerosols
to form SOA (soa1~5 in Table 1) based on their saturation vapor pressure calculated following Eq. (1) (Chung and Seinfeld,
125  2002):

$$P(T) = P(T_0) \cdot e^{\left[\frac{-\Delta H_{vap}}{R} \cdot \left(\frac{1}{T} - \frac{1}{T_0}\right)\right]}$$

(1)





where $P$ is the saturation vapor pressure at temperature $T$ and $T_0$=298 K; R is the ideal gas constant, and $\Delta H_{vap}$ is the enthalpies of vaporization which represents the energy to transform the liquid substance into gas phase (default parameterized values shown in Table 1).

The volatility bins of the default VBS scheme are too high to represent the formation processes of HOMs. The volatilities of most HOMs fall within the range of low volatility organic compounds (LVOCs) and extremely low-volatility organic compounds (ELVOCs) (Bianchi et al., 2019). Therefore, the original five volatility bins ($C^* = 0.01, 0.1, 1.0, 10.0,$ and $100.0$ µg/m³) are extended to eight bins, with the newly added bins ($C^* = $ 1.0e-3, 1.0e-5, 1.0e-9 µg/m³) explicitly representing the final products of HOMs chemistry (Table 1). The volatilities of these newly added HOMs are calculated based on their

molecular formula (Table S12) using Eq. (2) (Mohr et al., 2019).

$$log_{10}C^*(300\,K) =$$
$$(25 - n_C) \times b_C - (n_o - 3n_N) \times b_O - n_N \times b_N - 2\left[\frac{(n_o - 3n_N) \times n_C}{n_C + n_o - 3n_N}\right] \times b_{CO} \qquad (2)$$

where $n_C$, $n_o$, and $n_N$ are the number of carbon, oxygen, and nitrogen atoms; $b_C = 0.475$; $b_O = 0.2$; $b_N = 2.5$; $b_{CO} = 0.9$.

**Table 1.** $\Delta H_{vap}$ at eight VBS bins. The first five bins are the traditional VBS bins and the last three bins are extended for
HOMs mechanisms. $aC_{10}$-NON and $bC_{10}$-NON are the non-nitrate HOMs containing 10 carbons formed by $HO_2$ radical and NO pathways, respectively. $C_{10}$-ON are the nitrate HOMs containing 10 carbons. $C_{15}$ and $C_{20}$ are the HOMs containing 15 and 20 carbons, respectively.

| $C^*$ (µg/m³) | SOAG | SOA | $\Delta H_{vap}$ (kJ/mol) |
|---|---|---|---|
| $1.0 \times 10^{-2}$ | SOAG0 | soa1 | 153.0 |
| $1.0 \times 10^{-1}$ | SOAG1 | soa2 | 142.0 |
| 1.0 | SOAG2 | soa3 | 131.0 |
| $1.0 \times 10$ | SOAG3 | soa4 | 120.0 |
| $1.0 \times 10^2$ | SOAG4 | soa5 | 109.0 |
| $1.0 \times 10^{-3}$ | $aC_{10}$-NON (g) [a] | $aC_{10}$-NON (a) [b] | 164.0 |
| | $bC_{10}$-NON (g) [a] | $bC_{10}$-NON (a) [b] | |
| | $C_{10}$-ON (g) [a] | $C_{10}$-ON (a) [b] | |
| $1.0 \times 10^{-5}$ | $C_{15}$ (g) [a] | $C_{15}$ (a) [b] | 186.0 |
| $1.0 \times 10^{-9}$ | $C_{20}$ (g) [a] | $C_{20}$ (a) [b] | 230.0 |

[a] Gas-phase HOMs, corresponding to SOAGhma, SOAGhmb, SOAGhmn, SOAGac15, and SOAGac20 in the model (Table S12).

[b] Particle-phase HOMs, corresponding to soahma, soahmb, soahmn, soaac15, and soaac20 in the model (Table S12).



### 2.2.2 Newly added HOMs formation mechanisms

Figure 1 illustrates the flowchart of the HOMs mechanism implemented in CAM6-Chem in this study, based on Xu et al. (2022). In general, monoterpenes are oxidized by OH radicals or $O_3$ to form monoterpene-derived organic peroxy radicals (MT-$RO_2$). The MT-$RO_2$ formed by the oxidation of monoterpenes by $NO_3$ radicals are not considered in this study, as some studies report the branching ratio to be insignificant (Zhao et al., 2021; Nah et al., 2016; Yan et al., 2016), and the chemical process remains highly uncertain (Roldin et al., 2019).

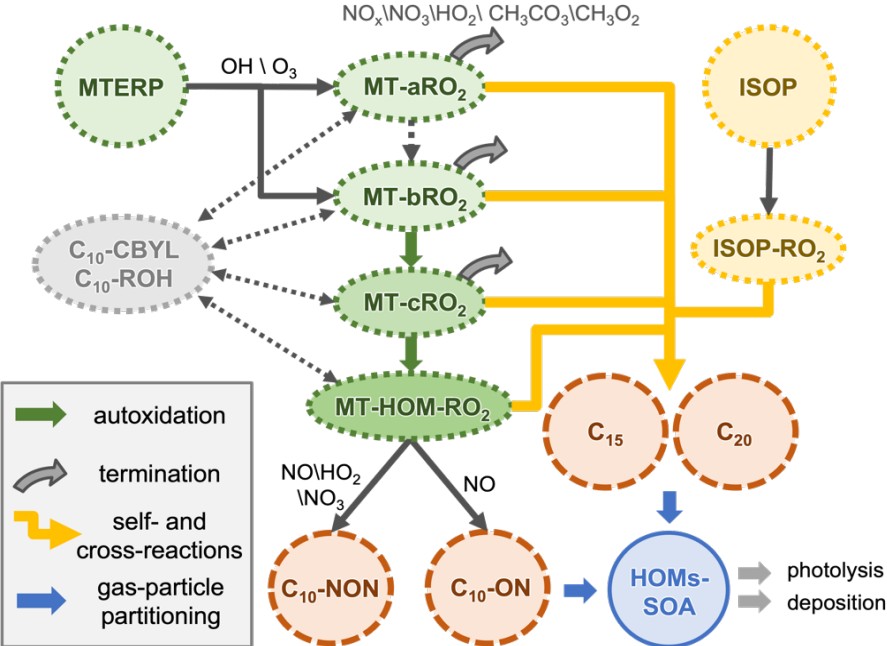

**Figure 1.** The flow chart of HOMs mechanism. The green arrows represent the autoxidation reactions. The gray curved solid arrows represent the termination reactions. The yellow arrows represent the self- and cross-reactions. The blue arrows represent the gas-particle partitioning processes. The gray dashed arrows represent the conversion between $C_{10}$-CBYL (Carbonyl with 10 carbon atoms)\$C_{10}$-ROH (Alcohol with 10 carbon atoms) and MT-$RO_2$ radicals.

Two types of MT-$RO_2$ radicals (MT-a$RO_2$ and MT-b$RO_2$) are formed from monoterpene + OH/$O_3$ reactions. MT-a$RO_2$ corresponds to the MT-$RO_2$ that cannot undergo autoxidation reactions in the original version of CAM6-Chem (with a yield of 100%). MT-b$RO_2$ corresponds to the newly added MT-$RO_2$ that can undergo further autoxidation reactions. The branching ratio of MT-a$RO_2$ and MT-b$RO_2$ radicals for monoterpene + OH reactions is 0.25:0.75 (Reactions 1-4 in Table S1), and for monoterpene + $O_3$ reactions, it is 0.92:0.08 (Reactions 5-8 in Table S1), which falls within the range reported in previous studies (Lee et al., 2023; Piletic and Kleindienst, 2022; Pye et al., 2019; Weber et al., 2021; Xu et al., 2019). The reaction rate constants used are the same as those in the default monoterpene + OH/$O_3$ reactions. MT-b$RO_2$ radicals can further undergo autoxidation to form MT-c$RO_2$ (1st autoxidation product) and MT-HOM-$RO_2$ (2nd autoxidation product). All MT-$RO_2$



radicals (MT-aRO₂, MT-bRO₂, MT-cRO₂, and MT-HOM-RO₂) undergo self- and cross-reactions to form accretion products (C₁₅ and C₂₀). MT-HOM-RO₂ can be directly terminated by several oxidants, including NO, NO₃ radicals, and HO₂ radicals, to form HOMs with 10 carbon atoms (C₁₀). The gas-phase HOMs will undergo gas-particle partitioning to form particle-phase HOMs (Table 1 and Eq. (1)). Particle-phase C₁₀ will be photolyzed (1.7% of the NO₂ photolysis frequency, Table S8), but the accretion products will not, due to large uncertainties (Zawadowicz et al., 2020; Xu et al., 2022). All HOMs are subject to wet

and dry deposition.

There are ten types of final HOMs products, five in the gas phase and five in the particle phase (see details in Table 1), along with many newly added intermediate products (Table S12). The volatilities of the newly added HOMs are calculated based on their molecular formulas (Table S12) using the method from Mohr et al. (2019) (Eq. (2)). In total, 24 original reactions are modified, and 96 new reactions are added (including 6 photolysis reactions) to simulate HOMs in CAM6-Chem. These include

autoxidation reactions, self- and cross-reactions for accretion products, and the termination of MT-RO₂ radicals. The detailed modifications to the original chemistry and the corresponding species are introduced in Text S1 and Tables S1-S9.

## 2.3 Sensitivity experiments

The formation of monoterpene-derived HOMs involves two key uncertainties: (1) the branching ratios of autoxidation-capable peroxy radicals (MT-bRO₂) formed via OH- and O₃-initiated oxidation (Lee et al., 2023; Weber et al., 2020; Pye et al., 2019;

Xu et al., 2019; Piletic and Kleindienst, 2022), and (2) the autoxidation rate of MT-bRO₂, which varies by over an order of magnitude in experimental studies (Berndt et al., 2018; Roldin et al., 2019; Weber et al., 2021). To systematically analyze these uncertainties, we conducted nine sensitivity experiments (Table 2). The Control experiment adopts the branching ratios from Xu et al. (2022) (MT-bRO₂: 75% for OH-initiated and 8% for O₃-initiated reactions), serving as a benchmark aligned with recent mechanistic frameworks. Four additional experiments (LowYield, HighYield, HighOH_LowO₃, LowOH_HighO₃)

span the full parameter space of MT-bRO₂ branching ratios reported in literature (OH: 7.5–83%; O₃: 0.01–22%) (Saunders et al., 2003; Roldin et al., 2019; Rolletter et al., 2019), while two experiments (Fast and Slow) explore autoxidation rate extremes (×10 and ×0.1 of the Control rate). To isolate pathway-specific uncertainties in the formation of nitrate HOMs containing 10 carbons (C₁₀-ON) (Bianchi et al., 2019; Yan et al., 2016; Xu et al., 2022; Weber et al., 2020), we further test NO-mediated HOM formation (no_HMB_NO) (Reaction 110 in Table S7). Besides, in comparison with the SENEX and BAECC field

campaigns, the simulated NO concentration in the Control experiment is overestimated by a factor of four (Figs. S1 and S2). Therefore, we multiplied the NOx emissions by 0.2 in the LowNOx experiment to assess the impact of anthropogenic NOx on HOM concentration. These experiments collectively quantify how mechanistic uncertainties propagate to HOMs predictions, bridging gaps between chamber-derived parameters and global model applications.

**Table 2.** Experiments used in this paper.

| Experiments | OH branching ratio | O₃ branching ratio | Autoxidation rate | RO pathway | NOx emissions |
|---|---|---|---|---|---|



| Control | 75% | 8% | $K_{auto}$ | √ | default |
|---|---|---|---|---|---|
| LowYield | 7.5% | 0.01% | | | |
| HighYield | 83% | 22% | | /a | |
| HighOH_lowO3 | 83% | 0.01% | | | |
| LowOH_HighO3 | 7.5% | 22% | | | |
| Fast | / | | $10 \times K_{auto}$ | / | |
| Slow | | | $0.1 \times K_{auto}$ | | |
| no_HMB_NO | / | | | Xb | / |
| LowNOx | | / | | | default/5 |

a The setting is the same as Control
b The yield of $bC_{10}$-NON is set to zero in the MT-HOM-$RO_2$ + NO reaction (reaction 110 in Table S7)

## 2.4 Observations

This study utilizes observational data from the Southeast Nexus (SENEX) (Warneke et al., 2016) and Biogenic Aerosols-Effects on Clouds and Climate (BAECC) (Petäjä et al., 2016) field campaigns (Table 3). The observed variables include NO, $O_3$, monoterpenes, isoprene (ISOP), HOMs with 10 carbon atoms ($C_{10}$), and particle-phase $C_{10}$ concentrations. Further details regarding the field campaigns can be found in Text S2 of Xu et al. (2022).

**Table 3.** Field campaigns used in this paper

| Campaigns | Dates | Locations |
|---|---|---|
| SENEX (Warneke et al., 2016) | 2013.06.01–07.15 | Centreville, Alabama, US (32.93°N, 87.13°W) |
| BAECC (Petäjä et al., 2016) | 2014.04.11–06.03 | Station for Measuring Ecosystem Atmosphere Relations (SMEAR II), Hyytiälä, Finland. (61.85°N,24.28°E) |

## 3 Results

### 3.1 Model evaluation

The revised model successfully captures the observed diurnal patterns of total HOMs ($C_{10}$-ON + $C_{10}$-NON), despite systematic biases in the concentrations of individual components. Most sensitivity tests underestimate nitrate HOMs ($C_{10}$-ON) by 20–





50% and overestimate non-nitrate HOMs ($C_{10}$-NON) by approximately 30% (Table 2), but they effectively reproduce the observed diurnal cycles (Figs. 2 and 3). Observations indicate that $C_{10}$-ON concentrations peak around midday, while $C_{10}$-NON shows minimal variation throughout the day. The simulations align well with these patterns: all experiments capture the

morning rise in total $C_{10}$-ON (Figs. 2b and 3b), driven by NO accumulation and increased photochemical activity (Figs. S1–S2). The diurnal evolution of the boundary layer height (BLH) also plays a significant role in modulating HOMs concentrations. The morning expansion of the BLH enhances vertical mixing, diluting HOMs near the surface, while its afternoon collapse concentrates HOMs in a shallower layer, amplifying their observed peaks. Meanwhile, $C_{10}$-NON trends remain relatively flat (within ±15% of daily means), as their formation is influenced by the competing effects of OH (morning

peak), $O_3$ (afternoon peak), and monoterpene emissions (which vary throughout the day) (Figs. S1–S2).

Simulations with increased MT-b$RO_2$ branching ratios and autoxidation rates (HighOH_LowO₃, HighYield, Fast) captured $C_{10}$-ON peak concentrations more accurately, whereas simulations with reduced branching ratios and autoxidation rates (LowYield, LowOH_HighO₃, Slow) showed better agreement with observed $C_{10}$-NON levels at both sites. Specifically, the HighOH_LowO₃, HighYield, Fast, and Control experiments overestimated total $C_{10}$ by ~50% (Figs. 2a and 3a) but reproduced

$C_{10}$-ON peaks within observational uncertainty (Figs. 2b and 3b). Conversely, the LowYield, LowOH_HighO₃, and Slow experiments underestimated total $C_{10}$ but aligned more closely with $C_{10}$-NON concentrations (Figs. 2c and 3c). These results underscore the necessity of tailoring MT-b$RO_2$ branching ratios to specific oxidation pathways ($C_{10}$-ON and $C_{10}$-NON) for accurate $C_{10}$ modeling. However, current uncertainties in autoxidation kinetics and the scarcity of pathway-resolved observational data (e.g., real-time measurements of $RO_2$ intermediates) limit our ability to further constrain these parameters

(Berndt et al., 2018; Weber et al., 2021; Xu et al., 2022). Addressing these gaps requires coordinated laboratory measurements and targeted ambient observations to disentangle competing chemical processes.

Compared to the dominant uncertainties in MT-b$RO_2$ branching ratios, the impacts of NOx emissions and NO-mediated $C_{10}$-NON formation pathways are less significant, though they provide complementary insights into HOM chemistry. In the LowNOx sensitivity experiment, total $C_{10}$ concentrations decreased from 736 to 339 ng/m$^3$ at the Centreville site

(anthropogenically influenced) due to reduced NOx emissions, with $C_{10}$-ON showing a more pronounced reduction (117 to 30 ng/m$^3$) than $C_{10}$-NON (619 to 310 ng/m$^3$), consistent with the NO-dependent formation of $C_{10}$-ON (Figs. 2 and 3). In contrast, at the SMEAR II site, total $C_{10}$ remained nearly unchanged (141 to 142 ng/m$^3$), reflecting minimal NOx influence in  the pristine region. Similarly, the no_HMB_NO experiment, which eliminates NO-mediated $C_{10}$-NON production, reduced $C_{10}$-NON concentrations by about 40% (619 to 398 ng/m$^3$ at Centreville; 112 to 57 ng/m$^3$ at SMEAR II) and improved agreement

with observations (Figs. 2c and 3c).



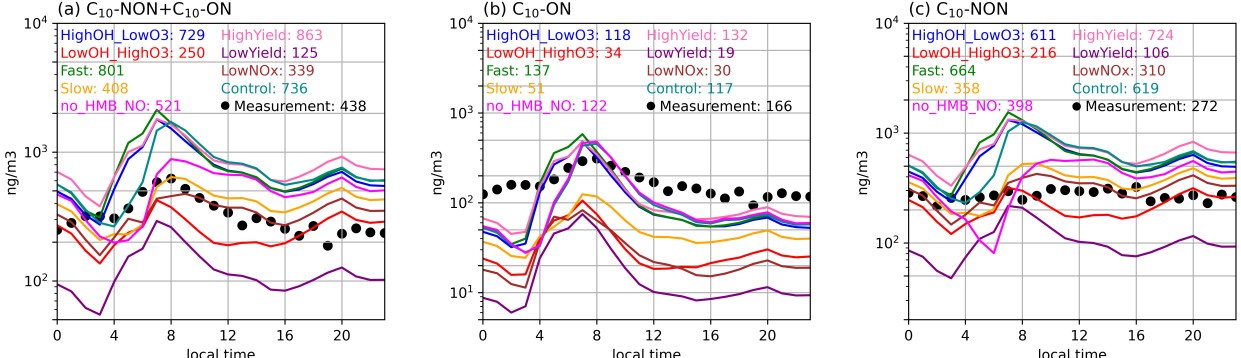

**Figure 2.** The diurnal cycle of observed (dots) surface and simulations (solid lines) total $C_{10}$ ($C_{10}$-ON+ $C_{10}$-NON) (a), $C_{10}$-ON (b) and $C_{10}$-NON (c) concentrations (unit: ng/m³) at the Centreville site during the SENEX campaign. The simulated surface $C_{10}$ ($C_{10}$-ON and $C_{10}$-NON) concentrations at the closest grid to the Centreville site are used from simulations (solid lines). The simulated $C_{10}$ at two sites are scaled by
the ratios of the observed monoterpene concentrations to the simulated monoterpene concentrations ($R_{MT}$, Figure S1a and S2a). The Normalized Mean Bias (NMB), Correlation Coefficient (R), and Root Mean Square Error (RMSE) values of $C_{10}$ comparing with observation are shown in Table S13. The $C_{10}$, $C_{10}$-NON and $C_{10}$-ON concentrations are the sum of gas-phase and particle-phase concentrations.

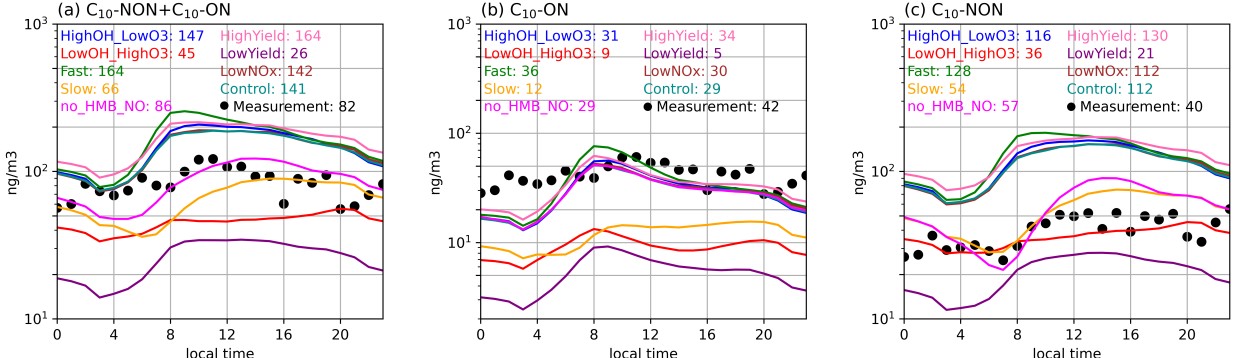

**Figure 3.** The diurnal cycle of observed (dots) surface $C_{10}$ ($C_{10}$-ON+ $C_{10}$-NON) (a), $C_{10}$-ON (b) and $C_{10}$-NON (c) concentrations (unit: ng/m³)
the SMEAR II sites during the BAECC campaign. The simulated surface $C_{10}$ ($C_{10}$-ON and $C_{10}$-NON) concentrations at the closest grid to the SMEAR II site are used from different simulations (solid lines). The simulated $C_{10}$ at two sites are scaled by the ratios of the observed monoterpene concentrations to the simulated monoterpene concentrations ($R_{MT}$, Figure S1a and S2a). The Normalized Mean Bias (NMB), Correlation Coefficient (R), and Root Mean Square Error (RMSE) values of $C_{10}$ comparing with observation are shown in Table S13. The $C_{10}$, $C_{10}$-NON and $C_{10}$-ON concentrations are the sum of gas-phase and particle-phase concentrations.

**3.2 Main formation pathways of HOMs**

Previous modeling studies on HOMs (Xu et al., 2022; Weber et al., 2021) have focused on isolated pathways or fixed parameters, limiting their capacity to resolve competing chemical mechanisms under varying environmental conditions. Our study advances this approach by conducting nine targeted sensitivity experiments that collectively investigate how MT-bRO₂ branching ratios, autoxidation rates, and NOx perturbations differentially influence the steering of HOMs toward distinct
formation pathways (Table 2).





$C_{10}$-ON and $C_{10}$-NON formation pathways exhibit distinct dependencies on oxidant availability and autoxidation kinetics. $C_{10}$-ON is exclusively formed via NO termination of MT-HOM-RO$_2$ radicals, whereas $C_{10}$-NON arises predominantly from MT-HOM-RO$_2$ reactions with HO$_2$ (~64%) and NO (~35%) (Fig. 4). This HO$_2$-dominated pathway to form $C_{10}$-NON remains robust across sensitivity tests (61~72% contribution), with the Slow experiment showing the highest HO$_2$ contribution. These

results align with Xu et al. (2022), who identified HO$_2$ as a key driver of non-nitrate HOMs, but extend their findings by quantifying NO's role in terminating $C_{10}$ pathways.

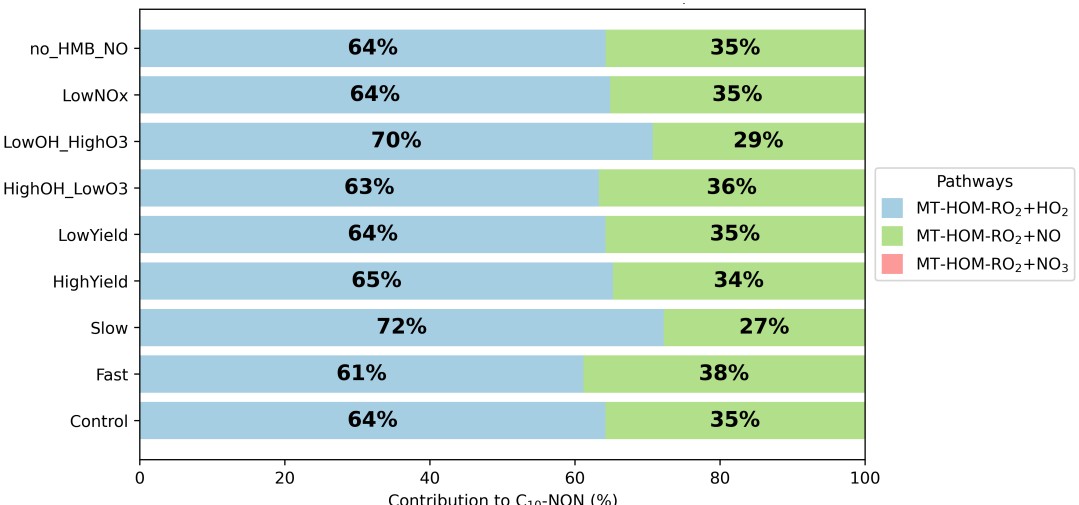

**Figure 4.** Contribution of different reaction pathways using different sensitivity tests (Table 2) to form $C_{10}$-NON (non-nitrate HOMs containing 10 carbons). "MT-HOM-RO$_2$+HO$_2$", "MT-HOM-RO$_2$+NO" and "MT-HOM-RO$_2$+NO$_3$" refer to Reactions 109, 110 and 111 in
Table S7.

The formation of $C_{15}$ and $C_{20}$ is governed by cross-reactions involving MT-RO$_2$ and isoprene-derived radicals (ISOP-RO$_2$), with sensitivity to MT-bRO$_2$ branching ratios and autoxidation rates. In the Control experiment, cross-reactions between MT-aRO$_2$/MT-HOM-RO$_2$ and ISOP-RO$_2$ account for 97% of $C_{15}$ (Fig. 5). When MT-bRO$_2$ branching ratios are enhanced (HighYield), MT-HOM-RO$_2$ concentrations increase, shifting $C_{15}$ formation from MT-aRO$_2$–ISOP-RO$_2$ (ranging from 35%

to 47%) to MT-HOM-RO$_2$–ISOP-RO$_2$ (ranging from 50% to 62%), while $C_{20}$ formation shifts from MT-aRO$_2$ self-reactions (ranging from 11% to 21%) to MT-HOM-RO$_2$ self-reactions (ranging from 27% to 38%).

Autoxidation rates critically regulate the formation pathways of accretion product generation by directly affecting the concentration of MT-HOM-RO$_2$. In the Slow experiment (×0.1 autoxidation rate), reduced MT-HOM-RO$_2$ production decreases its self-reaction contribution to $C_{20}$ formation by 44% (from 27% to 15%) and cross-reactions contribution to $C_{15}$ by

26% (from 50% to 37%). Conversely, the Fast experiment (with a 10-fold increase in the autoxidation rate) amplifies these contributions by 7% ($C_{20}$) and 4% ($C_{15}$), demonstrating a nonlinear response to rate changes. This differs from the approach of Weber et al. (2021), who assumed fixed branching ratios for accretion products and did not account for the dynamic interplay between autoxidation and cross-reaction kinetics highlighted in this study.



Anthropogenic NOx emissions and NO-mediated $C_{10}$-NON pathways have minimal influence on $C_{15}/C_{20}$ formation (<1%

variability), underscoring the dominance of MT-bRO$_2$ branching uncertainties. The LowNOx and no_HMB_NO experiments

show negligible changes in accretion pathways, as NOx perturbations primarily alter terminal products ($C_{10}$-ON/$C_{10}$-NON)

rather than radical pools. This contrasts with Xu et al. (2022), who emphasized NOx-driven HOM variability but did not isolate

its limited impact on accretion chemistry.

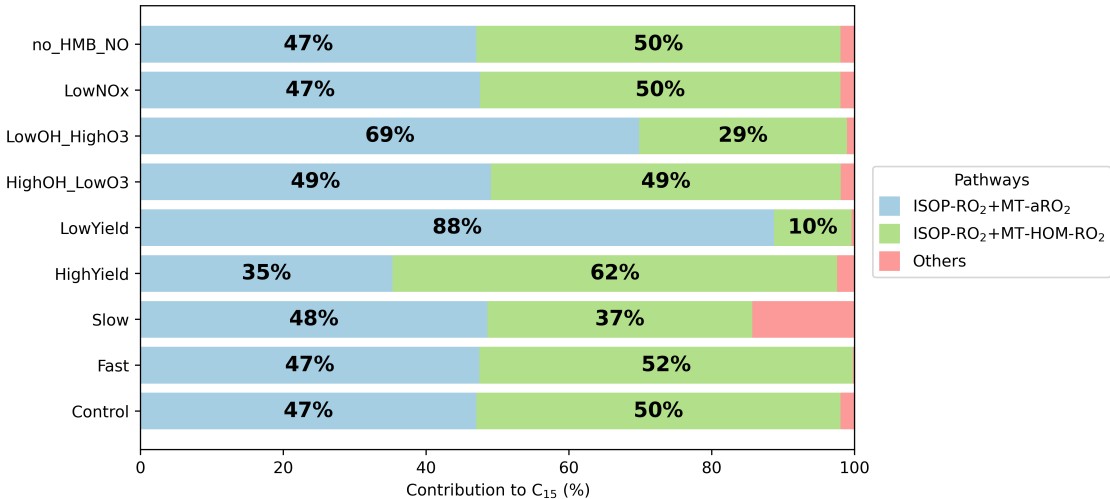

**Figure 5.** Contribution of different reaction pathways using different sensitivity tests (Table 2) to form $C_{15}$ (HOMs containing 15 carbons).
"ISOP-RO$_2$+MT-aRO$_2$" and "ISOP-RO$_2$+MT-HOM-RO$_2$" refer to Reactions 33-56 and 75-80 in Table S4. "others" refers to other reactions
forming $C_{15}$ in Table S4.

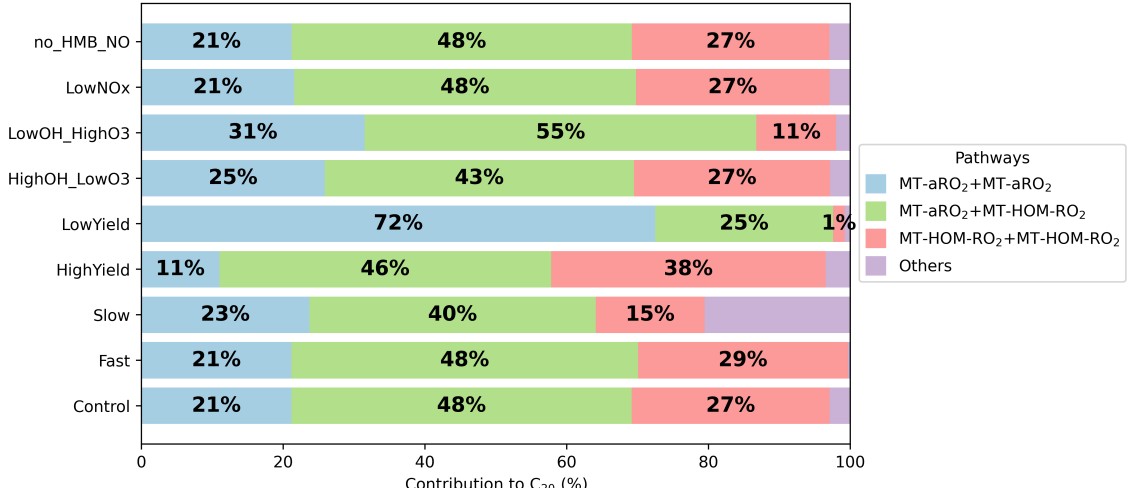

**Figure 6.** Contribution of different reaction pathways using different sensitivity tests (Table 2) to form $C_{20}$ (HOMs containing 20 carbons).
"MT-aRO$_2$+MT-aRO$_2$", "MT-aRO$_2$+MT-HOM-RO$_2$" and "MT-HOM-RO$_2$+MT-HOM-RO$_2$" refer to Reactions 11-20, 29-32 and 59 in
Table S4. "others" refers to other reactions forming $C_{20}$ in Table S4.



### 3.3 Spatial and temporal distribution of HOMs-SOA

To balance mechanistic completeness with observational consistency, the Control experiment, which incorporates median MT-bRO$_2$ branching ratios (Xu et al., 2022) and intermediate autoxidation rates, is selected as the baseline for
analyzing HOMs-SOA distributions. The Control experiment properly reproduces both diurnal cycles and mean values, indicating that the chemical mechanisms are reasonable, despite some biases. Building on this, we use sensitivity experiments (Table 2) to inform the uncertainties associated with the contribution of HOMs-SOA to MTSOA and total SOA.

Globally, HOMs-SOA (0.06 µg/m$^3$) contribute 36.6% of monoterpene-derived SOA and 10.7% of total SOA, with
spatial distributions tightly coupled to monoterpene emission hotspots (Figs. 7 and S4). Seasonal cycles mirror biogenic activity, peaking in summer (0.09 µg/m$^3$) and declining in winter (0.03 µg/m$^3$) (Fig. S6). Regionally, HOMs-SOA dominate monoterpene-SOA budgets (>20%) in key source areas, such as southeastern US (SEUS, 30°–37.5° N, 75°–95° W), southeastern China (SECN, 20°–35° N, 100°–122° E), Amazon (AMZ, 15° S–5° N, 50°–75° W), and other tropical continental regions (southeastern Asia and central Africa) with higher MTSOA concentrations (Figs. 7 and S3;
Table S10).

Four HOMs-SOA components display distinct regional characteristics (Fig. 8 and Table S10). Globally, C$_{10}$-NON are the largest proportion (54.6%), especially in polluted regions such as the southeastern United States and southeastern China, where it is driven by HO$_2$/NO termination pathways linked to anthropogenic oxidants. In contrast, in the Amazon, where NOx levels are low and RO$_2$ radical lifetimes are extended, cross-reactions with isoprene-derived RO$_2$ (ISOP-
RO$_2$) give rise to a higher concentration of C$_{15}$/C$_{20}$ accretion products, accounting for roughly 50% of HOMs-SOA.

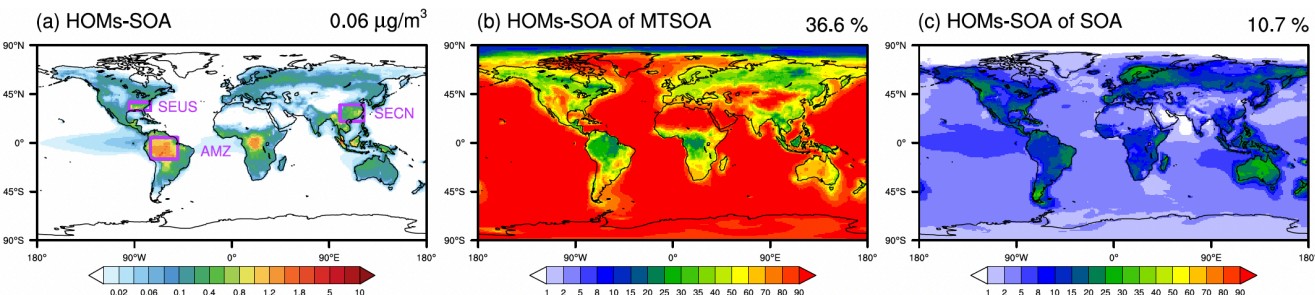

**Figure 7.** 2013 annual averaged surface (a) HOMs-SOA (unit: µg/m$^3$), (b) the contribution of HOMs-SOA to the MTSOA and (c) the contribution of HOMs-SOA to the total SOA (unit: %) in Control experiment. The global averaged value is shown in upper right corner of each figure. The three subregions (SEUS, SECN, and AMZ) are shown in the purple boxes in Figure 7a, respectively.






**Figure 8.** 2013 annual averaged HOMs-SOA components concentrations (left column, unit: μg/m³) and their contribution to total HOMs-SOA (right column, unit: %) in Control experiment. The global averaged value is shown in upper right corner of each figure.





The contribution of HOMs-SOA to total SOA varies significantly across different sensitivity experiments, ranging from 6%

to 15% (Fig. 9, Table S14), primarily controlled by the MT-bRO$_2$ branching ratio. The HighYield experiment (15%) and LowYield experiment (6%) encompass the uncertainty range reported in chamber studies (Xu et al., 2022), highlighting the dominant role of MT-bRO$_2$ branching in determining HOMs-SOA formation. Autoxidation has a secondary influence, as shown by the Fast (14%) and Slow (10%) experiments, which exhibit moderate variability consistent with rate-dependent parameterizations proposed by Weber et al. (2021). In contrast, NOx levels have a negligible effect (13% in LowNOx

experiment and 11% in no_HMB_NO experiment), which indicate that anthropogenic emissions do not significantly impact overall HOM-SOA burdens. Similarly, the contribution of HOMs-SOA to MTSOA ranges from 19% to 41% across different sensitivity experiments, with the highest proportion in the HighYield experiment and the lowest in the LowYield experiment, further reinforcing the dominant influence of MT-bRO$_2$ branching ratios.

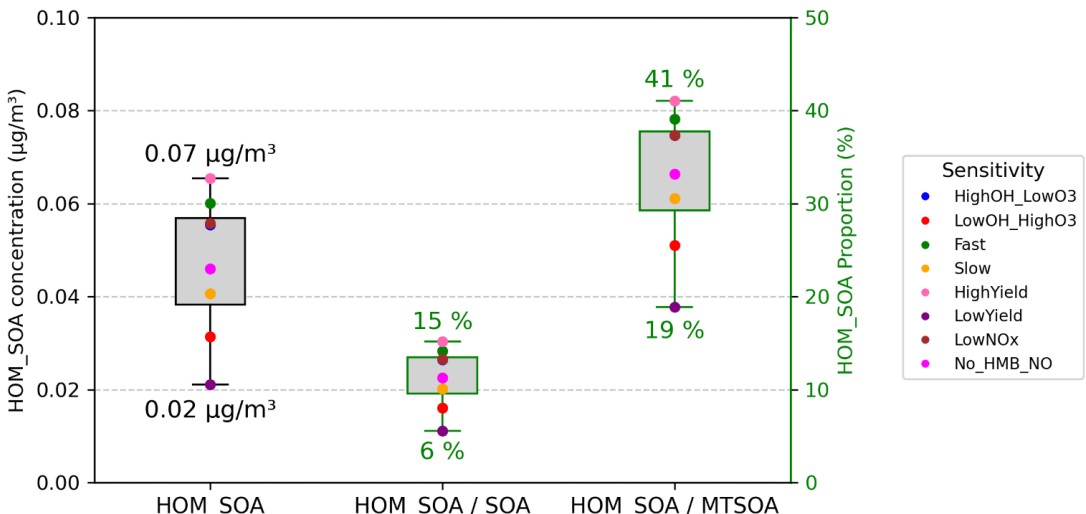

**Figure 9.** The global averaged value of 2013 annual mean surface HOMs-SOA (unit: $\mu g/m^3$), the contribution of HOMs-SOA to the total SOA and MTSOA (unit: %) using different sensitivity tests (Table 2). The specific value of different tests is shown in Table S14.

## 4 Conclusions

The advanced chemical mechanism is coupled with CAM6-Chem model to provide a comprehensive understanding of the formation and spatiotemporal distribution of Highly Oxygenated Organic Molecules (HOMs). Unlike fixed-yield or fixed-

branch approaches (Gordon et al., 2016; Gordon et al., 2017; Zhu et al., 2020; Weber et al., 2021; Zhao et al., 2024), our updated model with semi-explicit HOMs chemistry and sensitivity experiments captures pathway-specific dependencies, revealing nonlinear responses to the branching ratio of MT-bRO$_2$ (monoterpene-derived peroxy radicals which can undergo autoxidation), anthropogenic NOx emission and autoxidation kinetics. By resolving C$_{10}$ (HOMs containing 10 carbons), C$_{15}$ (HOMs containing 15 carbons), and C$_{20}$ (HOMs containing 20 carbons) component, we advance beyond lumped HOMs



classifications (e.g. "ELVOC" in Zhu et al. (2019) and Zhao et al. (2024)), enabling a more targeted analysis of contributions from various biogenic sources.

The improved model generally captures the observed diurnal cycles and average concentrations of $C_{10}$ ($C_{10}$-nitrate + $C_{10}$-non-nitrate). Specifically, when adjusting a higher branching ratio of MT-b$RO_2$ radicals or autoxidation rate, the model captures the timing of peak concentrations of $C_{10}$-nitrate ($C_{10}$-ON), accurately although overestimates the average $C_{10}$-non-nitrate ($C_{10}$-

NON) concentrations by approximately 50%. In contrast, when using lower branching ratio of MT-b$RO_2$ radicals or autoxidation rate, the model captures the values of $C_{10}$-NON relatively well but fails to agree with $C_{10}$-ON concentrations. These results highlight the need for further chamber experiments to accurately simulate the yields and reaction rates of different HOMs components.

The branching ratio of autoxidation-capable monoterpene radicals (MT-b$RO_2$) is the primary driver of uncertainties in HOMs

formation, accounting for 19–41% of the variability in HOMs-SOA contributions to monoterpene-derived SOA. While autoxidation rates modulate radical lifetimes, their impact is secondary and dependent on MT-b$RO_2$ branching thresholds. This hierarchy of controls may help resolve long-standing discrepancies between chamber-derived parameterizations and global model predictions, , offering insights toward addressing a key gap identified in previous studies (Xu et al., 2022; Weber et al., 2021). The branching ratio of MT-b$RO_2$ mainly influences the formation of accretion products ($C_{15}$/$C_{20}$), which are dominated

by cross-reactions between monoterpene- and isoprene-derived peroxy radicals, contributing approximately 50% of HOMs-SOA in pristine regions such as the Amazon. When the MT-b$RO_2$ branching ratio is higher, the concentrations of MT-HOM-$RO_2$ increase, resulting in more $C_{20}$ formed from MT-HOM-$RO_2$ self-reactions and more $C_{15}$ formed from ISOP-$RO_2$ + MT-HOM-$RO_2$ cross-reactions. This mechanism, underrepresented in earlier global models (Gordon et al., 2016; Zhu et al., 2019), highlights the competition between biogenic autoxidation and anthropogenic NOx-driven termination.

While $NO_x$ levels have minimal impact on total HOMs-SOA burdens, the formation pathways of $C_{10}$-ON and $C_{10}$-NON respond differently to NOx variations. Increasing NOx enhances $C_{10}$-ON production by promoting NO termination of MT-HOM-$RO_2$ radicals (e.g., 70% reduction in Centreville under the LowNOx experiment), while simultaneously suppressing $HO_2$-driven $C_{10}$-NON formation. This is contrasts with the "NOx-oxidant-HOM" positive feedback mechanism proposed by Pye et al. (2019). They demonstrated that although NOx reduction increases the relative efficiency of autoxidation (enhancing

$RO_2$ competition with NO), the concurrent decline in OH concentrations reduce the absolute HOM production rate. This leads to limited sensitivity of global HOMs-SOA burdens to NOx variations. Our results show that while NOx concentrations locally promote $C_{10}$-ON formation (e.g., in polluted regions), it simultaneously suppresses $C_{15}$/$C_{20}$ accretion product formation in pristine areas (e.g., the Amazon) by accelerating NO termination of MT-HOM-$RO_2$, inhibiting $RO_2$ autoxidation and cross-reactions. This leads to limited sensitivity of global HOMs-SOA burdens to NOx variations. This mechanistic decoupling, ,



not fully emphasized in earlier nucleation-focused studies (e.g., Gordon et al., 2016), highlights the possible value of representing NOx-HOMs interactions more explicitly in climate models.

Climate-driven increases in biogenic emissions may enhance HOMs-SOA production, but concurrent declines in anthropogenic NOx could shift the dominant formation pathway from $C_{10}$-ON to $C_{10}$-NON, which is a feedback mechanism currently absent in projections. Additionally, aromatic-derived HOMs, which dominate nucleation and growth in polluted

regions (Ren et al., 2021; Zhang et al., 2021; Shrivastava et al., 2024), remain underrepresented in global model. To address persistent gaps between model predictions and observations, field campaigns targeting accretion product speciation and chamber studies that constrain MT-bRO$_2$ branching ratios are needed.

**Competing interests.** At least one of the (co-)authors is a member of the editorial board of Atmospheric Chemistry and Physics.

**Data availability.** SENEX and BAECC field campaigns data are from (Xu et al., 2022).

**Author contributions.** XD and MW designed the study. XS and YL performed the data analysis, produced the figures, and wrote the manuscript draft. RX, JT and MY collected the dataset. WS, MS, SA, and KS contributed to the analysis methods. DJ provided the model. All the authors contributed to discussion, writing, and editing of the manuscript.

**Acknowledgments.** This work is supported by the National Natural Science Foundation of China [grant numbers 41925023, 2024YFC3711905, and 91744208]. This research was also supported by the Collaborative Innovation Center of Climate

Change, Jiangsu Province, and supported by the Frontiers Science Center for Critical Earth Material Cycling. We greatly thank the High Performance Computing Center of Nanjing University for providing the computational resources used in this work. Manish Shrivastava was supported by the U.S. Department of Energy (DOE) Office of Science, Office of Biological and Environmental Research (BER) through the Early Career Research Program (Grant KP1701010/72144) and DOE BER's Atmospheric System Research (ASR) program (Grant KP1701010/57131). The Pacific Northwest National Laboratory

(PNNL) is operated for DOE by Battelle Memorial Institute under contract DE-AC06-76RL01830. The CESM project is supported primarily by the United States National Science Foundation (NSF). This material is based upon work supported by the National Center for Atmospheric Research, which is a major facility sponsored by the NSF under Cooperative Agreement No. 1852977. We thank all the scientists, software engineers, and administrators who contributed to the development of CESM2.





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
