# Peer review of "A Modeling Study of Global Distribution and Formation Pathways of Highly Oxygenated Organic Molecules (HOMs) from Monoterpenes"

_EGUsphere, 2025_

## Referee Comment (RC2)

**General Assessment**

This manuscript presents a comprehensive modeling study of HOMs (Highly Oxygenated Organic Molecules) derived from monoterpenes using the CAM6-Chem global climate model. The authors implement a semi-explicit HOM chemical mechanism, extend the volatility basis set (VBS), and conduct a suite of sensitivity experiments to explore the impacts of branching ratios, autoxidation rates, and NOx levels on HOMs-SOA formation.

The study is timely and addresses a significant gap in the representation of HOMs in global models. The methodology is sound, and the model development is well-documented. However, a major weakness of the manuscript lies in the limited and underdeveloped discussion of the results, particularly those from the global model simulations. The current treatment of global HOMs-SOA distributions lacks the depth and detail expected for a study of this scope and importance. This significantly reduces the scientific value and interpretability of the findings.

For this reason, I recommend **major revisions**, with particular emphasis on expanding and deepening the analysis of the global model results. Additionally, improvements are needed in the clarity of some sections, including the presentation of the chemical mechanism.

**Major Comments**

1. **Expansion of the discussion on global model results**

   The current discussion of global HOMs-SOA distributions (Section 3.3) is relatively brief and lacks depth. I strongly recommend expanding this analysis and relocating it to a new Section 4 to allow for a more comprehensive and focused discussion, including:
   - A more detailed regional analysis beyond SEUS, SECN, and Amazon. For example, Europe, South Asia, and Africa are not discussed despite being relevant for OMs and/or biogenic and anthropogenic interactions.
   - A more thorough treatment of seasonality, including regional seasonal cycles and their drivers (e.g., oxidant levels, emissions).
   - A spatially resolved analysis of how HOMs-SOA distributions change across the nine sensitivity experiments. This could include difference maps or regional statistics showing the impact of branching ratios and autoxidation rates.
   - Consideration of vertical distributions of HOMs-SOA, especially given their relevance for new particle formation and cloud interactions.

2. **Clarity of Reaction Mechanism Description**

   The description of the HOMs chemical mechanism in Section 2.2.2 and Figure 1 is currently quite dense and could be made clearer with a few structural adjustments. I recommend breaking the flowchart in Figure 1 into sub-panels, each representing a distinct stage of the mechanism, and aligning the discussion in the text with these steps using corresponding subsections or paragraphs. Additionally, it would be helpful to include a summary table outlining the key reaction pathways and their roles in forming C10, C15, and C20 HOMs. Finally, I suggest adopting clearer chemical nomenclatures and ensuring that all species and acronyms used in figures and tables are clearly defined in this section.

3. **Justification for Excluding NO₃-Initiated HOMs**

The manuscript excludes $NO_3$-initiated HOMs due to uncertainties. While this is understandable, recent studies suggest this pathway may be more important than previously thought. Therefore, a brief discussion of their potential importance (especially in polluted nighttime conditions) would provide a perspective for this HOM formation pathway.

**Minor Comments**

- **Lines 45–46:** The introduction discusses the role of HOMs in radiative forcing, but this is not revisited in the results. Including even a qualitative discussion in the results or conclusions on how HOMs-SOA might influence regional or global climate would enhance the impact of the study.

- **Lines 89–90:** Please provide references for the models mentioned.

- **Lines 90–95:** Include details on the simulation period used in the study.

- **Lines 99–100:** Specify the total amounts of monoterpene and isoprene emissions considered in the simulations.

- **Lines 114–115:** Please rephrase this sentence, the current wording is difficult to follow.

- **Line 125:** In addition to the vapor pressure equation, please include the gas-particle partitioning equation used in the model.

- **Line 189:** The field campaigns referenced here have not yet been introduced. Consider moving this sentence or providing context earlier.

- **Section 2.4:** This section should be expanded. Please include more details about the field campaigns, the types of measurements conducted, the instruments used, and a brief summary of key findings relevant to the model evaluation.

- **Lines 225–226:** This sentence appears more appropriate for the conclusions section rather than the results.

- **Section 3.3:** Consider moving the global model results to a new standalone Section 4 to give them greater emphasis and allow for a more structured discussion.

- **Figures 7b–7c:** Consider applying a minimum threshold for MTSOA (7b) and SOA (7c) before calculating the contributions of HOMs. Reporting high percentage contributions in regions with negligible SOA concentrations may be misleading and could skew global averages.

- **Figures 8b, 8d, 8f, 8h:** The same recommendation applies as for Figures 7b and 7c.

- **Terminology:** Ensure consistent use of terms such as "C10-NON", "C10-ON", and "HOMs-SOA" throughout the manuscript and figures.

- **Supplement:** The supplementary material is well-organized and informative. It would be helpful to reference specific tables and figures more explicitly in the main text to guide the reader.

---

## Author Comment (AC1)

We are very grateful to the evaluations from the reviewers, which have allowed us to clarify and improve the manuscript. Below we addressed the reviewer comments, with the reviewer comments in black and our response in **blue**.

**Reply for the referee comment#1**

**General Comments:** HOMS are key players in atmospheric new particle formation and subsequent growth of secondary organic aerosol (SOA). They are formed through complex oxidation reaction networks and the process is called autoxidation. Due to the inherent complexity in gas-phase organic oxidation reactions combined with the multitude of uncertain and unknown branching pathways with minimal information of their number, efficiency and extent, describing HOM formation in atmospheric models is a formidable task. Hence, notwithstanding the importance of atmospheric autoxidation and direct aerosol precursor formation, describing it especially at the atmospheric scale remains an important barrier to overcome. Thus the work is novel, timely and aims to alleviate a persistent pain on the shoulders of the community. As such, it is certainly within scope of ACP and should be of interest to its readers. However, the inadequacies in reporting the work together with badly justified choice of parameters make the work untractable and not representative, and thus I can't propose publishing the work in ACP. I'll detail my concerns in the comments below.

**Response**: We sincerely thank the reviewer for their thoughtful and constructive assessment of our manuscript. We have carefully considered these comments and have undertaken substantial revisions to improve the clarity and robustness of our work. Specifically:

1. Reporting Improvements: We have thoroughly reorganized and expanded the methodology and results sections to more clearly and comprehensively describe the modeling framework and the improvements. In addition, we have incorporated schematic diagrams and flowcharts where appropriate to better guide the reader through the workflow and analytical steps.

2. Parameter Justification: We have provided more detailed explanations and references regarding the selection of key model parameters. Additionally, we explicitly discuss the associated uncertainties and their potential impact on the results through sensitivity analyses.

We hope that these revisions address the reviewer's concerns. A detailed account of all changes and specific responses is provided in the following section.

**Major comments:**

**The two major issues with the current work are i) the apparent deliberate choice of parameters and ii) the insufficient documentation of the work. These are detailed separately below.**

**About the choice of parameters:**

**Major Comment#1:** It seems the authors have chosen to use very high values for the crucial autoxidation parameters in several parts of the work, and thus it is not possible to assess the results at more realistic settings representing more atmospherically relevant conditions. The current study is probably closer to the maximum impact of monoterpenes on ambient HOM loads, though with the necessarily very reduced description of the oxidation chemistry in global models, no one could know today.

Importantly, the current work appears to almost completely hinge on the previous work of Xu et al., and uses its parameterizations without explaining the choices or what has actually been done. Whereas in Xu et al., there is a good discussion on the choice of parameters, here it is absent and the reader is left with very little information to understand the basis of the choices. It is also important to realize that the lower limit rate coefficients for autoxidation used here (i.e., around 0.1 s^-1 if I read it correctly) are already termed rapid rates in Xu et al., as they should be, as at around this rate the autoxidation is competitive in most atmospheric environments. In order to understand such a complex modelling work, it would be crucial to carefully detail the choice of parameters.

Related: How do you justify so high branching ratios to autoxidation, and why are they not described in the main text? Generally HOM yields have been found to lie between 0.1 and 7% of the VOC turnover.

**Response**: It is important to state that we agree with the reviewer that large uncertainties remain in the parameters, and we acknowledge this point in the manuscript. To address these uncertainties, we conduct sensitivity simulations, incorporating those related to key autoxidation parameters. Our current discussion of autoxidation parameter selection is presented in Text S1 of the Supporting Information. Given the importance of this topic, we have revised it in the updated manuscript the manuscript to include a more detailed explanation of the parameter choices in the main text, as outlined below:

*To account for the H-shift chemistry of MT-RO$_2$ leading to HOM formation (i.e., autoxidation), the first-generation monoterpene-derived RO$_2$ (MT-RO$_2$), formed via reactions of monoterpenes (MT) with OH or O$_3$, is classified into two categories: MT-aRO$_2$ and MT-bRO$_2$ (Fig. 1). Both categories undergo standard bimolecular reactions, but only MT-bRO$_2$ species*

*proceed through autoxidation. In contrast, MT-aRO₂ species (such as APINO₂, BPINO₂, LIMONO₂, and MYRCO₂, listed in Table S12) do not participate in autoxidation.*

*Relatively high branching ratios for the formation of MT-bRO₂ are adopted, based on the values used in Table S3 of Xu et al. (2022). Specifically, the branching ratio of MT-bRO₂ is 0.75 for monoterpene + OH reactions, and 0.08 for monoterpene + O₃ reactions (Fig. 1). These values fall within the ranges reported in previous studies. Literature-based yields for MT-bRO₂ range from 0.075 to 0.83 for OH-initiated reactions (Lee et al., 2023; Piletic and Kleindienst, 2022; Pye et al., 2019; Weber et al., 2020; Xu et al., 2019) and from 0 to 0.22 for O₃-initiated reactions (Ehn et al., 2014; Jokinen et al., 2015; Roldin et al., 2019; Berndt et al., 2016; Kurtén et al., 2015; Richters et al., 2016). The reaction rate constants for OH and O₃ oxidation of monoterpenes are the same as those used in the default mechanism (Table 3), and apply equally to the formation of both MT-aRO₂ and MT-bRO₂. This approach is fully consistent with the implementation in GEOS-Chem by Xu et al. (2022), who demonstrated that such simplification can reasonably reproduce the formation of HOMs and the fate of RO₂ radicals. Furthermore, studies by Roldin et al. (2019) and Weber et al. (2020) confirmed that using the same reaction rate for MT-bRO₂ and MT-aRO₂ also yields HOM concentrations that agree well with observations under forested conditions.*

*MT-bRO₂ are assumed to undergo one or multiple generations of autoxidation (Table 4). These reactions follow a temperature-dependent rate with an activation energy of 74.1 kJ/mol, consistent with previous studies (Lee et al., 2023; Möller et al., 2020; Pye et al., 2019; Roldin et al., 2019; Schervish and Donahue, 2020; Xu et al., 2019). The corresponding autoxidation rates are 0.27 s⁻¹ at 283 K, 1.30 s⁻¹ at 298 K, and 4.12 s⁻¹ at 310 K. The yield of HOMs depends on both the autoxidation rate and the fraction of MT-RO₂ that undergoes autoxidation. To reflect the uncertainty associated with these parameters, this fraction is varied in both OH- and O₃-initiated pathways as part of sensitivity experiments. A detailed discussion of these tests is provided in Section 2.3.*

[Figure]

**Figure 1.** *Schematic of monoterpene (MT) oxidation and subsequent autoxidation pathways. MT reacts with OH or O₃ to form MT-aRO₂ or MT-bRO₂, with the latter undergoing autoxidation steps to yield HOMs. Branching ratios are shown for OH and O₃ pathways.*

**Table 3.** *Initial oxidation reactions of four representative monoterpenes (APIN, BPIN, LIMON, and MYRC) with OH and O₃, leading to the formation of MT-aRO₂ (non-autoxidizable) and MT-bRO₂ (autoxidizable). Detailed descriptions of the intermediate species are provided in Table S12.*

| Index | Reactions | Reaction rate |
|---|---|---|
| 1 | APIN[a] + OH → 0.25*MT-aRO$_2$ + 0.75*MT-bRO$_2$ | 1.34e-11*exp(410/T) |
| 2 | BPIN[a] + OH → 0.25*MT-aRO$_2$ + 0.75*MT-bRO$_2$ | 1.62e-11*exp(460/T) |
| 3 | LIMON[a] + OH → 0.25*MT-aRO$_2$ + 0.75*MT-bRO$_2$ | 3.41e-11*exp(470/T) |
| 4 | MYRC[a] + OH → 0.25*MT-aRO$_2$ + 0.75*MT-bRO$_2$ | 2.1e-10 |
| 5 | APIN[a] + O$_3$ → 0.736*MT-aRO$_2$ + 0.064*MT-bRO$_2$ + 0.77*OH + 0.066*TERPA2O2 + 0.22*H$_2$O$_2$ + 0.044*TERPA + 0.002*TERPACID + 0.034*TERPA2 + 0.17*HO$_2$ + 0.17*CO + 0.27*CH$_2$O + 0.054*TERPA2CO3 | 1.34e-11*exp(410/T) |
| 6 | BPIN[a] + O$_3$ → 0.736*MT-aRO$_2$ + 0.064*MT-bRO$_2$ + 0.102*TERPK + 0.3*OH + 0.06*TERPA2CO3 + 0.32*H$_2$O$_2$ + 0.038*BIGALK + 0.19*CO$_2$ + 0.81*CH$_2$O + 0.11*HMHP + 0.08*HCOOH | 1.62e-11* exp(460/T) |
| 7 | LIMON[a] + O$_3$ → 0.736*MT-aRO$_2$ + 0.064*MT-bRO$_2$ + 0.66*OH + 0.132*TERPF1 + 0.33*CH$_3$CO$_3$ + 0.33*CH$_2$O + 0.066*TERPA3CO3 + 0.33*H$_2$O$_2$ + 0.002*TERPACID | 3.41e-11*exp(470/T) |
| 8 | MYRC[a] + O$_3$ → 0.736*MT-aRO$_2$ + 0.064*MT-bRO$_2$ + 0.2*TERPF2 + 0.63*OH + 0.63*HO$_2$ + 0.25*CH$_3$COCH$_3$ +0.39*CH$_2$O + 0.18*HYAC | 2.1e-10 |

[a] APIN, BPIN, LIMON, and MYRC represent α-pinene, β-pinene, limonene, and myrcene, respectively.

**Table 4.** *Autoxidation reactions of MT-bRO₂ leading to the formation of MT-cRO₂ and subsequently MT-HOM-RO₂.*

| Index | Reactions | Reaction rate |
|---|---|---|
| 9 | MT-bRO$_2$ → MT-cRO$_2$ | 9.8e12*exp(-8836/T) |
| 10 | MT-cRO$_2$ → MT-HOM-RO$_2$ | |

The autoxidation rate from the "Fast" experiment in Xu et al. (2022) was adopted as the "control" rate in this study. The lowest autoxidation rate constant used in the simulations is not 0.1 s⁻¹, but rather 0.1 times the control rate. Accordingly, the Fast experiment applies a rate 10 times higher than the control. This is explicitly stated in line 186, as shown below:

To assess the reliability of the autoxidation rates in the sensitivity experiments, the rates used in the Fast and Slow experiments were compared with those reported in previous chamber studies (Weber et al., 2019; Roldin et al., 2020). The results indicate that the Fast and Slow sensitivity experiments effectively cover the upper and lower bounds of the autoxidation rates observed in chamber experiments (see Figure and Table below).

[Figure]

**Figure.** Comparison of temperature-dependent autoxidation rate used in this study (solid yellow, red, and dashed red lines for the Control, Slow, and Fast experiments, respectively) with first- and multi-generation autoxidation rates derived from previous chamber studies (Weber et al., 2019; Roldin et al., 2020; Xu et al., 2022).

**Table.** Description of the autoxidation rates used in different studies (Roldin et al., 2019; Weber et al., 2020).

| Test Name | Reaction rate for generating | |
| --- | --- | --- |
| | MT-cRO$_2$ (first-generation autoxidation products) | MT-HOM-RO$_2$ (multi-generation autoxidation products) |
| Weber et al. (2020) | 1.009E9*exp(-6000/T) | 9.500E8*exp(-6000/T) |
| Roldin et al. (2019) | 7.768E17*exp(-12077/T) | 7.311E17*exp(-12077/T) |
| This study | 9.800E12*exp(-8836/T) | 9.800E12*exp(-8836/T) |

Our choice of high branching ratios for autoxidation has already been added in the main text (in abovemeionted response). It is important to note that the overall HOM yield and the branching ratio to autoxidation are not the same. The HOM yield is estimated by calculating the ratio of HOM concentration to monoterpene concentration in the atmosphere, which yields approximately 0.15% (see Figure below). This value falls within the 0.1% to 7% range mentioned by the reviewer.

[Figure]

**Figure.** Annual mean surface concentrations of HOMs (left), monoterpenes (middle), and their mass concentration ratio (HOM/monoterpene, right) in the Control simulation for 2013. The global mean ratio is shown in the right panel.

**Major Comment#2:** "The reaction rate constants used are the same as those in the default monoterpene + OH/O3 reactions." So, what are they exactly and how reasonable is their use here? How does the lumping affect the diurnal cycles for example, as the actual rates depend on individual $RO_2$ concentrations.

**Response**: We are sorry for any confusion caused by the original sentence. To clarify, we have now included Table 3 in the revised manuscript, which explicitly lists the rate constants used for the monoterpene + OH/$O_3$ reactions (see the last response).

Currently, most models adopt similar parameterization schemes (the lumping effect). To make this clearer for readers, the following description will be added in the main text (the underlined content is newly added or modified):

> *The reaction rate constants for OH and $O_3$ oxidation of monoterpenes are the same as those used in the default mechanism (Table 3), and apply equally to the formation of both MT-aRO₂ and MT-bRO₂. This approach is fully consistent with the implementation in GEOS-Chem by Xu et al. (2022), who demonstrated that such simplification can reasonably reproduce the formation of HOMs and the fate of RO₂ radicals. Furthermore, studies by Roldin et al. (2019) and Weber et al. (2020) confirmed that using the same reaction rate for MT-bRO₂ and MT-aRO₂ also yields HOM concentrations that agree well with observations under forested conditions.*

**Major Comment#3:** I don't understand how the autoxidation rate can "critically regulate the formation pathways of accretion product generation by directly affecting the concentration of MT-HOM-RO2". Autoxidation only converts R'O2 to R''O2, so how would it affect total RO2 abundance? Are you modelling the RO2 + RO2 with increasing k as a function of the oxygen content?

**Response**: We appreciate the reviewer's thoughtful comment. Autoxidation does not directly alter the total RO₂ abundance, but instead redistributes the RO₂ species, thereby affecting the distribution of formation pathways for accretion products. To clarify this, we have revised the corresponding sentence in the manuscript (Line 272) as follows (The underlined content is newly added or modified):

>  Autoxidation rates also influence the formation pathways of accretion products by affecting the distribution of peroxy radical intermediates. *In the Slow experiment (×0.1 autoxidation rate), reduced MT-HOM-RO₂ production decreases its self-reaction contribution to C₂₀ formation by 44% (from 27% to 15%) and cross-reactions contribution to C₁₅ by 26% (from 50% to 37%). Conversely, the Fast experiment (with a 10-fold increase in the autoxidation rate) amplifies these contributions by 7% (C₂₀) and 4% (C₁₅), demonstrating a nonlinear response to rate changes.*

[Figure]

***Figure 5.*** ** ***Figure 7****. Contribution of different reaction pathways using different sensitivity tests (Table 8) to form C₁₅ (HOMs containing 15 carbons). "ISOP-RO₂+MT-aRO₂" and "ISOP-RO₂+MT-HOM-RO₂" refer to Reactions 33-56 and 75-80 in Table 5. "Others" refers to other reactions forming C₁₅ in Table 5.*

[Figure]

***Figure 6.*** **_Figure 8_**_. Contribution of different reaction pathways using different sensitivity tests (Table 8) to form $C_{20}$ (HOMs containing 20 carbons). "MT-aRO$_2$+MT-aRO$_2$", "MT-aRO$_2$+MT-HOM-RO$_2$" and "MT-HOM-RO$_2$+MT-HOM-RO$_2$" refer to Reactions 11-20, 29-32 and 59 in Table 5. "others" refers to other reactions forming $C_{20}$ in Table 5._

**Major Comment#4:** Similarly this is a somewhat confusing statement "In addition, the reaction rates of autoxidation reactions remain highly uncertain, with different measurements in different chamber experiments ranging from 0.6 to 21 /s, differing by 1 to 2 orders of magnitude (Lee et al., 2023; Berndt et al., 2016; Moller et al., 2020)."

From this it appears that you are using too high autoxidation rates. Are the actual used rates described anywhere in the paper? 21 s^-1 is an exceptionally fast isomerization that outcompetes almost any RO2 loss process in almost any atmospheric environment. It is not representative number for general autoxidation in the atmosphere. It's also stated that autoxidation rates vary by 2 orders of magnitude, which is wrong. The meaningful variation is around 5 to 6 orders of magnitude (i.e., from around 10-4 to 100 s-1), but obviously the values can vary more than this.

**Response**: Sorry for the confusion caused by the original sentence. To clarify, we will add a new table (Table 3) and figure (Figure 1) in the revised manuscript to explicitly present the temperature-dependent autoxidation rate used in our simulations. We will also revise the relevant text for a clearer explanation, as follows:

> *MT-bRO$_2$ are assumed to undergo one or multiple generations of autoxidation (Table 4). These reactions follow a temperature-dependent rate with an activation energy of 74.1 kJ/mol, consistent with previous studies (Lee et al., 2023; Möller et al., 2020; Pye et al., 2019; Roldin et al., 2019; Schervish and Donahue, 2020; Xu et al., 2019). The corresponding autoxidation rate are 0.27 s⁻¹ at 283 K, 1.30 s⁻¹ at 298 K, and 4.12 s⁻¹ at 310 K.*

[Figure]

*Figure 1.* *Schematic of monoterpene (MT) oxidation and subsequent autoxidation pathways. MT reacts with OH or O₃ to form MT-aRO₂ or MT-bRO₂, with the latter undergoing successive autoxidation steps to yield MT-HOM-RO₂. Branching ratios are shown for OH and O₃ pathways.*

*Table 4.* *Autoxidation reactions of MT-bRO₂ leading to the formation of MT-cRO₂ and subsequently MT-HOM-RO₂.*

| Index | Reactions | Reaction rate |
|-------|-----------|---------------|
| 9 | $MT\text{-}bRO_2 \rightarrow MT\text{-}cRO_2$ | $9.8e12*exp(-8836/T)$ |
| 10 | $MT\text{-}cRO_2 \rightarrow MT\text{-}HOM\text{-}RO_2$ | |

We set the autoxidation rate in our sensitivity experiments to be 0.1 and 10 times the control value at a given temperature. However, when considering the full range of atmospheric temperatures in real general conditions, the resulting rate constants span approximately 5 orders of magnitude (from around $10^{-4}$ to $10$ $s^{-1}$, see Figure and Table below). Moreover, the "Fast" and "Slow" experiments effectively capture the uncertainty range reported in previous chamber studies (Roldin et al., 2019; Weber et al., 2020).

[Figure]

**Figure.** Comparison of temperature-dependent autoxidation rate used in this study (solid yellow, red, and dashed red lines for the Control, Slow, and Fast experiments, respectively) with first- and multi-generation autoxidation rates derived from previous chamber studies (Weber et al., 2019; Roldin et al., 2020; Xu et al., 2022).

**Table.** Description of the autoxidation rates used in different studies (Roldin et al., 2019; Weber et al., 2020).

| Test Name | Reaction rate for generating | |
|---|---|---|
| | MT-cRO$_2$ (first-generation autoxidation products) | MT-HOM-RO$_2$ (multi-generation autoxidation products) |
| Weber et al. (2020) | 1.009E9*exp(-6000/T) | 9.500E8*exp(-6000/T) |
| Roldin et al. (2019) | 7.768E17*exp(-12077/T) | 7.311E17*exp(-12077/T) |
| This study | 9.800E12*exp(-8836/T) | 9.800E12*exp(-8836/T) |

**Major Comment#5:** Moreover you say "The yields and reaction rates of the accretion products also vary by one to two orders of magnitude in different experimental measurements (Berndt et al., 2018; Zhao et al., 2018)."

Commonly RO2 + RO2 rates have been found to vary by over 6 orders of magnitude, which should be relevant for the RO2 +RO2 here as well. It appears that here all the RO2 + RO2 in the work have been given very high rate coefficients. Also the chosen CH3O2 rate coefficients seem strangely high (see e.g., https://doi.org/10.1016/j.atmosenv.2004.09.072)

**Response**: As shown in the Table below, the reactions involving CH$_3$O$_2$ (methylperoxy radicals) in our study are all between C$_{10}$-RO$_2$ and CH$_3$O$_2$. The reference you mentioned does not include reactions between CH$_3$O$_2$ and C$_{10}$ or C$_{10}$-RO$_2$ species; it primarily focuses on reactions involving CH$_3$O$_2$ with smaller RO$_2$ species, such as C$_5$ compounds. Therefore, the rate coefficients we use fall outside the scope of that study and are chosen based on reaction types more relevant to the high-carbon RO$_2$ species considered in this work.

**Table .** MT-RO$_2$ reactions with methylperoxy radicals. Detailed descriptions of the intermediate species are provided in Table S12.

| Reactions | Reaction rate |
|---|---|
| APINO$_2$ + CH$_3$O$_2$ → 0.05*MT-bRO$_2$ + 0.83*CH$_2$O + 0.133*TERPF1 + 0.399*TERPA + 0.19*TERPA3 + 0.1235*TERP1OOH + 0.17*CH$_3$OH + 0.1045*TERPK + 0.06*CH$_3$COCH$_3$ + 1.16*HO$_2$ | 2e-12 |
| BPINO$_2$ + CH$_3$O$_2$ → 0.05*MT-bRO$_2$ + 1.4*CH$_2$O + 0.3515*TERPF1 + 0.304*TERPK + 1.5*HO$_2$ + 0.08*CH$_3$COCH$_3$ + 0.2945*TERPA3 | 2e-12 |
| LIMONO$_2$ + CH$_3$O$_2$ → 0.05*MT-bRO$_2$ + 0.25*CH$_3$OH + 0.95*TERPF1 + 1.03*CH$_2$O + HO$_2$ | 2e-12 |
| MYRCO$_2$ + CH$_3$O$_2$ → 0.05*MT-bRO$_2$ + 0.25*CH$_3$OH + 0.95*TERPF2 + 0.75*CH$_2$O + HO$_2$ | 2e-12 |
| MT-bRO$_2$ \ MT-cRO$_2$ \ MT-HOM-RO$_2$ + CH$_3$O$_2$ → 0.15*CH$_3$OH + 0.85*CH$_2$O + 1.4*HO$_2$ + 0.7*HYDRALD + 0.7*CH$_3$COCH$_3$ + 0.15*C$_{10}$-ROH + 0.15*C$_{10}$-CBYL | 3.56e-14* exp(708/T) |

Based on a literature review, the uncertainty in reaction rate coefficients for highly oxygenated peroxy radicals to be within approximately two orders of magnitude. For shorter-chain $RO_2$ species, the uncertainty may be even greater, as noted in the reference you provided (https://doi.org/10.1016/j.atmosenv.2004.09.072). Accordingly, we have made the following revision in Line 67 (the underlined content is newly added or modified):

*The yields and reaction rates of the accretion products also vary by one to two orders of magnitude in different experimental measurements (Berndt et al., 2018; Zhao et al., 2018; Roldin et al., 2019; Baker et al., 2024; Zhao et al., 2017; Molteni et al., 2019), with reported values ranging from $5 \times 10^{-12}$ cm³ s⁻¹ (Baker et al., 2024) to $1 \times 10^{-10}$ cm³ s⁻¹ (Berndt et al., 2018).*

The rate coefficients selected in this study (Table 5) fall within this reported range.

**Table 5.** *Summary of the self- and cross-reactions involving MT-RO₂ and ISOP-RO₂ peroxy radicals considered in this study. Detailed descriptions of the intermediate species are provided in Table S12.*

| Index | Reactions | Reaction rate |
|---|---|---|
| 11–20 | $MT\text{-}aRO_2 + MT\text{-}aRO_2 \rightarrow 0.893*C_{10}\text{-}CBYL + 0.29*C_{10}\text{-}ROH + 0.603*HO_2 + 1.34*HYDRALD + 0.067*MT\text{-}bRO_2 + 0.04* C_{20}$ | 4.0e-11 |
| 21–24 | $MT\text{-}aRO_2 + MT\text{-}bRO_2 \rightarrow 0.96*C10\text{-}CBYL + 0.29*C10\text{-}ROH + 0.67*HO2 + 1.34*HYDRALD + 0.04* C_{20}$ | 4.0e-11 |
| 25–28 | $MT\text{-}aRO_2 + MT\text{-}cRO_2 \rightarrow 0.96*C10\text{-}CBYL + 0.29*C10\text{-}ROH + 0.67*HO2 + 1.34*HYDRALD + 0.04* C_{20}$ | 2.6e-10 |
| 29–32 | $MT\text{-}aRO_2 + MT\text{-}HOM\text{-}RO_2 \rightarrow 0.96*C10\text{-}CBYL + 0.29*C10\text{-}ROH + 0.67*HO2 + 1.34*HYDRALD + 0.04* C_{20}$ | 2.6e-10 |
| 33–56 | $MT\text{-}aRO_2 + ISOP\text{-}RO_2 \rightarrow 0.4465*C10\text{-}CBYL + 0.145*C10\text{-}ROH + 0.145*ROH + 0.603*HO2+1.485*HYDRALD+0.0335*MT\text{-}bRO_2+ 0.04* C_{15}$ | 2.0e-10 |
| 57 | $MT\text{-}bRO_2 + MT\text{-}bRO_2 \rightarrow 0.96*C_{10}\text{-}CBYL + 0.29*C_{10}\text{-}ROH + 0.67*HO_2 + 1.34*HYDRALD + 0.04* C_{20}$ | 4.0e-11 |
| 58 | $MT\text{-}cRO_2 + MT\text{-}cRO_2 \rightarrow 0.96*C_{10}\text{-}CBYL + 0.29*C_{10}\text{-}ROH + 0.67*HO_2 + 1.34*HYDRALD + 0.04* C_{20}$ | 2.6e-10 |
| 59 | $MT\text{-}HOM\text{-}RO_2 + MT\text{-}HOM\text{-}RO_2 \rightarrow 0.96*C_{10}\text{-}CBYL + 0.29*C_{10}\text{-}ROH + 0.67*HO_2 + 1.34*HYDRALD + 0.04* C_{20}$ | 2.6e-10 |
| 60 | $MT\text{-}bRO_2 + MT\text{-}cRO_2 \rightarrow 0.96*C_{10}\text{-}CBYL + 0.29*C_{10}\text{-}ROH + 0.67*HO_2 + 1.34*HYDRALD + 0.04* C_{20}$ | 2.6e-10 |

| | | |
|---|---|---|
| *61* | *MT-bRO₂ + MT-HOM-RO₂ → 0.96\*C₁₀-CBYL + 0.29\*C₁₀-ROH + 0.67\*HO₂ + 1.34\*HYDRALD + 0.04\* C₂₀* | *2.6e-10* |
| *62* | *MT-cRO₂ + MT-HOM-RO₂ → 0.96\*C₁₀-CBYL + 0.29\*C₁₀-ROH + 0.67\*HO₂ + 1.34\*HYDRALD + 0.04\* C₂₀* | *2.6e-10* |
| *63–68* | *MT-bRO₂ + ISOP-RO₂ → 0.48\*C₁₀-CBYL + 0.145\*C₁₀-ROH + 0.145\*ROH + 0.67\*HO₂ + 1.485\*HYDRALD + 0.04\* C₁₅* | *2.0e-11* |
| *69–74* | *MT-cRO₂ + ISOP-RO₂ → 0.48\*C₁₀-CBYL + 0.145\*C₁₀-ROH + 0.145\*ROH + 0.67\*HO₂ + 1.485\*HYDRALD + 0.04\* C₁₅* | *4.0e-11* |
| *75–80* | *MT-HOM-RO₂ + ISOP-RO₂ → 0.48\*C₁₀-CBYL + 0.145\*C₁₀-ROH + 0.145\*ROH + 0.67\*HO₂ + 1.485\*HYDRALD + 0.04\* C₁₅* | *4.0e-11* |

The reactions are written in the image as:

Reaction 61: $MT\text{-}bRO_2 + MT\text{-}HOM\text{-}RO_2 \rightarrow 0.96*C_{10}\text{-}CBYL + 0.29*C_{10}\text{-}ROH + 0.67*HO_2 + 1.34*HYDRALD + 0.04* C_{20}$, rate $2.6e\text{-}10$

Reaction 62: $MT\text{-}cRO_2 + MT\text{-}HOM\text{-}RO_2 \rightarrow 0.96*C_{10}\text{-}CBYL + 0.29*C_{10}\text{-}ROH + 0.67*HO_2 + 1.34*HYDRALD + 0.04* C_{20}$, rate $2.6e\text{-}10$

Reactions 63–68: $MT\text{-}bRO_2 + ISOP\text{-}RO_2 \rightarrow 0.48*C_{10}\text{-}CBYL + 0.145*C_{10}\text{-}ROH + 0.145*ROH + 0.67*HO_2 + 1.485*HYDRALD + 0.04* C_{15}$, rate $2.0e\text{-}11$

Reactions 69–74: $MT\text{-}cRO_2 + ISOP\text{-}RO_2 \rightarrow 0.48*C_{10}\text{-}CBYL + 0.145*C_{10}\text{-}ROH + 0.145*ROH + 0.67*HO_2 + 1.485*HYDRALD + 0.04* C_{15}$, rate $4.0e\text{-}11$

Reactions 75–80: $MT\text{-}HOM\text{-}RO_2 + ISOP\text{-}RO_2 \rightarrow 0.48*C_{10}\text{-}CBYL + 0.145*C_{10}\text{-}ROH + 0.145*ROH + 0.67*HO_2 + 1.485*HYDRALD + 0.04* C_{15}$, rate $4.0e\text{-}11$

**Major Comment#6:** Related, you mention you have modelled self and cross reactions of the accretion products, but I suppose this is not what you meant. "while two experiments (Fast and Slow) explore autoxidation rate extremes (~10 and ~0.1 of the Control rate)." What is the actual Control Rate?

**Response**: Sorry for the confusion. What we modeled are the self- and cross-reactions of biogenic peroxy radicals that lead to the formation of accretion products, not the self- and cross-reactions of the accretion products themselves. To avoid any misunderstanding, we have revised the original sentence in the main text (the underlined content is newly added or modified):

*HOM chemistry is also incorporated, including autoxidation reactions and self- and cross-reactions  of biogenic peroxy radicals forming accretion products, as described in Section 2.2.*

Additionally, revisions have been made to other areas that could be potentially ambiguous:

Line 67: "The yields and reaction rates of the accretion products" → "The yields and reaction rates to form accretion products"

Line 111 and Line 175: "self- and cross-reactions for accretion products" → "self- and cross-reactions to form accretion products"

Line 277: "fixed branching ratios for accretion products" → "fixed branching ratios to form accretion products"

As for the autoxidation rate in the Control experiment, the actual values used are now clarified in the revised manuscript in Table 4 (see details in the response to Major Comment 1).

**Major Comment#7:** "Building on this, we use sensitivity experiments (Table 2) to inform the uncertainties associated with the contribution of HOMs-SOA to MTSOA and total SOA". The Table 2 is hard to follow and thus it is not very clear what has been accomplished.

**Response:** Sorry for the confusion. Our main purpose here is to illustrate that several sensitivity experiments listed in Table 2, which primarily reflect chemical uncertainties related to HOMs, lead to variations in HOM concentrations. These changes subsequently affect HOMs-SOA concentrations and ultimately influence the contribution of HOMs-SOA to total SOA. To clarify, we have revised the sentence as follows (the underlined content is newly added or modified):

> *Building on this, we use the sensitivity experiments  listed in Table 2 to examine how variations in HOM concentrations influence their contributions to MTSOA and total SOA.*

We believe that combining Table 2 with the description in Lines 178 to 194 (see below) provides a clear overview of the sensitivity experiments conducted.

> *The formation of monoterpene-derived HOMs involves two key uncertainties: (1) the branching ratios of autoxidation-capable peroxy radicals (MT-bRO₂) formed via OH- and O₃-initiated oxidation (Lee et al., 2023; Weber et al., 2020; Pye et al., 2019; Xu et al., 2019; Piletic and Kleindienst, 2022), and (2) the autoxidation rate of MT-bRO₂, which varies by over an order of magnitude in experimental studies (Berndt et al., 2018; Roldin et al., 2019; Weber et al., 2021). To systematically analyze these uncertainties, we conducted nine sensitivity experiments (Table 2). The Control experiment adopts the branching ratios from Xu et al. (2022) (MT-bRO₂: 75% for OH-initiated and 8% for O₃-initiated reactions), serving as a benchmark aligned with recent mechanistic frameworks. Four additional experiments (LowYield, HighYield, HighOH_LowO3, LowOH_HighO3) span the full parameter space of MT-bRO₂ branching ratios reported in literature (OH: 7.5–83%; O₃: 0.01–22%) (Saunders et al., 2003; Roldin et al., 2019; Rolletter et al., 2019), while two experiments (Fast and Slow) explore autoxidation rate extremes (×10 and ×0.1 of the Control rate). To isolate pathway-specific uncertainties in the formation of nitrate HOMs containing 10 carbons (C₁₀-ON) (Bianchi et al., 2019; Yan et al., 2016; Xu et al., 2022; Weber et al., 2020), we further test NO-mediated HOM formation (no_HMB_NO) (Reaction 110 in Table 6). Besides, in comparison with the SENEX and BAECC field campaigns, the simulated NO concentration in the Control experiment is overestimated by a factor of four (Figs. S1 and S2). Therefore, we multiplied the NO emissions by 0.2 in the LowNO experiment to assess the impact of anthropogenic NO on HOM concentration. These experiments collectively quantify how mechanistic uncertainties propagate to HOMs predictions, bridging gaps between chamber-derived parameters and global model applications.*

**Table 2̶8̶.** *Experiments used in this paper.*

| Experiments | OH branching ratio | $O_3$ branching ratio | Autoxidation rate | RO pathway | NO emissions |
|---|---|---|---|---|---|
| Control | 75% | 8% | $K_{auto}$ [a] | √ | default |
| LowYield | 7.5% | 0.01% | | | |
| HighYield | 83% | 22% | | /[b] | |
| HighOH_lowO3 | 83% | 0.01% | | | |
| LowOH_HighO3 | 7.5% | 22% | | | |
| Fast | | / | $10 \times K_{auto}$ | / | |
| Slow | | | $0.1 \times K_{auto}$ | | |
| no_HMB_NO | | / | | X[c] | / |
| L̶o̶w̶N̶O̶x̶ LowNO | | / | | | default/5 |

[a] The specific values of $K_{auto}$ are provided in Table 4
[b] The setting is the same as Control
[c] The yield of b̶$C_{10}$-bNON is set to zero in the MT-HOM-RO$_2$ + NO reaction (reaction 110 in Table 6)

**About the documentation:**

**Major Comment#1:** The minimum requirement of reporting a research work is that the work needs to be repeatable with the information given. Evenmore, the work has to be repeatable with the information given in the main text, and the supportting material is there to avoid unnecessary repetition and too big tables, etc. One should not need to look at the supporting material to comprehend what is presented in the main text. With the current level of documentation, I don't know how I could repeat the work.

**Response**: The schematic figure in the main text was intended only as a conceptual overview of the dominant oxidation steps, not as a full kinetic mechanism. We now recognize that this may have caused some confusion, and in response, we have provided additional schematic diagrams and summary tables to highlight the most important reactions (The full set of reactions can be found in the Tables S1–S9 and S12 of Supplementary Information). The specific additions are as follows:

*2.2.2 Autoxidation*

*To account for the H-shift chemistry of MT-$RO_2$ leading to HOM formation (i.e., autoxidation), the first-generation monoterpene-derived $RO_2$ (MT-$RO_2$), formed via reactions of monoterpenes (MT) with OH or $O_3$, is classified into two categories: MT-a$RO_2$ and MT-b$RO_2$ (Fig. 1). Both categories undergo standard bimolecular reactions, but only MT-b$RO_2$ species proceed through autoxidation. In contrast, MT-a$RO_2$ species (such as $APINO_2$, $BPINO_2$, $LIMONO_2$, and $MYRCO_2$, listed in Table S12) do not participate in autoxidation.*

*Relatively high branching ratios for the formation of MT-b$RO_2$ are adopted, based on the values used in Table S3 of Xu et al. (2022). Specifically, the branching ratio of MT-b$RO_2$ is 0.75 for monoterpene + OH reactions, and 0.08 for monoterpene + $O_3$ reactions (Fig. 1). These values fall within the ranges reported in previous studies. Literature-based yields for MT-b$RO_2$ range from 0.075 to 0.83 for OH-initiated reactions (Lee et al., 2023; Piletic and Kleindienst, 2022; Pye et al., 2019; Weber et al., 2020; Xu et al., 2019) and from 0 to 0.22 for $O_3$-initiated reactions (Ehn et al., 2014; Jokinen et al., 2015; Roldin et al., 2019; Berndt et al., 2016; Kurtén et al., 2015; Richters et al., 2016). The reaction rate constants for OH and $O_3$ oxidation of monoterpenes are the same as those used in the default mechanism (Table 3), and apply equally to the formation of both MT-a$RO_2$ and MT-b$RO_2$. This approach is fully consistent with the implementation in GEOS-Chem by Xu et al. (2022), who demonstrated that such simplification can reasonably reproduce the formation of HOMs and the fate of $RO_2$ radicals. Furthermore, studies by Roldin et al. (2019) and Weber et al. (2020) confirmed that using the same reaction rate for MT-b$RO_2$ and MT-a$RO_2$ also yields HOM concentrations that agree well with observations under forested conditions.*

*MT-bRO₂ are assumed to undergo one or multiple generations of autoxidation (Table 4). These reactions follow a temperature-dependent rate with an activation energy of 74.1 kJ/mol, consistent with previous studies (Lee et al., 2023; Möller et al., 2020; Pye et al., 2019; Roldin et al., 2019; Schervish and Donahue, 2020; Xu et al., 2019). The corresponding autoxidation rate are $0.27\ s^{-1}$ at 283 K, $1.30\ s^{-1}$ at 298 K, and $4.12\ s^{-1}$ at 310 K. The yield of HOMs depends on both the autoxidation rate and the fraction of MT-RO₂ that undergoes autoxidation. To reflect the uncertainty associated with these parameters, this fraction is varied in both OH- and O₃-initiated pathways as part of sensitivity experiments. A detailed discussion of these tests is provided in Section 2.3.*

[Figure]

**Figure 1.** *Schematic of monoterpene (MT) oxidation and subsequent autoxidation pathways. MT reacts with OH or O₃ to form MT-aRO₂ or MT-bRO₂, with the latter undergoing autoxidation steps to yield HOMs. Branching ratios are shown for OH and O₃ pathways.*

**Table 3.** *Initial oxidation reactions of four representative monoterpenes (APIN, BPIN, LIMON, and MYRC) with OH and O₃, leading to the formation of MT-aRO₂ (non-autoxidizable) and MT-bRO₂ (autoxidizable). Detailed descriptions of the intermediate species are provided in Table S12.*

| Index | Reactions | Reaction rate |
|:---:|:---|:---:|
| 1 | APIN[a] + OH → 0.25*MT-aRO₂ + 0.75*MT-bRO₂ | 1.34e-11*exp(410/T) |
| 2 | BPIN[a] + OH → 0.25*MT-aRO₂ + 0.75*MT-bRO₂ | 1.62e-11*exp(460/T) |
| 3 | LIMON[a] + OH → 0.25*MT-aRO₂ + 0.75*MT-bRO₂ | 3.41e-11*exp(470/T) |
| 4 | MYRC[a] + OH → 0.25*MT-aRO₂ + 0.75*MT-bRO₂ | 2.1e-10 |
| 5 | APIN[a] + O₃ → 0.736*MT-aRO₂ + 0.064*MT-bRO₂ + 0.77*OH + 0.066*TERPA2O₂ + 0.22*H₂O₂ + 0.044*TERPA + 0.002*TERPACID + 0.034*TERPA2 + 0.17*HO₂ + 0.17*CO + 0.27*CH₂O + 0.054*TERPA2CO₃ | 1.34e-11*exp(410/T) |
| 6 | BPIN[a] + O₃ → 0.736*MT-aRO₂ + 0.064*MT-bRO₂ + 0.102*TERPK + 0.3*OH + 0.06*TERPA2CO₃ + 0.32*H₂O₂ + 0.038*BIGALK + 0.19*CO₂ + 0.81*CH₂O + 0.11*HMHP + 0.08*HCOOH | 1.62e-11* exp(460/T) |

| | | |
|---|---|---|
| 7 | $LIMON^a + O_3 \rightarrow 0.736*MT\text{-}aRO_2 + 0.064*MT\text{-}bRO_2 + 0.66*OH +$ $0.132*TERPF1 + 0.33*CH_3CO_3 + 0.33*CH_2O +$ $0.066*TERPA3CO_3 + 0.33*H_2O_2 + 0.002*TERPACID$ | $3.41e\text{-}11*exp(470/T)$ |
| 8 | $MYRC^a + O_3 \rightarrow 0.736*MT\text{-}aRO_2 + 0.064*MT\text{-}bRO_2 + 0.2*TERPF2 +$ $0.63*OH + 0.63*HO_2 + 0.25*CH_3COCH_3 + 0.39*CH_2O +$ $0.18*HYAC$ | $2.1e\text{-}10$ |

[a] *APIN, BPIN, LIMON, and MYRC represent α-pinene, β-pinene, limonene, and myrcene, respectively.*

**Table 4.** *Autoxidation reactions of MT-bRO$_2$ leading to the formation of MT-cRO$_2$ and subsequently MT-HOM-RO$_2$.*

| Index | Reactions | Reaction rate |
|---|---|---|
| 9 | $MT\text{-}bRO_2 \rightarrow MT\text{-}cRO_2$ | $9.8e12*exp(\text{-}8836/T)$ |
| 10 | $MT\text{-}cRO_2 \rightarrow MT\text{-}HOM\text{-}RO_2$ | |

**2.2.3 Self-Reactions and Cross-Reactions**

*Due to isomers of MT-RO$_2$ and ISOP-RO$_2$, self- and cross-reactions are included (Table 5), with three branches considered for the products. First, intermediate products are produced and are lumped as C$_{10}$-ROH and C$_{10}$-CBYL. Second, RO radicals are generated, which may produce HO$_2$ and C$_{10}$-CBYL or decompose into smaller compounds. Half of the RO radicals are assumed to decompose into smaller carbonyls. Third, accretion products (C$_{15}$ and C$_{20}$) are produced. The branching ratios of the three pathways above are set as 0.29:0.67:0.04, respectively (Xu et al., 2022). However, for the self- and cross-reactions involving MT-aRO$_2$ (APINO$_2$, BPINO$_2$, LIMONO$_2$, and MYRCO$_2$ in Table S12) and ISOP-RO$_2$, a small fraction of RO radicals may undergo a unimolecular H-shift to form MT-bRO$_2$, with the branching ratio set to 0.05 (Xu et al., 2022). The fast reaction rate is applied here based on Table S4 in Xu et al. (2022).*

[Figure]

**Figure 2.** *Schematic illustration of self- and cross-reactions between MT-RO$_2$ and ISOP-RO$_2$ peroxy radicals.*

**Table 5.** *Summary of the self- and cross-reactions involving MT-RO$_2$ and ISOP-RO$_2$ peroxy radicals considered in this study. Detailed descriptions of the intermediate species are provided in Table S12.*

| Index | Reactions | Reaction rate |
|-------|-----------|---------------|
| 11–20 | MT-aRO$_2$ + MT-aRO$_2$ → 0.893*C$_{10}$-CBYL + 0.29*C$_{10}$-ROH + 0.603*HO$_2$ + 1.34*HYDRALD + 0.067*MT-bRO$_2$ + 0.04* C$_{20}$ | 4.0e-11 |
| 21–24 | MT-aRO$_2$ + MT-bRO$_2$ → 0.96*C10-CBYL + 0.29*C10-ROH + 0.67*HO2 + 1.34*HYDRALD + 0.04* C$_{20}$ | 4.0e-11 |
| 25–28 | MT-aRO$_2$ + MT-cRO$_2$ → 0.96*C10-CBYL + 0.29*C10-ROH + 0.67*HO2 + 1.34*HYDRALD + 0.04* C$_{20}$ | 2.6e-10 |
| 29–32 | MT-aRO$_2$ + MT-HOM-RO$_2$ → 0.96*C10-CBYL + 0.29*C10-ROH + 0.67*HO2 + 1.34*HYDRALD + 0.04* C$_{20}$ | 2.6e-10 |
| 33–56 | MT-aRO$_2$ + ISOP-RO$_2$ → 0.4465*C10-CBYL + 0.145*C10-ROH + 0.145*ROH + 0.603*HO2+1.485*HYDRALD+0.0335*MT-bRO$_2$+ 0.04* C$_{15}$ | 2.0e-10 |
| 57 | MT-bRO$_2$ + MT-bRO$_2$ → 0.96*C$_{10}$-CBYL + 0.29*C$_{10}$-ROH + 0.67*HO$_2$ + 1.34*HYDRALD + 0.04* C$_{20}$ | 4.0e-11 |
| 58 | MT-cRO$_2$ + MT-cRO$_2$ → 0.96*C$_{10}$-CBYL + 0.29*C$_{10}$-ROH + 0.67*HO$_2$ + 1.34*HYDRALD + 0.04* C$_{20}$ | 2.6e-10 |
| 59 | MT-HOM-RO$_2$ + MT-HOM-RO$_2$ → 0.96*C$_{10}$-CBYL + 0.29*C$_{10}$-ROH + 0.67*HO$_2$ + 1.34*HYDRALD + 0.04* C$_{20}$ | 2.6e-10 |
| 60 | MT-bRO$_2$ + MT-cRO$_2$ → 0.96*C$_{10}$-CBYL + 0.29*C$_{10}$-ROH + 0.67*HO$_2$ + 1.34*HYDRALD + 0.04* C$_{20}$ | 2.6e-10 |
| 61 | MT-bRO$_2$ + MT-HOM-RO$_2$ → 0.96*C$_{10}$-CBYL + 0.29*C$_{10}$-ROH + 0.67*HO$_2$ + 1.34*HYDRALD + 0.04* C$_{20}$ | 2.6e-10 |
| 62 | MT-cRO$_2$ + MT-HOM-RO$_2$ → 0.96*C$_{10}$-CBYL + 0.29*C$_{10}$-ROH + 0.67*HO$_2$ + 1.34*HYDRALD + 0.04* C$_{20}$ | 2.6e-10 |
| 63–68 | MT-bRO$_2$ + ISOP-RO$_2$ → 0.48*C$_{10}$-CBYL + 0.145*C$_{10}$-ROH + 0.145*ROH + 0.67*HO$_2$ + 1.485*HYDRALD + 0.04* C$_{15}$ | 2.0e-11 |
| 69–74 | MT-cRO$_2$ + ISOP-RO$_2$ → 0.48*C$_{10}$-CBYL + 0.145*C$_{10}$-ROH + 0.145*ROH + 0.67*HO$_2$ + 1.485*HYDRALD + 0.04* C$_{15}$ | 4.0e-11 |
| 75–80 | MT-HOM-RO$_2$ + ISOP-RO$_2$ → 0.48*C$_{10}$-CBYL + 0.145*C$_{10}$-ROH + 0.145*ROH + 0.67*HO$_2$ + 1.485*HYDRALD + 0.04* C$_{15}$ | 4.0e-11 |

**2.2.4 $C_{10}$ HOMs formation**

When MT-HOM-RO$_2$ are oxidized by HO$_2$, NO, or NO$_3$ (Fig. 3), three types of gas-phase $C_{10}$ HOMs are formed: two types of $C_{10}$ non-nitrate HOMs ($C_{10}$-aNON and $C_{10}$-bNON) and $C_{10}$ nitrate HOMs ($C_{10}$-ON), as shown in Table 6. The rate constants used are the same as those for the MT-RO$_2$ + HO$_2$, NO, and NO$_3$ reactions in Xu et al. (2022).

[Figure]

**Figure 3**. *Schematic diagram illustrating the oxidation of MT-HOM-RO$_2$ by HO$_2$, NO, or NO$_3$, leading to the formation of three types of gas-phase $C_{10}$-HOMs.*

**Table 6.** $C_{10}$ HOMs formation. Detailed descriptions of the intermediate species are provided in Table S12.

| Index | Reactions | Reaction rate |
|---|---|---|
| 109 | MT-HOM-RO$_2$ + HO$_2$ → $C_{10}$-aNON + O$_2$ | 1.5e-11 |
| 110 | MT-HOM-RO$_2$ + NO →   0.8*NO$_2$ + 0.8*HO$_2$ + 0.4* $C_{10}$-bNON + 0.8*HYDRALD + 0.2* $C_{10}$-ON | 4.0e-12 |
| 111 | MT-HOM-RO$_2$ + NO$_3$ → HO$_2$ + NO$_2$ + 0.5* $C_{10}$-ON + HYDRALD | 1.2e-12 |

**Major Comment#2:** You talk about "comprehensive HOMs chemical mechanism", but you are only showing a crude and rather ambiguous schematic of a handful of reaction steps that you apply for the whole pool of monoterpenes. This is really not a mechanism, which has a very specific meaning in the chemical literature. If there is no real base mechanism, then the involvement of NOx is even harder to understand. The NOx involvement seems to be particularly important for the current work, yet only the final results in the form of formed products seem to be represented and the mechanistic steps are not discussed. I would have really liked to see more discussion around the chemistry, which should be at the heart and sould of this work based on the title.

Figure 1 actually proposes a rather complex reaction chemistry but the text says you use 5 gaseous and 5 particle phase HOM in total. Where is this mismatch coming from?

**Response**: I agree that "comprehensive" is not appropriate, so we have used "a lumped HOMs mechanism" instead. While referring to it as a "comprehensive" chemical mechanism may not be

entirely accurate, we believe this constitutes a chemical mechanism, including reaction pathways, rate constants, and temperature dependence. These mechanisms (Figures 1-3 and Tables 3-6 in the last response) can be validated or reproduced in different models.

The influence of NO is first considered through the MOZART mechanism, and then further addressed by the scheme we implemented in this study. We did not consider the full impact of NO, but rather focused on the influence of NO on HOM formation. This is reflected in the termination reactions, which can lead to the formation of organic nitrates ($C_{10}$-ON) and non-organic nitrates ($C_{10}$-bNON) (Fig. 3). While we do not aim to propose a box-model mechanism here, we have expanded the discussion with more detailed sensitivity simulations around the chemistry.

[Figure]

**Figure 3**. Schematic diagram illustrating the oxidation of MT-HOM-RO₂ by HO₂, NO, or NO₃, leading to the formation of three types of gas-phase $C_{10}$ HOMs.

The uncertainties associated with NO are mainly discussed using two sensitivity experiments: no_HMB_NO (which eliminates NO-mediated $C_{10}$-bNON production due to large uncertainties in this reaction) (Bianchi et al., 2019; Yan et al., 2016; Xu et al., 2022; Weber et al., 2020) (Reaction 110 in Table 6) and LowNO (where NO emissions are reduced to 0.2 of the original value). The discussion of uncertainty in the formation pathway for both experiments can be found in lines 227 to 235 and lines 280 to 284 of the manuscript:

> *Compared to the dominant uncertainties in MT-bRO₂ branching ratios, the impacts of $NO_x$ emissions and NO-mediated $C_{10}$-NON formation pathways are less significant, though they provide complementary insights into HOM chemistry. In the LowNO$_x$ sensitivity experiment, total $C_{10}$ concentrations decreased from 736 to 339 ng/m³ at the Centreville site (anthropogenically influenced) due to reduced $NO_x$ emissions, with $C_{10}$-ON showing a more pronounced reduction (117 to 30 ng/m³) than $C_{10}$-NON (619 to 310 ng/m³), consistent with the NO-dependent formation of $C_{10}$-ON (Figs. 4 and 5). In contrast, at the SMEAR II site, total $C_{10}$ remained nearly unchanged (141 to 142 ng/m³), reflecting minimal $NO_x$ influence in the pristine region. Similarly, the no_HMB_NO experiment, which eliminates NO-mediated $C_{10}$-NON production, reduced $C_{10}$-NON concentrations by about 40% (619 to 398 ng/m³ at Centreville; 112 to 57 ng/m³ at SMEAR II) and improved agreement with observations (Figs. 4c and 5c).*

*The LowNO and no_HMB_NO experiments show negligible changes in accretion pathways, as NO perturbations primarily alter terminal products ($C_{10}$-ON/$C_{10}$-bNON) rather than radical pools. This contrasts with Xu et al. (2022), who emphasized NO-driven HOM variability but did not isolate its limited impact on accretion chemistry.*

**Table 6.** *$C_{10}$ HOMs formation. Detailed descriptions of the intermediate species are provided in Table S12.*

| Index | Reactions | Reaction rate |
|---|---|---|
| 109 | MT-HOM-RO$_2$ + HO$_2$ → C$_{10}$-aNON + O$_2$ | 1.5e-11 |
| 110 | MT-HOM-RO$_2$ + NO →  0.8*NO$_2$ + 0.8*HO$_2$ + 0.4* C$_{10}$-bNON + 0.8*HYDRALD + 0.2* C$_{10}$-ON | 4.0e-12 |
| 111 | MT-HOM-RO$_2$ + NO$_3$ → HO$_2$ + NO$_2$ + 0.5* C$_{10}$-ON + HYDRALD | 1.2e-12 |

The discussion on the uncertainty of the contribution of HOMs-SOA to total SOA in the two experiments can be found in the original manuscript from lines 324 to 326:

*NO levels have a negligible effect (13% in LowNO experiment and 11% in no_HMB_NO experiment), which indicate that anthropogenic emissions do not significantly impact overall HOMs-SOA burdens.*

The statement "use 5 gaseous and 5 particle-phase HOMs in total" refers to the five categories of final HOMs formed. After the partitioning of HOMs between the gas and particle phases, they generate 5 categories of SOA. There are numerous intermediate products in the entire reaction mechanism, all of which are summarized in Table S12. To avoid confusion, we will split Table 1.

The original Table 1 is as follows:

**Table 1.** *$\Delta H_{vap}$ at eight VBS bins. The first five bins are the traditional VBS bins and the last three bins are extended for HOMs mechanisms. a$C_{10}$-NON and b$C_{10}$-NON are the non-nitrate HOMs containing 10 carbons formed by HO$_2$ radical and NO pathways, respectively. $C_{10}$-ON are the nitrate HOMs containing 10 carbons. $C_{15}$ and $C_{20}$ are the HOMs containing 15 and 20 carbons, respectively.*

| $C^*$ ($\mu g/m^3$) | SOAG | SOA | $\Delta H_{vap}$ (kJ/mol) |
|---|---|---|---|
| 1.0 x 10$^{-2}$ | SOAG0 | soa1 | 153.0 |
| 1.0 x 10$^{-1}$ | SOAG1 | soa2 | 142.0 |
| 1.0 | SOAG2 | soa3 | 131.0 |
| 1.0 x 10 | SOAG3 | soa4 | 120.0 |

| | | | |
|---|---|---|---|
| $1.0 \times 10^2$ | SOAG4 | soa5 | 109.0 |
| | $aC_{10}$-NON (g) [a] | $aC_{10}$-NON (a) [b] | |
| $1.0 \times 10^{-3}$ | $bC_{10}$-NON (g) [a] | $bC_{10}$-NON (a) [b] | 164.0 |
| | $C_{10}$-ON (g) [a] | $C_{10}$-ON (a) [b] | |
| $1.0 \times 10^{-5}$ | $C_{15}$ (g) [a] | $C_{15}$ (a) [b] | 186.0 |
| $1.0 \times 10^{-9}$ | $C_{20}$ (g) [a] | $C_{20}$ (a) [b] | 230.0 |

[a] Gas-phase HOMs, corresponding to SOAGhma, SOAGhmb, SOAGhmn, SOAGac15, and SOAGac20 in the model (Table S12).

[b] Particle-phase HOMs, corresponding to soahma, soahmb, soahmn, soaac15, and soaac20 in the model (Table S12).

The Table 1 will be revised as follows:

**Table 1.** *The saturated vapor concentration (C\*) and vaporization enthalpies (ΔH$_{vap}$) of SOAG (SOA precursor gas) at the traditional VBS bins.*

| $C^*$ ($\mu g/m^3$) | SOAG | $\Delta H_{vap}$ (kJ/mol) |
|---|---|---|
| $1.0 \times 10^{-2}$ | SOAG0 | 153.0 |
| $1.0 \times 10^{-1}$ | SOAG1 | 142.0 |
| $1.0$ | SOAG2 | 131.0 |
| $1.0 \times 10$ | SOAG3 | 120.0 |
| $1.0 \times 10^2$ | SOAG4 | 109.0 |

Also, we added a new Table 2 to characterize the newly generated HOMs:

**Table 2**. *The saturated vapor concentration (C\*) and vaporization enthalpies (ΔH$_{vap}$) of HOMs.*

| $C^*$ ($\mu g/m^3$) | $\Delta H_{vap}$ (kJ/mol) | Short Name |
|---|---|---|
| $1.0 \times 10^{-3}$ | 164.0 | $C_{10}$-aNON |
| | | $C_{10}$-bNON |
| | | $C_{10}$-ON |
| $1.0 \times 10^{-5}$ | 186.0 | $C_{15}$ |
| $1.0 \times 10^{-9}$ | 230.0 | $C_{20}$ |

**Major Comment#3:** Please use actual molecular compositions and not symbolic language. "TERP1OOH" is hardly a chemical name.

**Response**: Thank you for your valuable comment. It is not feasible for us to use actual molecular compositions throughout the manuscript because many intermediate products share identical molecular formulas, which would make it difficult to distinguish them in the reaction schemes (see Table S12 for details). Therefore, we have compiled Table S12 to list the molecular formulas and provide clear descriptions for all intermediate species. In addition, we have added the note "*Detailed descriptions of the intermediate species are provided in Table S12*" to the caption of each table presenting chemical reactions.

**Major Comment#4:** The photolysis assumptions. You say that photolysis of accretion products is not considered, but based on first principles they should be even more photosensitive as they contain the parent compound cromophores together with the added peroxide bond. Right? Or do you expect the HOM photochemistry to change considerably by addition of the peroxide bond? Also, shouldn't the photolysis frequency go down with the secondary particle size (i.e., shielding) or not? It is also unclear to me that where do you base the particle phase photolysis frequency that is as high as 1/60 of the jNO2?

**Response**: The reason for assuming either 1/60 of the $j_{NO_2}$ or no photolysis for accretion products is explained in the main text, lines 167 to 169 (as shown below):

*The gas-phase HOMs will undergo gas-particle partitioning to form particle-phase HOMs (Table 1 and Eq. (1)). Particle-phase C10 will be photolyzed (1.7% of the $NO_2$ photolysis frequency, Table S8), but the accretion products will not, due to large uncertainties (Zawadowicz et al., 2020; Xu et al., 2022).*

In our original simulations, we did not consider the decrease in photolysis frequency with increasing particle size (due to shielding effects). Although a previous study (Murphy et al., 2023) suggested that the addition of peroxide bonds may make compounds more susceptible to photolysis, there is currently no reliable chamber experiment providing an accurate photolysis rate for accretion products. Given the large uncertainty associated with this process, we added a sensitivity test in which accretion products are assumed to photolyze at the same rate as particle-phase C10 HOMs (i.e., 1.7% of the $NO_2$ photolysis rate). We have also included a corresponding explanation in the Section 5 (Conclusion) of the main text (see below) to help readers better understand the potential impact of this uncertainty on the simulation results.

*To assess the potential influence of photolysis uncertainties, we performed a sensitivity experiment assuming that accretion products photolyze at the same rate as particle-phase C10 HOMs (i.e., 1.7% of the $NO_2$ photolysis rate). While the impact on smaller species such as*

*$C_{10}$-ON and $C_{10}$-NON was negligible (<0.1%), substantial reductions were observed for $C_{15}$ and $C_{20}$ accretion products (75.2% and 68.1%, respectively) (Fig. S6). Overall, the total HOMs-SOA decreased by approximately 25.3% globally, highlighting that assumptions about photolysis rates can significantly affect model estimates of HOMs-SOA.*

[Figure]

***Figure S6.*** *Global distributions of particle-phase HOMs in the control simulation (left column), the absolute changes due to accretion product photolysis (ACC_photolysis - Control; middle column), and the relative changes (right column). Each row represents a specific HOM category: $C_{10}$-ON, $C_{10}$-NON, $C_{15}$, $C_{20}$, and the total.*

**Major Comment#5:** Unclear how the species would react together in Figure 1. Also, in each and every Figure you should explain all the names and symbols used. As an example, the Figure 2 is terribly hard to understood with the details given and one is left pondering about the numbers in them.

**Response**: Thank you for your comment. We have split Figure 1 into several smaller panels and added new textual descriptions and chemical reaction equations for each subfigure (see our response to Comment 1 for details). We apologize for the lack of clarity in the captions for Figures 2 and 3. We will revise both captions to more clearly convey the intended information. The underlined content in the revised captions indicates newly added or modified text (the underlined content is newly added or modified):

[Figure]

 **Figure 4.** ~~The diurnal cycle of observed (dots) surface and simulations (solid lines) total $C_{10}$ ($C_{10}$-ON + $C_{10}$-NON) (a), $C_{10}$-ON (b) and $C_{10}$-NON (c) concentrations (unit: ng/m$^3$) at the Centreville site during the SENEX campaign. The simulated surface $C_{10}$ ($C_{10}$-ON and $C_{10}$-NON) concentrations at the closest grid to the Centreville site are used from simulations (solid lines). The simulated $C_{10}$ at two sites are scaled by the ratios of the observed monoterpene concentrations to the simulated monoterpene concentrations ($R_{MT}$, Figure S1a and S2a). The Normalized Mean Bias (NMB), Correlation Coefficient (R), and Root Mean Square Error (RMSE) values of $C_{10}$ comparing with observation are shown in Table S13. The $C_{10}$, $C_{10}$-NON and $C_{10}$-ON concentrations are the sum of gas-phase and particle-phase concentrations.~~ Diurnal variations of observed (dots) and simulated (solid lines) surface concentrations of (a) total $C_{10}$ ($C_{10}$-aON + $C_{10}$-bON + $C_{10}$-NON), (b) $C_{10}$-aON + $C_{10}$-bON, and (c) $C_{10}$-NON at the Centreville site. Simulations are scaled by the observed-to-simulated monoterpene ratios (see Figures S1a and S2a). All concentrations (ng/m$^3$) include both gas and particle phases. Numbers shown represent daily mean values. Sensitivity experiment information is provided in Table 8. Model performance metrics (NMB, R, RMSE) are provided in Table S13.

[Figure]

**Figure 3.** **Figure 5.** ~~The diurnal cycle of observed (dots) surface C₁₀ (C₁₀-ON+ C₁₀-NON) (a), C₁₀-ON (b) and C₁₀-NON (c) concentrations (unit: ng/m³) the SMEAR II sites during the BAECC campaign. The simulated surface C₁₀ (C₁₀-ON and C₁₀-NON) concentrations at the closest grid to the SMEAR II site are used from different simulations (solid lines). The simulated C₁₀ at two sites are scaled by the ratios of the observed monoterpene concentrations to the simulated monoterpene concentrations (R_MT, Figure S1a and S2a). The Normalized Mean Bias (NMB), Correlation Coefficient (R), and Root Mean Square Error (RMSE) values of C₁₀ comparing with observation are shown in Table S13. The C₁₀, C₁₀-NON and C₁₀-ON concentrations are the sum of gas-phase and particle-phase concentrations.~~ _Diurnal variations of observed (dots) and simulated (solid lines) surface concentrations of (a) total C₁₀ (C₁₀-aON + C₁₀-bON + C₁₀-NON), (b) C₁₀-aON + C₁₀-bON, and (c) C₁₀-NON at the SMEAR II site. Simulations are scaled by the observed-to-simulated monoterpene ratios (see Figures S1a and S2a). All concentrations (ng/m³) include both gas and particle phases. Numbers shown represent daily mean values. Sensitivity experiment information is provided in Table 8. Model performance metrics (NMB, R, RMSE) are provided in Table S13._

**Major Comment#6:** Explain all the terms and symbols in Table and Figure captions. For example, can't understand Table 2.

**Response**: Thank you for pointing this out. We have carefully revised all table and figure captions to clearly define all terms, abbreviations, and symbols used (see our response to Comment 1 for details). Regarding the description of Table 2, it is mentioned in the main text from lines 178 to 194 (the underlined content is newly added or modified):

_The formation of monoterpene-derived HOMs involves two key uncertainties: (1) the branching ratios of autoxidation-capable peroxy radicals (MT-bRO₂) formed via OH- and O₃-initiated oxidation (Lee et al., 2023; Weber et al., 2020; Pye et al., 2019; Xu et al., 2019; Piletic and Kleindienst, 2022), and (2) the autoxidation rate of MT-bRO₂, which varies by over an order of magnitude in experimental studies (Berndt et al., 2018; Roldin et al., 2019; Weber et al., 2021). To systematically analyze these uncertainties, we conducted nine sensitivity experiments (Table 2). The Control experiment adopts the branching ratios from Xu et al. (2022) (MT-bRO₂: 75% for OH-initiated and 8% for O₃-initiated reactions), serving as a benchmark aligned with recent mechanistic frameworks. Four additional experiments (LowYield, HighYield, HighOH_LowO3, LowOH_HighO3) span the full parameter space of_

*MT-bRO$_2$ branching ratios reported in literature (OH: 7.5–83%; O$_3$: 0.01–22%) (Saunders et al., 2003; Roldin et al., 2019; Rolletter et al., 2019), while two experiments (Fast and Slow) explore autoxidation rate extremes (×10 and ×0.1 of the Control rate). To isolate pathway-specific uncertainties in the formation of nitrate HOMs containing 10 carbons (C$_{10}$-ON) (Bianchi et al., 2019; Yan et al., 2016; Xu et al., 2022; Weber et al., 2020), we further test NO-mediated HOM formation (no_HMB_NO) (Reaction 110 in Table 6). Besides, in comparison with the SENEX and BAECC field campaigns, the simulated NO concentration in the Control experiment is overestimated by a factor of four (Figs. S1 and S2). Therefore, we multiplied the NO$_x$ emissions by 0.2 in the LowNO$_x$ experiment to assess the impact of anthropogenic NO$_x$ on HOM concentration.*  *These experiments help quantify how uncertainties in chemical mechanisms affect HOM concentrations in global models.*

**Less Major Comments**

**Comment#1:** "In the LowNOx sensitivity experiment, total C10 concentrations decreased from 736 to 339 ng/m3 at the Centreville site (anthropogenically influenced) due to reduced NOx emissions, with C10-ON showing a more pronounced reduction (117 to 30 ng/m3) than C10-NON (619 to 310 ng/m3), consistent with the NO-dependent formation of C10-ON (Figs. 2 and 3)." There seems to be a bad disconnect here as the C10 How does the C10 concentrations decrease with reducing NOx? Is the NO involvement through RO formation taken into consideration.

**Response**: NO acts as a reactant and is involved in the formation of both $C_{10}$-ON and $C_{10}$-NON (as shown in Table 6 and Fig. 3). When NO concentrations decrease, the production rates of both species decrease, which in turn affects their concentrations. The main point here is that $C_{10}$-ON is more sensitive to changes in NO concentrations compared to $C_{10}$-NON. This is because $C_{10}$-ON formation is primarily driven by NO (Fig. 3), while $C_{10}$-NON can also be formed through other oxidants, such as $HO_2$ and $NO_3$.

To avoid misunderstanding, we have revised the sentence in lines 228 to 231 (the underlined content is newly added or modified):

> In the  LowNO sensitivity experiment, total $C_{10}$ concentrations decreased from 736 to 339 ng/m³ at the Centreville site (anthropogenically influenced) due to reduced  NO emissions, with $C_{10}$-ON showing a more pronounced reduction (117 to 30 ng/m³) than $C_{10}$-NON (619 to 310 ng/m³), consistent with the NO-dependent formation of $C_{10}$-ON ( Fig. 3 and Table 6).

Additionally, the name of the sensitivity experiment has been changed throughout the manuscript from LowNOx to LowNO.

[Figure]

**Figure 3**. Schematic diagram illustrating the oxidation of MT-HOM-RO₂ by HO₂, NO, or NO₃, leading to the formation of three types of gas-phase C₁₀ HOMs.

**Table 6.** $C_{10}$ *HOMs formation. Detailed descriptions of the intermediate species are provided in Table S12.*

| Index | Reactions | Reaction rate |
|---|---|---|
| 109 | $MT\text{-}HOM\text{-}RO_2 + HO_2 \rightarrow C_{10}\text{-}aNON + O_2$ | 1.5e-11 |
| 110 | $MT\text{-}HOM\text{-}RO_2 + NO \rightarrow \quad 0.8*NO_2 + 0.8*HO_2 + 0.4* C_{10}\text{-}bNON + 0.8*HYDRALD + 0.2* C_{10}\text{-}ON$ | 4.0e-12 |
| 111 | $MT\text{-}HOM\text{-}RO_2 + NO_3 \rightarrow HO_2 + NO_2 + 0.5* C_{10}\text{-}ON + HYDRALD$ | 1.2e-12 |

**Comment#2:** "The MT-RO2 formed by the oxidation of monoterpenes by $NO_3$ radicals are not considered in this study, as some studies report the branching ratio to be insignificant". This seems strange as several recent studies are finding NO3 oxidation far more important than has been previously thought. Perhaps you're confusing with the work of Kurtén et al., who explained that the one monoterpene that most people seem to concentrate do not have facile paths to HOM upon NO3 initiated oxidation (https://pubs.acs.org/doi/10.1021/acs.jpclett.7b01038), but it is unlikely that this result transfers to other monoterpene systems.

**Response**: Sorry for ignoring the role of $NO_3$-initiated oxidation pathways in our previous discussion. We have revised the manuscript in Lines 138 to 150 (the underlined content is newly added or modified):

*The MT-RO₂ formed by the oxidation of monoterpenes by NO₃ radicals is not considered in this study, as some studies report that the branching ratio remains highly uncertain to be insignificant (Zhao et al., 2021; Nah et al., 2016; Yan et al., 2016; Roldin et al., 2019), and the chemical process remains highly uncertain (Roldin et al., 2019).*

Additionally, the following discussion has been added in Section 5 (Conclusion):

*This study investigates the formation of HOMs from monoterpene oxidation in a global simulation, yet significant uncertainties remain in the representation of NO₃-initiated pathways. Recent studies suggest that NO₃-initiated HOM formation may be more important than previously thought, particularly under polluted nighttime conditions. Chamber experiments on α- and β-phellandrene oxidation by NO₃ have shown significant SOA and HOM production, with SOA yields reaching approximately 35% and 60%, respectively, accompanied by abundant HOM monomers and dimers (Harb et al., 2024). Furthermore, field observations from the southeastern United States indicate that NO₃ remains the dominant oxidant of monoterpenes at night, accounting for around 60% (observed) to 80% (modeled) of total monoterpene oxidation (Desai et al., 2024). These results highlight the potential importance of NO₃-initiated HOM formation in contributing to organic aerosol formation under polluted nighttime conditions. However, due to structural differences in monoterpenes,*

*such as ring strain and double-bond position, HOM yields vary widely among different species (Dam et al., 2022; Draper et al., 2024) and are highly sensitive to ambient NO$_x$ concentrations and humidity (Pasik et al., 2025; Li et al., 2022). The incomplete understanding of these mechanisms limits the accuracy of HOM predictions in models. Future research should combine field observations, laboratory constraints, and updated reaction schemes to reduce these uncertainties and improve global-scale modeling of nighttime organic aerosol formation.*

**Comment#3:** It is unclear how you are using the field data here. Do you estimate the individual C10 species concentrations from the experimental data? Did you obtain the raw data from the authors, or how did you come up with the signals? What sort of calibration factors were used?

**Response**: HOM measurements in this study were obtained using a high-resolution time-of-flight chemical ionization mass spectrometer (HRToF-CIMS), following the approach of Lopez-Hilfiker et al. (2014) where available. For the definition of HOMs, we selected compounds with molecular formulas containing 10 carbon atoms and at least 7 oxygen atoms.

The signals originate from FIGAERO-HRToF-CIMS thermal desorption measurements, in which sampled aerosol compounds are desorbed via a temperature ramp and subsequently detected as ion signals by the HRToF-CIMS. For quantification, a formic acid sensitivity was applied as the calibration factor, with a typical value of ~10 counts per second (cps) per ppt, to convert ion signals to concentrations (Lopez-Hilfiker et al., 2014).

The data used in this study were obtained from publicly available repositories:
SOAS campaign:
https://csl.noaa.gov/groups/csl7/measurements/2013senex/Ground/DataDownload/

BAECC campaign: https://www.arm.gov/research/campaigns/amf2014baecc

SMEAR II dataset: https://smear.avaa.csc.fi/download

To improve readability and clarity for the reader, we have expanded Section 2.3 (Observations) to include additional details regarding the field campaigns. The updated Section 2.3 is provided below:

*Data from two campaigns were used for comparison: the Southern Oxidant and Aerosol Study (SOAS) in the southeastern USA, and the Biogenic Aerosols – Effects on Clouds and Climate (BAECC) in Hyytiälä, Finland (Carlton et al., 2018; Martin et al., 2016; Petäjä et al., 2016) (Table 7). HOM measurements were obtained using high-resolution time-of-flight chemical ionization mass spectrometer (HRToF-CIMS) when available (Lopez-Hilfiker et al., 2014). For HOM measurements, molecular formulas of compounds containing 10 carbon atoms and at least 7 oxygen atoms were selected as HOMs. The compounds with one nitrate and without*

*nitrate were compared to the simulated $C_{10}$-aNON, $C_{10}$-bNON, and $C_{10}$-ON, respectively. In addition to HOMs, related species such as NO, $O_3$, monoterpenes, and isoprene were also compared when the data was available (Figs. S1 and S2). The primary HOM species identified in the SENEX (Southeast Nexus) and BAECC campaigns (Tables S15 and S16).*

**Table 7.** *Field campaigns used in this paper*

| Campaigns | Dates | Locations |
|---|---|---|
| SOAS (Warneke et al., 2016) | 2013.06.01–07.15 | Centreville, Alabama, US (32.93°N, 87.13°W) |
| BAECC (Petäjä et al., 2016) | 2014.04.11–06.03 | Station for Measuring Ecosystem Atmosphere Relations (SMEAR II), Hyytiälä, Finland. (61.85°N,24.28°E) |

**Table S15.** *Molecular formulas of top 5 contributing HOM-ON and HOM-NON species (gas- and particle-phase) at Centreville, Alabama*

| HOM-ON | | HOM-NON | |
|---|---|---|---|
| **Gas-phase** | **Particle-phase** | **Gas-phase** | **Particle-phase** |
| C10H15O7N1 | C10H15O7N1 | C10H14O7 | C10H14O7 |
| C10H17O7N1 | C10H15O8N1 | C10H12O7 | C10H12O7 |
| C10H15O8N1 | C10H17O7N1 | C10H22O8 | C10H16O7 |
| C10H17O8N1 | C10H17O8N1 | C10H22O7 | C10H22O8 |
| C10H13O8N1 | C10H15O9N1 | C10H16O7 | C10H22O7 |

**Table S16.** *Molecular formulas of top 5 contributing HOM-ON and HOM-NON species (gas- and particle-phase) at Hyytiälä, Finland*

| HOM-ON | | HOM-NON | |
|---|---|---|---|
| **Gas-phase** | **Particle-phase** | **Gas-phase** | **Particle-phase** |
| C10H15O7N1 | C10H15O8N1 | C10H12O11 | C10H14O7 |
| C10H15O8N1 | C10H15O7N1 | C10H14O8 | C10H22O9 |
| C10H17O7N1 | C10H17O7N1 | C10H16O8 | C10H22O7 |
| C10H13O7N1 | C10H17O8N1 | C10H14O7 | C10H22O8 |

| *C10H17O8N1* | *C10H15O9N1* | *C10H22O7* | *C10H16O7* |

**Comment#4:** Related, I would like the authors to comment on the assumed volatility classes. How sure it is that the compounds assumed ELVOC, are actually ELVOC? This seems critical for understanding the work.

**Response**: We thank the reviewer for raising this critical point. The volatility classification, particularly the assignment of certain oxidation products as ELVOCs, indeed has important implications for interpreting their roles in SOA formation and growth processes. In our study, the assignment of compounds to volatility bins (ELVOC, LVOC, SVOC) was based on molecular formulae (see equation (2)). Although our volatility estimates are consistent with current literature and methodology (Schervish and Donahue, 2020), some studies (Stolzenburg et al., 2018; Ye et al., 2018; Schervish and Donahue, 2020) have presented molecular formulas and concentrations of accretion products in different volatility bins. However, the explicit chemical kinetics of the related reactions (such as intermediate products and their yields) are not provided. Therefore, we are unable to accurately represent all of the final products mentioned in these studies within the CAM6-Chem. We acknowledge that this simplification may lead to uncertainty in the volatility of HOMs, and this should be thoroughly discussed. We have added the following discussion at the end of Section 5 (Conclusion):

*There may be some overestimations of C$_{15}$ and C$_{20}$ if all the accretion products are assumed to be ELVOC or ULVOC. In the updated model, C$_{15}$H$_{18}$O$_9$ (C$_{15}$, extremely low volatility) and C$_{20}$H$_{32}$O$_8$ (C$_{20}$, ultra-low volatility) are used as simplified representatives for all C$_{15}$ and C$_{20}$ dimers. While additional low-volatility dimer species have been detected in chamber experiments (Stolzenburg et al., 2018; Ye et al., 2019; Schervish and Donahue, 2020), these studies did not provide explicit chemical kinetics for the reactions (i.e., intermediate products and their yields), which limits the ability to consider more precise volatility estimates for the accretion products in the model. This uncertainty may influence the contribution of both the accretion products and HOMs-SOA to the overall SOA.*

$$log_{10} C^*(300\ K) =$$

$$(25 - n_C) \times b_C - (n_o - 3n_N) \times b_O - n_N \times b_N - 2\left[\frac{(n_o - 3n_N) \times n_C}{n_C + n_o - 3n_N}\right] \times b_{CO} \qquad (2)$$

*where $n_C$, $n_o$, and $n_N$ are the number of carbon, oxygen, and nitrogen atoms; $b_C$ = 0.475; $b_O$ = 0.2; $b_N$ = 2.5; $b_{CO}$ = 0.9.*

**Comment#5:** Finally, If you want to claim "Addressing these gaps requires coordinated laboratory measurements and targeted ambient observations to disentangle competing chemical processes."

Then could you please specifically explain what type of ambient measurement would help in this task. How do you envision one could speciate the corresponding chemicals from the ambient gas-phase.

**Response**: Regarding specific ambient measurements, we believe that advanced techniques such as high-resolution time-of-flight chemical ionization mass spectrometry (HR-ToF-CIMS) combined with targeted chemical ionization methods will be particularly critical. These approaches enable real-time, highly sensitive detection of trace gases and their oxidation products in the gas phase, thereby facilitating the speciation of relevant compounds in the ambient atmosphere.

However, our current understanding of how to practically implement such measurements in the field, and how to effectively transfer laboratory-derived parameters to ambient observations for optimization, remains limited. Therefore, we can only propose directions from the perspective of model development: on the one hand, field observations are needed to characterize the molecular composition and volatility of accretion products; on the other hand, chamber studies are required to constrain MT-bRO$_2$ branching ratios. In the future, it may be possible to first obtain key parameters in the laboratory and then progressively refine and apply them under ambient conditions. To emphasize these priorities, we have included a directional statement in the final paragraph of the conclusion:

> *To address persistent gaps between model predictions and observations, field campaigns targeting accretion product speciation and chamber studies that constrain MT-bRO$_2$ branching ratios are needed.*

**Picked up**

There's an error: "As the number of oxygen atoms in the functional group increases, the volatility of the organics gradually decreases."

**Response**: Thank you for your valuable feedback. Considering that this statement could lead to some confusion, we have removed the sentence from the original text.

Please reword (page 3): "but the models still lack fully understand the uncertainties."
**Response**: We have revised the sentence "but the models still lack fully understanding the uncertainties" to "but the models still lack a full understanding of the uncertainties."

"nitrogen dioxide (NOx)" – nitrogen oxides
NOx is not either of the NO and NO2, it is both. Please clarify the staments claiming NOx can help autoxidation.

**Response**: Thank you for pointing out the error. Yes, we will correct this mistake throughout the text, replacing most instances of "NOx" with "NO".

**Reference**

Baker, Y., Kang, S., Wang, H., Wu, R., Xu, J., Zanders, A., He, Q., Hohaus, T., Ziehm, T., Geretti, V., Bannan, T. J., O'Meara, S. P., Voliotis, A., Hallquist, M., McFiggans, G., Zorn, S. R., Wahner, A., and Mentel, T. F.: Impact of HO2/RO2 ratio on highly oxygenated α-pinene photooxidation products and secondary organic aerosol formation potential, Atmos. Chem. Phys., 24, 4789–4807, https://doi.org/10.5194/acp-24-4789-2024, 2024.

Berndt, T., Richters, S., Jokinen, T., Hyttinen, N., Kurten, T., Otkjaer, R. V., Kjaergaard, H. G., Stratmann, F., Herrmann, H., Sipila, M., Kulmala, M., and Ehn, M.: Hydroxyl radical-induced formation of highly oxidized organic compounds, Nat Commun, 7, 13677, 10.1038/ncomms13677, 2016.

Bianchi, F., Kurten, T., Riva, M., Mohr, C., Rissanen, M. P., Roldin, P., Berndt, T., Crounse, J. D., Wennberg, P. O., Mentel, T. F., Wildt, J., Junninen, H., Jokinen, T., Kulmala, M., Worsnop, D. R., Thornton, J. A., Donahue, N., Kjaergaard, H. G., and Ehn, M.: Highly Oxygenated Organic Molecules (HOM) from Gas-Phase Autoxidation Involving Peroxy Radicals: A Key Contributor to Atmospheric Aerosol, Chem Rev, 119, 3472-3509, 10.1021/acs.chemrev.8b00395, 2019.

Carlton, A. G., de Gouw, J., Jimenez, J. L., Ambrose, J. L., Attwood, A. R., Brown, S., Baker, K. R., Brock, C., Cohen, R. C., Edgerton, S., Farkas, C. M., Farmer, D., Goldstein, A. H., Gratz, L., Guenther, A., Hunt, S., Jaeglé, L., Jaffe, D. A., Mak, J., McClure, C., Nenes, A., Nguyen, T. K., Pierce, J. R., de Sa, S., Selin, N. E., Shah, V., Shaw, S., Shepson, P. B., Song, S., Stutz, J., Surratt, J. D., Turpin, B. J., Warneke, C., Washenfelder, R. A., Wennberg, P. O., and Zhou, X.: Synthesis of the Southeast Atmosphere Studies: Investigating Fundamental Atmospheric Chemistry Questions, B. Am. Meteorol. Soc., 99, 547–567, https://doi.org/10.1175/BAMS-D-16-0048.1, 2018

Dam, M., Draper, D. C., Marsavin, A., Fry, J. L., and Smith, J. N.: Observations of gas-phase products from the nitrate-radical-initiated oxidation of four monoterpenes, Atmos. Chem. Phys., 22, 9017–9031, https://doi.org/10.5194/acp-22-9017-2022, 2022.

Desai, N. S., Moore, A. C., Mouat, A. P., Liang, Y., Xu, T., Takeuchi, M., Pye, H. O. T., Murphy, B., Bash, J., Pollack, I. B., Peischl, J., Ng, N. L., and Kaiser, J.: Impact of Heatwaves and Declining NOx on Nocturnal Monoterpene Oxidation in the Urban Southeastern United States, Journal of Geophysical Research: Atmospheres, 129, e2024JD041482, https://doi.org/10.1029/2024JD041482, 2024.

Draper, D., Almeida, T. G., Iyer, S., Smith, J. N., Kurtén, T., and Myllys, N.: Unpacking the diversity of monoterpene oxidation pathways via nitrooxy–alkyl radical ring-opening reactions and nitrooxy–alkoxyl radical bond scissions, J. Aerosol Sci., 179, 106379, https://doi.org/10.1016/j.jaerosci.2024.106379, 2024.
Ehn, M., Thornton, J. A., Kleist, E., Sipila, M., Junninen, H., Pullinen, I., Springer, M., Rubach, F., Tillmann, R., Lee, B., Lopez-Hilfiker, F., Andres, S., Acir, I. H., Rissanen, M., Jokinen, T., Schobesberger, S., Kangasluoma, J., Kontkanen, J., Nieminen, T., Kurten, T., Nielsen, L. B., Jorgensen, S., Kjaergaard, H. G., Canagaratna, M., Maso, M. D., Berndt, T., Petaja, T., Wahner, A., Kerminen, V. M., Kulmala, M., Worsnop, D. R., Wildt, J., and Mentel, T. F.: A large source of low-volatility secondary organic aerosol, Nature, 506, 476-479, 10.1038/nature13032, 2014.

Harb, S., Cirtog, M., Alage, S., Cantrell, C., Cazaunau, M., Michoud, V., Pangui, E., Bergé, A., Giorio, C., Battaglia, F., and Picquet-Varrault, B.: HOMs and SOA formation from the oxidation of α- and β-phellandrenes by NO3 radicals, EGUsphere [preprint], https://doi.org/10.5194/egusphere-2024-3419, 2024.

Jokinen, T., Berndt, T., Makkonen, R., Kerminen, V. M., Junninen, H., Paasonen, P., Stratmann, F., Herrmann, H., Guenther, A. B., Worsnop, D. R., Kulmala, M., Ehn, M., and Sipila, M.: Production of extremely low volatile organic compounds from biogenic emissions: Measured yields and atmospheric implications, Proc Natl Acad Sci U S A, 112, 7123-7128, 10.1073/pnas.1423977112, 2015.
Kurten, T., Rissanen, M. P., Mackeprang, K., Thornton, J. A., Hyttinen, N., Jorgensen, S., Ehn, M., and Kjaergaard, H. G.: Computational Study of Hydrogen Shifts and Ring-Opening Mechanisms in alpha-Pinene Ozonolysis Products, J Phys Chem A, 119, 11366-11375, 10.1021/acs.jpca.5b08948, 2015.

Lee, B. H., Iyer, S., Kurtén, T., Varelas, J. G., Luo, J., Thomson, R. J., and Thornton, J. A.: Ring-opening yields and auto-oxidation rates of the resulting peroxy radicals from OH-oxidation of α-pinene and β-pinene, Environmental Science: Atmospheres, 3, 399-407, 10.1039/d2ea00133k, 2023.

Lopez-Hilfiker, F. D., Mohr, C., Ehn, M., Rubach, F., Kleist, E., Wildt, J., Mentel, Th. F., Lutz, A., Hallquist, M., Worsnop, D., and Thornton, J. A.: A novel method for online analysis of gas and particle composition: description and evaluation of a Filter Inlet for Gases and AEROsols (FIGAERO), Atmos. Meas. Tech., 7, 983–1001, https://doi.org/10.5194/amt-7-983-2014, 2014.

Li, D.; Huang, W.; Wang, D.; Wang, M.; Thornton, J. A.; Caudillo, L.; Rorup, B.; Marten, R.; Scholz, W.; Finkenzeller, H.; Marie, G.; Baltensperger, U.; Bell, D. M.; Brasseur, Z.; Curtius, J.; Dada, L.; Duplissy, J.; Gong, X.; Hansel, A.; He, X. C.; Hofbauer, V.; Junninen, H.; Krechmer, J. E.; Kurten, A.; Lamkaddam, H.; Lehtipalo, K.; Lopez, B.; Ma, Y.; Mahfouz, N. G. A.; Manninen, H. E.; Mentler, B.; Perrier, S.; Petaja, T.; Pfeifer, J.; Philippov, M.; Schervish, M.; Schobesberger, S.; Shen, J.; Surdu, M.; Tomaz, S.; Volkamer, R.; Wang, X.; Weber, S. K.; Welti, A.; Worsnop, D. R.; Wu, Y.; Yan, C.; Zauner-Wieczorek, M.; Kulmala, M.; Kirkby, J.; Donahue, N. M.; George, C.; El-Haddad, I.; Bianchi, F.; Riva, M. Nitrate Radicals Suppress Biogenic New Particle Formation from Monoterpene Oxidation. Environ. Sci. Technol. 2024, 58 (3), 1601−1614.

Molteni, U., Simon, M., Heinritzi, M., Hoyle, C. R., Bernham- mer, A.-K., Bianchi, F., Breitenlechner, M., Brilke, S., Dias, A., Duplissy, J., Frege, C., Gordon, H., Heyn, C., Jokinen, T., Kürten, A., Lehtipalo, K., Makhmutov, V., Petäjä, T., Pieber, S. M., Praplan, A. P., Schobesberger, S., Steiner, G., Stozhkov, Y., Tomé, A., Tröstl, J., Wagner, A. C., Wagner, R., Williamson, C., Yan, C., Baltensperger, U., Curtius, J., Donahue, N. M., Hansel, A., Kirkby, J., Kulmala, M., Worsnop, D. R., and Dom- men, J.: Formation of highly-oxygenated organic molecules from α-pinene ozonolysis: chemical characteristics, mechanism and kinetic model development, Earth Space Chem., 3, 873–883, https://doi.org/10.1021/acsearthspacechem.9b00035, 2019.

Moller, K. H., Otkjaer, R. V., Chen, J., and Kjaergaard, H. G.: Double Bonds Are Key to Fast Unimolecular Reactivity in First-Generation Monoterpene Hydroxy Peroxy Radicals, J Phys Chem A, 124, 2885-2896, 10.1021/acs.jpca.0c01079, 2020.

Murphy, S. E., Crounse, J. D., Møller, K. H., Rezgui, S. P., Hafeman, N. J., Park, J., Kjaergaard, H. G., Stoltz, B. M., and Wennberg, P. O.: Accretion product formation in the selfreaction of ethene-derived hydroxy peroxy radicals, Environ. Sci.-Atmos., 3, 882–893, https://doi.org/10.1039/D3EA00020F, 2023.

Nah, T., Sanchez, J., Boyd, C. M., and Ng, N. L.: Photochemical Aging of α-pinene and β-pinene Secondary Organic Aerosol formed from Nitrate Radical Oxidation, Environ. Sci. Technol., 50, 222-231, 10.1021/acs.est.5b04594, 2016.

Pasik, D., Golin Almeida, T., Ahongshangbam, E., Iyer, S., and Myllys, N.: Monoterpene oxidation pathways initiated by acyl peroxy radical addition, Atmos. Chem. Phys., 25, 4313–4331, https://doi.org/10.5194/acp-25-4313-2025, 2025.

Pye, H. O. T., D'Ambro, E. L., Lee, B. H., Schobesberger, S., Takeuchi, M., Zhao, Y., Lopez-Hilfiker, F., Liu, J., Shilling, J. E., Xing, J., Mathur, R., Middlebrook, A. M., Liao, J., Welti, A., Graus, M., Warneke, C., de Gouw, J. A., Holloway, J. S., Ryerson, T. B., Pollack, I. B., and Thornton, J. A.: Anthropogenic enhancements to production of highly oxygenated molecules from autoxidation, Proc Natl Acad Sci U S A, 116, 6641-6646, 10.1073/pnas.1810774116, 2019.

Richters, S., Herrmann, H., and Berndt, T.: Highly Oxidized RO2 Radicals and Consecutive Products from the Ozonolysis of Three Sesquiterpenes, Environ Sci Technol, 50, 2354-2362, 10.1021/acs.est.5b05321, 2016.

Rissanen, M. P., Kurtén, T., Sipilä, M., Thornton, J. A., Kausiala, O., Garmash, O., Kjaergaard, H. G., Petäjä, T., Worsnop, D. R., Ehn, M., and Kulmala, M.: Effects of Chemical Complexity on the Autoxidation Mechanisms of Endocyclic Alkene Ozonolysis Products: From Methylcyclohexenes toward Understanding á-Pinene, J. Phys. Chem. A, 119, 4633–4650, https://doi.org/10.1021/jp510966g, 2015.

Roldin, P., Ehn, M., Kurten, T., Olenius, T., Rissanen, M. P., Sarnela, N., Elm, J., Rantala, P., Hao, L., Hyttinen, N., Heikkinen, L., Worsnop, D. R., Pichelstorfer, L., Xavier, C., Clusius, P., Ostrom, E., Petaja, T., Kulmala, M., Vehkamaki, H., Virtanen, A., Riipinen, I., and Boy, M.: The role of highly oxygenated organic molecules in the Boreal aerosol-cloud-climate system, Nat Commun, 10, 4370, 10.1038/s41467-019-12338-8, 2019.

Rolletter, M., Kaminski, M., Acir, I. H., Bohn, B., Dorn, H. P., Li, X., Lutz, A., Nehr, S., Rohrer, F., Tillmann, R., Wegener, R., Hofzumahaus, A., Kiendler-Scharr, A., Wahner, A., and Fuchs, H.: Investigation of the α-pinene photooxidation by OH in the atmospheric simulation chamber SAPHIR, Atmos. Chem. Phys., 19, 11635-11649, 10.5194/acp-19-11635-2019, 2019.

Saunders, S. M., Jenkin, M. E., Derwent, R. G., and Pilling, M. J.: Protocol for the development of the Master Chemical Mechanism, MCM v3 (Part A): tropospheric degradation of non-aromatic volatile organic compounds, Atmos. Chem. Phys., 3, 161-180, 10.5194/acp-3-161-2003, 2003.

Schervish, M. and Donahue, N. M.: Peroxy radical chemistry and the volatility basis set, Atmos. Chem. Phys., 20, 1183–1199, https://doi.org/10.5194/acp-20-1183-2020, 2020.

Stolzenburg, D., Fischer, L., Vogel, A. L., Heinritzi, M., Schervish, M., Simon, M., Wagner, A. C., Dada, L., Ahonen, L. R., Amorim, A., Baccarini, A., Bauer, P. S., Baumgartner, B., Bergen, A., Bianchi, F., Breitenlechner, M., Brilke, S., Buenrostro Mazon, S., Chen, D., Dias, A., Draper, D. C., Duplissy, J., El Haddad, I., Finkenzeller, H., Frege, C., Fuchs, C., Garmash, O., Gordon, H., He, X., Helm, J., Hofbauer, V., Hoyle, C. R., Kim, C., Kirkby, J., Kontkanen, J., Kürten, A., Lampilahti, J., Lawler, M., Lehtipalo, K., Leiminger, M., Mai, H., Mathot, S., Mentler, B., Molteni, U., Nie, W., Nieminen, T., Nowak, J. B., Ojdanic, A., Onnela, A., Passananti, M., Petäjä, T., Quéléver, L. L. J., Rissanen, M. P., Sarnela, N., Schallhart, S., Tauber, C., Tomé, A., Wagner, R., Wang, M., Weitz, L., Wimmer, D., Xiao, M., Yan, C., Ye, P., Zha, Q., Baltensperger, U., Curtius, J., Dommen, J., Flagan, R. C., Kulmala, M., Smith, J. N., Worsnop, D. R., Hansel, A., Donahue, N. M., and Winkler, P. M.: Rapid growth of organic aerosol nanoparticles over a wide tropospheric temperature range, P. Natl. Acad. Sci. USA, 115, 9122–9127, https://doi.org/10.1073/pnas.1807604115, 2018.

Weber, J., Archer-Nicholls, S., Griffiths, P., Berndt, T., Jenkin, M., Gordon, H., Knote, C., and Archibald, A. T.: CRI-HOM: A novel chemical mechanism for simulating highly oxygenated organic molecules (HOMs) in global chemistry–aerosol–climate models, Atmos. Chem. Phys., 20, 10889-10910, 10.5194/acp-20-10889-2020, 2020.

Xu, L., Moller, K. H., Crounse, J. D., Otkjaer, R. V., Kjaergaard, H. G., and Wennberg, P. O.: Unimolecular Reactions of Peroxy Radicals Formed in the Oxidation of alpha-Pinene and beta-Pinene by Hydroxyl Radicals, J Phys Chem A, 123, 1661-1674, 10.1021/acs.jpca.8b11726, 2019.
Yan, C., Nie, W., Äijälä, M., Rissanen, M. P., Canagaratna, M. R., Massoli, P., Junninen, H., Jokinen, T., Sarnela, N., Häme, S. A. K., Schobesberger, S., Canonaco, F., Yao, L., Prévôt, A. S. H., Petäjä, T., Kulmala, M., Sipilä, M., Worsnop, D. R., and Ehn, M.: Source characterization of highly oxidized multifunctional compounds in a boreal forest environment using positive matrix factorization, Atmos. Chem. Phys., 16, 12715-12731, 10.5194/acp-16-12715-2016, 2016.

Schervish, M. and Donahue, N. M.: Peroxy radical chemistry and the volatility basis set, Atmospheric Chemistry and Physics, 20, 1183-1199, 10.5194/acp-20-1183-2020, 2020.

Stolzenburg, D., Fischer, L., Vogel, A. L., Heinritzi, M., Schervish, M., Simon, M., Wagner, A. C., Dada, L., Ahonen, L. R., Amorim, A., Baccarini, A., Bauer, P. S., Baumgartner, B., Bergen, A., Bianchi, F., Breitenlechner, M., Brilke, S., Buenrostro Mazon, S., Chen, D., Dias, A., Draper, D. C., Duplissy, J., El Haddad, I., Finkenzeller, H., Frege, C., Fuchs, C., Garmash, O., Gordon, H., He, X., Helm, J., Hofbauer, V., Hoyle, C. R., Kim, C., Kirkby, J., Kontkanen, J., Kürten, A., Lampilahti, J., Lawler, M., Lehtipalo, K., Leiminger, M., Mai, H., Mathot, S., Mentler, B., Molteni, U., Nie, W., Nieminen, T., Nowak, J. B., Ojdanic, A., Onnela, A., Passananti, M., Petäjä, T., Quéléver, L. L. J., Rissanen, M. P., Sarnela, N., Schallhart, S., Tauber, C., Tomé, A., Wagner, R., Wang, M., Weitz, L., Wimmer, D., Xiao, M., Yan, C., Ye, P., Zha, Q., Baltensperger, U., Curtius, J., Dommen, J., Flagan, R. C., Kulmala, M., Smith, J. N., Worsnop, D. R., Hansel, A., Donahue, N. M., and Winkler, P. M.: Rapid growth of organic aerosol nanoparticles over a wide tropospheric temperature range, P. Natl. Acad. Sci. USA, 115, 9122-9127, 10.1073/pnas.1807604115, 2018.

Xu, R. C., Thornton, J. A., Lee, B., Zhang, Y. X., Jaegle, L., Lopez-Hilfiker, F. D., Rantala, P., and Petaja, T.: Global simulations of monoterpene-derived peroxy radical fates and the distributions of highly

oxygenated organic molecules (HOMs) and accretion products, Atmos. Chem. Phys., 22, 5477-5494, 10.5194/acp-22-5477-2022, 2022.

Ye, Q., Wang, M., Hofbauer, V., Stolzenburg, D., Chen, D., Schervish, M., Vogel, A., Mauldin, R. L., Baalbaki, R., Brilke, S., Dada, L., Dias, A., Duplissy, J., El Haddad, I., Finkenzeller, H., Fischer, L., He, X., Kim, C., Kürten, A., Lamkaddam, H., Lee, C. P., Lehtipalo, K., Leiminger, M., Manninen, H. E., Marten, R., Mentler, B., Partoll, E., Petäjä, T., Rissanen, M., Schobesberger, S., Schuchmann, S., Simon, M., Tham, Y. J., Vazquez-Pufleau, M., Wagner, A. C., Wang, Y., Wu, Y., Xiao, M., Baltensperger, U., Curtius, J., Flagan, R., Kirkby, J., Kulmala, M., Volkamer, R., Winkler, P. M., Worsnop, D., and Donahue, N. M.: Molecular Composition and Volatility of Nucleated Particles from á-Pinene Oxidation between −50 °C and +25 °C, Environ. Sci. Technol., 53, 12357–12365, https://doi.org/10.1021/acs.est.9b03265, 2019.

Zawadowicz, M. A., Lee, B. H., Shrivastava, M., Zelenyuk, A., Zaveri, R. A., Flynn, C., Thornton, J. A., and Shilling, J. E.: Photolysis Controls Atmospheric Budgets of Biogenic Secondary Organic Aerosol, Environ. Sci. Technol., 54, 3861-3870, 10.1021/acs.est.9b07051, 2020.

Zhao, Y., Thornton, J. A., and Pye, H. O. T.: Quantitative constraints on autoxidation and dimer formation from direct probing of monoterpene-derived peroxy radical chemistry, Proc Natl Acad Sci U S A, 115, 12142-12147, 10.1073/pnas.1812147115, 2018.

Zhao, Y., Saleh, R., Saliba, G., Presto, A. A., Gordon, T. D., Drozd, G. T., Goldstein, A. H., Donahue, N. M., and Robinson, A. L.: Reducing secondary organic aerosol formation from gaso- line vehicle exhaust, P. Natl. Acad. Sci. USA, 114, 6984–6989, https://doi.org/10.1073/pnas.1620911114,, 2017.

---

## Author Comment (AC2)

We are very grateful to the evaluations from the reviewers, which have allowed us to clarify and improve the manuscript. Below we addressed the reviewer comments, with the reviewer comments in black and our response in **blue**.

**Reply for the referee comment#2**

**General Comments:** This manuscript presents a comprehensive modeling study of HOMs (Highly Oxygenated Organic Molecules) derived from monoterpenes using the CAM6-Chem global climate model. The authors implement a semi-explicit HOM chemical mechanism, extend the volatility basis set (VBS), and conduct a suite of sensitivity experiments to explore the impacts of branching ratios, autoxidation rates, and NOx levels on HOMs-SOA formation. The study is timely and addresses a significant gap in the representation of HOMs in global models. The methodology is sound, and the model development is well-documented. However, a major weakness of the manuscript lies in the limited and underdeveloped discussion of the results, particularly those from the global model simulations. The current treatment of global HOMs-SOA distributions lacks the depth and detail expected for a study of this scope and importance. This significantly reduces the scientific value and interpretability of the findings. For this reason, I recommend major revisions, with particular emphasis on expanding and deepening the analysis of the global model results. Additionally, improvements are needed in the clarity of some sections, including the presentation of the chemical mechanism.

**Response**: Thank you very much for the reviewer's thoughtful review and valuable comments. In response to the suggestions, we have strengthened the analysis and discussion of the global simulation results, and improved the clarity of the chemical mechanism presentation. All relevant issues raised have been carefully addressed in the revised manuscript. Detailed responses to each comment are provided below.

**Major comments**

**1. Expansion of the discussion on global model results**

The current discussion of global HOMs-SOA distributions (Section 3.3) is relatively brief and lacks depth. I strongly recommend expanding this analysis and relocating it to a new Section 4 to allow for a more comprehensive and focused discussion, including:

- A more detailed regional analysis beyond SEUS, SECN, and Amazon. For example, Europe, South Asia, and Africa are not discussed despite being relevant for OMs and/or biogenic and anthropogenic interactions.
- A more thorough treatment of seasonality, including regional seasonal cycles and their drivers (e.g., oxidant levels, emissions).
- A spatially resolved analysis of how HOMs-SOA distributions change across the nine sensitivity experiments. This could include difference maps or regional statistics showing the impact of branching ratios and autoxidation rates.
- Consideration of vertical distributions of HOMs-SOA, especially given their relevance for new particle formation and cloud interactions.

**Response**: We sincerely thank the reviewer for the insightful and constructive comments. We have substantially expanded the analysis of global HOMs-SOA distributions and moved this discussion to a new Section 4, as suggested. The updated content of Section 4 is as follows:

*4 Spatial and temporal distribution of HOMs-SOA*

*4.1 Spatial and distribution of HOMs-SOA*

*Globally, the annual mean concentration of HOMs-SOA is 0.0556 µg m⁻³, accounting for 13.3 % of total SOA and 37.3 % of MTSOA (Fig. 12). Across the sensitivity experiments, the contribution of HOMs-SOA to total SOA varies between 6 % and 15 % (Fig. 9). The HighYield (15 %) and LowYield (6 %) experiments encompass the uncertainty range reported in Xu et al. (2022) and highlight the dominant influence of the MT-bRO₂ branching ratio on HOMs-SOA formation. The effect of autoxidation rates is secondary, as indicated by the Fast (14 %) and Slow (10 %) experiments (Fig. 12), where the differences are moderate and consistent with the rate-dependent parameterizations proposed by Weber et al. (2021). In contrast, the influence of NO levels is negligible, with the LowNO (13 %) and no_HMB_NO (11 %) experiments indicating that anthropogenic NO emissions exert little impact on the global HOMs-SOA concentration. Similarly, the contribution of HOMs-SOA to MTSOA ranges from 19 % to 41 % across the sensitivity experiments, with the highest fraction in HighYield and the lowest in LowYield, further highlighting the critical role of the MT-bRO₂ branching ratio in determining the HOMs-SOA contribution.*

[Figure]

**Figure 9.** *The global averaged value of 2013 annual mean surface HOMs-SOA (unit: µg/m3), the contribution of HOMs-SOA to the total SOA and MTSOA (unit: %) using different sensitivity tests (Table 8). The specific value of different tests is shown in Table S14.*

*The contribution of HOMs-SOA to total SOA and MTSOA shows pronounced spatial variability globally (Fig. 10). In the Control experiment, HOMs-SOA concentrations are highest in tropical and parts of mid-latitude regions, particularly over the Amazon rainforest, central Africa, the southeastern United States, and Australia, where warm and humid conditions coincide with high biogenic emissions (Fig. 10a). In these regions, HOMs-SOA typically accounts for more than 10 % of total SOA, with some areas exceeding 40 % (Fig. 10b), and contributes more than 20 % to MTSOA (Fig. 10c). Also, in regions strongly influenced by anthropogenic emissions, such as Southeast Asia and Europe, the contribution of HOMs-SOA to MTSOA remains substantial, consistent with the role of anthropogenic NO in HOM formation (Fig. 3).*

*Across the sensitivity experiments, the spatial patterns of HOMs-SOA contributions to total SOA and MTSOA are generally consistent with those in the Control experiment (Fig. S7). In the HighYield experiment, the contribution of HOMs-SOA to MTSOA exceeds 30 % over most tropical regions with high biogenic emissions. In contrast, in the LowYield experiment, the contribution to MTSOA drops markedly across most regions, falling below 15 %. These results indicate that uncertainties in the MT-bRO$_2$ branching ratio not only affect the global mean but also amplify the contrast between biogenically dominated regions and other areas. Changes in the Fast and Slow experiments are more moderate, occurring mainly over high-emission regions and their downwind areas, where Fast increases the HOMs-SOA contribution to MTSOA by ~5%, while Slow yields comparable decreases. By comparison, the NO-related experiments (LowNO and no_HMB_NO) show limited changes in most regions, with slight decreases (< 2%) in both MTSOA and total SOA contributions over high-NO emission regions such as East Asia and Europe. These results further support that, although NO can alter HOM composition under polluted conditions, its influence on the global burden and spatial distribution of HOMs-SOA is minimal.*

[Figure]

***Figure 10.*** *2013 annual averaged surface (a) HOMs-SOA (unit: µg/m3), (b) the contribution of HOMs-SOA to the MTSOA and (c) the contribution of HOMs-SOA to the total SOA (unit: %) in Control experiment. The global averaged value is shown in upper right corner of each figure. Proportions are only shown in regions where MTSOA or total SOA is greater than 10% of the global average.*

*Vertically, HOMs-SOA is primarily concentrated in the near-surface and lower troposphere (below 800 hPa), reflecting its close link to surface biogenic emissions (Fig. 11). There are significant regional differences in the contribution of HOMs-SOA to total SOA between the Northern and Southern Hemispheres. In most regions of the Northern Hemisphere, anthropogenic emissions dominate, leading to a low contribution from biogenic HOMs-SOA (<10%). In contrast, the contribution increases markedly in the Southern Hemisphere (>14%). This difference is largely driven by the high emissions of monoterpenes and isoprene in tropical regions, such as the Amazon and central Africa (Fig. S4), which promote the substantial formation of HOMs-SOA, especially C15 and C20 (Fig. 12). These compounds are then transported to higher altitudes through deep convection. As a result, HOMs-SOA remains at high concentrations in the 400–200 hPa range in the tropics, contributing more than 20% to SOA at these altitudes (Fig. 11). Additionally, gaseous HOMs may also be transported to higher layers, where they significantly enhance new particle formation, influencing cloud condensation nuclei (CCN) concentrations (Shao et al., 2024; Zhao et al., 2024), and further affecting cloud properties and radiative effects (Shao et al., 2025).*

[Figure]

***Figure 11.*** *Vertical distribution of 2013 annual averaged (a) HOMs-SOA concentration (µg/m³) and (b) proportion of HOMs-SOA to total SOA (%) in the Control experiment. The global average value is shown in the upper right corner of each panel. Proportions are only shown in regions where total SOA is greater than 10% of the global average.*

[Figure]

*Figure S7.* *Global distribution of HOMs-SOA concentrations (μg/m³) and their contributions to SOA (middle column) and MTSOA (right column) across different sensitivity experiments. The global average value is displayed in the upper right corner of each panel. Proportions are only shown in regions where MTSOA or total SOA is greater than 10% of the global average.*

**4.2 Temporal variation of HOMs-SOA**

*On a global scale, the seasonal variation of HOMs-SOA are primarily determined by the intensity of biogenic emissions (monoterpenes and isoprene), with a more significant effect in the Northern Hemisphere summer (Fig. 13). The levels of oxidants (OH and $O_3$), however, play a greater role in modulating the relative contribution of different HOM species (Figs. 12 and 14). In high NO regions, such as the United States (US), Southeast Asian continental regions (SECN), and Europe (EUR), the concentrations of OH and $O_3$ facilitate the formation of MT-HOM-$RO_2$. Combined with higher NO concentrations, this promotes the generation of $C_{10}$-ON (Fig. 3). In contrast, in low NO tropical regions, such as the Amazon and central Africa (CAF), the proportion of low-volatility dimers ($C_{15}$ and $C_{20}$) significantly increases.*

*In tropical regions, the absolute concentration of HOMs-SOA is significantly higher than in mid-latitudes, with a greater proportion of dimers ($C_{15}$ and $C_{20}$) (Fig. 12). The annual mean concentration in the AMZ is 0.90 μg/m³, peaking during the dry season (August–October) at approximately 1.4 μg/m³. During this period, monoterpene (~230 ng/m³) and isoprene (~900 ng/m³) emissions are elevated (Fig. 13), while reduced wet deposition favors the formation of $C_{15}$ and $C_{20}$ dimers. As a result, the dimer fraction reaches 58%, with $C_{15}$ contributing 34% and $C_{20}$ contributing 24%. $O_3$ concentrations (Fig. 14) also promote multi-step autoxidation reactions. Meanwhile, the relatively low anthropogenic emissions in the Amazon result in low NO concentrations, preventing the rapid formation of $C_{10}$-ON via NO reactions with MT-HOM-$RO_2$ (Fig. 3). Instead, MT-HOM-$RO_2$ undergoes self- and cross-reactions to form dimers. Similarly, the CAF region exhibits similar characteristics, with an annual mean concentration of 0.47 μg/m³, maintaining relatively high levels throughout the year. During the peak emission period (January–May), both $C_{15}$ and $C_{20}$ levels increase simultaneously (Fig. 12). These findings highlight that low NO emissions and high biogenic emissions play an important role in the generation dimers.*

*Mid-latitude regions exhibit significant seasonal variations, with a higher proportion of $C_{10}$-ON (Fig. 12). In regions such as the US, SECN, and EUR, a clear summer peak occurs between June and August, corresponding to significant increases in monoterpene and isoprene emissions (Fig. 13). South Asia (SAS) shows a bimodal distribution, with higher emissions in March–May and October–November, respectively. During the rainy season (June–September), enhanced wet deposition leads to reduced biogenic emissions, resulting in a bimodal distribution of HOMs-SOA (Fig. 13). In the Southern Hemisphere, Australia (AUS) exhibits a seasonal peak opposite to that of the Northern Hemisphere (occurring from October to March) (Fig. 13), but the seasonal distribution still remains consistent with biogenic emissions (Fig. 13). In these regions, where anthropogenic NO emissions are high, the proportion of $C_{10}$-ON (exceeding 60%) is significantly higher than that of dimers, highlighting the dominant role of the MT-HOM-$RO_2$ + NO termination reaction under high NO conditions.*

[Figure]

**Figure 12.** *Seasonal variations of HOMs-SOA concentrations (μg/m³) in different regions. The contributions of $C_{10}$-ON, $C_{10}$-NON, $C_{15}$, and $C_{20}$ are shown in different colors, with the percentage contribution of each component indicated in the legend. Details of each geographic region can be found in Figure S9.*

[Figure]

***Figure 13.*** *Seasonal variations in monoterpene and isoprene emissions (ng/m³/s) across different regions. The average annual emissions for each species are provided in the caption. Details of each geographic region can be found in Figure S9.*

[Figure]

***Figure 14.*** *Seasonal variations of OH and O₃ concentrations (ppt and ppb) in different regions. The mean OH and O₃ values for each region are shown in the upper right corner of each panel. Details of each geographic region can be found in Figure S9.*

[Figure]

**Figure S9.** *Map showing the geographic locations of different regions analyzed in this study: Amazon Basin (AMZ), Continental U.S. (US), Southeastern China (SECN), Europe (EUR), Australia (AUS), South Asia (SAS), Central Africa (CAF).*

[Figure]

**Figure 3**. *Schematic diagram illustrating the oxidation of MT-HOM-RO$_2$ by HO$_2$, NO, or NO$_3$, leading to the formation of three types of gas-phase C$_{10}$ HOMs.*

We also add the following paragraph in Section 5 (Conclusion):

*The seasonal variation of HOMs-SOA is largely influenced by the intensity of biogenic emissions, with oxidant levels playing a secondary regulatory role. The background levels of NO, oxidant concentrations, and wet deposition conditions in different regions collectively shape the HOMs-SOA formation process. In high NO emission regions (such as the United States, Southeast Asian Continental Regions, Europe, and South Asia), C$_{10}$-ON predominates, while in low NO emission regions with high biogenic emissions (such as the Amazon rainforest and central Africa), the proportion of dimers (C$_{15}$ and C$_{20}$) is significantly increased.*

**2. Clarity of Reaction Mechanism Description**

The description of the HOMs chemical mechanism in Section 2.2.2 and Figure 1 is currently quite dense and could be made clearer with a few structural adjustments. I recommend breaking the flowchart in Figure 1 into sub-panels, each representing a distinct stage of the mechanism, and aligning the discussion in the text with these steps using corresponding subsections or paragraphs. Additionally, it would be helpful to include a summary table outlining the key reaction pathways and their roles in forming C10, C15, and C20 HOMs. Finally, I suggest adopting clearer chemical nomenclatures and ensuring that all species and acronyms used in figures and tables are clearly defined in this section.

**Response**: Figure 1 was intended only as a conceptual overview of the key reaction pathways, not as a detailed representation of the entire chemical mechanism. We now recognize that this explanation could be further clarified. In response, we have divided the flowchart into sub-panels, each representing a distinct stage of the mechanism. After each sub-panel, a table summarizing the key chemical reactions has been added. Additionally, descriptions have been provided for each set of sub-panels and tables, offering an overview of the key reaction pathways and their roles in the formation of $C_{10}$, $C_{15}$, and $C_{20}$, while also clarifying the chemical nomenclature used throughout the section. The added sections are as follows:

**2.2.2 Autoxidation**

*To account for the H-shift chemistry of MT-RO₂ leading to HOM formation (i.e., autoxidation), the first-generation monoterpene-derived RO₂ (MT-RO₂), formed via reactions of monoterpenes (MT) with OH or O₃, is classified into two categories: MT-aRO₂ and MT-bRO₂ (Fig. 1). Both categories undergo standard bimolecular reactions, but only MT-bRO₂ species proceed through autoxidation. In contrast, MT-aRO₂ species (such as APINO₂, BPINO₂, LIMONO₂, and MYRCO₂, listed in Table S12) do not participate in autoxidation.*

*Relatively high branching ratios for the formation of MT-bRO₂ are adopted, based on the values used in Table S3 of Xu et al. (2022). Specifically, the branching ratio of MT-bRO₂ is 0.75 for monoterpene + OH reactions, and 0.08 for monoterpene + O₃ reactions (Fig. 1). These values fall within the ranges reported in previous studies. Literature-based yields for MT-bRO₂ range from 0.075 to 0.83 for OH-initiated reactions (Lee et al., 2023; Piletic and Kleindienst, 2022; Pye et al., 2019; Weber et al., 2020; Xu et al., 2019) and from 0 to 0.22 for O₃-initiated reactions (Ehn et al., 2014; Jokinen et al., 2015; Roldin et al., 2019; Berndt et al., 2016; Kurtén et al., 2015; Richters et al., 2016). The reaction rate constants for OH and O₃ oxidation of monoterpenes are the same as those used in the default mechanism (Table 3), and apply equally to the formation of both MT-aRO₂ and MT-bRO₂. This approach is fully consistent with the implementation in GEOS-Chem by Xu et al. (2022), who demonstrated that such simplification can reasonably reproduce the formation of HOMs and the fate of RO₂ radicals. Furthermore, studies by Roldin et*

*al. (2019) and Weber et al. (2020) confirmed that using the same reaction rate for MT-bRO₂ and MT-aRO₂ also yields HOM concentrations that agree well with observations under forested conditions.*

*MT-bRO₂ are assumed to undergo one or multiple generations of autoxidation (Table 4). These reactions follow a temperature-dependent rate with an activation energy of 74.1 kJ/mol, consistent with previous studies (Lee et al., 2023; Möller et al., 2020; Pye et al., 2019; Roldin et al., 2019; Schervish and Donahue, 2020; Xu et al., 2019). The corresponding autoxidation rate are 0.27 s⁻¹ at 283 K, 1.30 s⁻¹ at 298 K, and 4.12 s⁻¹ at 310 K. The yield of HOMs depends on both the autoxidation rate and the fraction of MT-RO₂ that undergoes autoxidation. To reflect the uncertainty associated with these parameters, this fraction is varied in both OH- and O₃-initiated pathways as part of sensitivity experiments. A detailed discussion of these tests is provided in Section 2.3.*

[Figure]

**Figure 1.** *Schematic of monoterpene (MT) oxidation and subsequent autoxidation pathways. MT reacts with OH or O₃ to form MT-aRO₂ or MT-bRO₂, with the latter undergoing autoxidation steps to yield HOMs. Branching ratios are shown for OH and O₃ pathways.*

**Table 3.** *Initial oxidation reactions of four representative monoterpenes (APIN, BPIN, LIMON, and MYRC) with OH and O₃, leading to the formation of MT-aRO₂ (non-autoxidizable) and MT-bRO₂ (autoxidizable). Detailed descriptions of the intermediate species are provided in Table S12.*

| Index | Reactions | Reaction rate |
|:---:|:---|:---:|
| *1* | *APIN[a] + OH → 0.25\*MT-aRO₂ + 0.75\*MT-bRO₂* | *1.34e-11\*exp(410/T)* |
| *2* | *BPIN[a] + OH → 0.25\*MT-aRO₂ + 0.75\*MT-bRO₂* | *1.62e-11\*exp(460/T)* |
| *3* | *LIMON[a] + OH → 0.25\*MT-aRO₂ + 0.75\*MT-bRO₂* | *3.41e-11\*exp(470/T)* |
| *4* | *MYRC[a] + OH → 0.25\*MT-aRO₂ + 0.75\*MT-bRO₂* | *2.1e-10* |
| *5* | *APIN[a] + O₃ → 0.736\*MT-aRO₂ + 0.064\*MT-bRO₂ + 0.77\*OH + 0.066\*TERPA2O₂ + 0.22\*H₂O₂ + 0.044\*TERPA + 0.002\*TERPACID + 0.034\*TERPA2 + 0.17\*HO₂ + 0.17\*CO + 0.27\*CH₂O + 0.054\*TERPA2CO₃* | *1.34e-11\*exp(410/T)* |
| *6* | *BPIN[a] + O₃ → 0.736\*MT-aRO₂ + 0.064\*MT-bRO₂ + 0.102\*TERPK + 0.3\*OH + 0.06\*TERPA2CO₃ + 0.32\*H₂O₂ +* | *1.62e-11\* exp(460/T)* |

| | | |
|---|---|---|
| | $0.038*BIGALK + 0.19*CO_2 + 0.81*CH_2O +$ $0.11*HMHP + 0.08*HCOOH$ | |
| 7 | $LIMON^a + O_3 \rightarrow 0.736*MT\text{-}aRO_2 + 0.064*MT\text{-}bRO_2 + 0.66*OH +$ $0.132*TERPF1 + 0.33*CH_3CO_3 + 0.33*CH_2O +$ $0.066*TERPA3CO_3 + 0.33*H_2O_2 + 0.002*TERPACID$ | $3.41e\text{-}11*exp(470/T)$ |
| 8 | $MYRC^a + O_3 \rightarrow 0.736*MT\text{-}aRO_2 + 0.064*MT\text{-}bRO_2 + 0.2*TERPF2 +$ $0.63*OH + 0.63*HO_2 + 0.25*CH_3COCH_3 + 0.39*CH_2O +$ $0.18*HYAC$ | $2.1e\text{-}10$ |

$^a$ APIN, BPIN, LIMON, and MYRC represent α-pinene, β-pinene, limonene, and myrcene, respectively.

**Table 4.** *Autoxidation reactions of MT-bRO₂ leading to the formation of MT-cRO₂ and subsequently MT-HOM-RO₂.*

| Index | Reactions | Reaction rate |
|---|---|---|
| 9 | $MT\text{-}bRO_2 \rightarrow MT\text{-}cRO_2$ | $9.8e12*exp(\text{-}8836/T)$ |
| 10 | $MT\text{-}cRO_2 \rightarrow MT\text{-}HOM\text{-}RO_2$ | |

**2.2.3 Self-Reactions and Cross-Reactions**

*Due to isomers of MT-RO₂ and ISOP-RO₂, self- and cross-reactions are included (Table 5), with three branches considered for the products. First, intermediate products are produced and are lumped as C₁₀-ROH and C₁₀-CBYL. Second, RO radicals are generated, which may produce HO₂ and C₁₀-CBYL or decompose into smaller compounds. Half of the RO radicals are assumed to decompose into smaller carbonyls. Third, accretion products (C₁₅ and C₂₀) are produced. The branching ratios of the three pathways above are set as 0.29:0.67:0.04, respectively (Xu et al., 2022). However, for the self- and cross-reactions involving MT-aRO₂ (APINO₂, BPINO₂, LIMONO₂, and MYRCO₂ in Table S12) and ISOP-RO₂, a small fraction of RO radicals may undergo a unimolecular H-shift to form MT-bRO₂, with the branching ratio set to 0.05 (Xu et al., 2022). The fast reaction rate is applied here based on Table S4 in Xu et al. (2022).*

[Figure]

**Figure 2.** *Schematic illustration of self- and cross-reactions between MT-RO₂ and ISOP-RO₂ peroxy radicals.*

**Table 5.** *Summary of the self- and cross-reactions involving MT-RO$_2$ and ISOP-RO$_2$ peroxy radicals considered in this study. Detailed descriptions of the intermediate species are provided in Table S12.*

| Index | Reactions | Reaction rate |
|-------|-----------|---------------|
| 11–20 | MT-aRO$_2$ + MT-aRO$_2$ → 0.893*C$_{10}$-CBYL + 0.29*C$_{10}$-ROH + 0.603*HO$_2$ + 1.34*HYDRALD + 0.067*MT-bRO$_2$ + 0.04* C$_{20}$ | 4.0e-11 |
| 21–24 | MT-aRO$_2$ + MT-bRO$_2$ → 0.96*C10-CBYL + 0.29*C10-ROH + 0.67*HO2 + 1.34*HYDRALD + 0.04* C$_{20}$ | 4.0e-11 |
| 25–28 | MT-aRO$_2$ + MT-cRO$_2$ → 0.96*C10-CBYL + 0.29*C10-ROH + 0.67*HO2 + 1.34*HYDRALD + 0.04* C$_{20}$ | 2.6e-10 |
| 29–32 | MT-aRO$_2$ + MT-HOM-RO$_2$ → 0.96*C10-CBYL + 0.29*C10-ROH + 0.67*HO2 + 1.34*HYDRALD + 0.04* C$_{20}$ | 2.6e-10 |
| 33–56 | MT-aRO$_2$ + ISOP-RO$_2$ → 0.4465*C10-CBYL + 0.145*C10-ROH + 0.145*ROH + 0.603*HO2+1.485*HYDRALD+0.0335*MT-bRO$_2$+ 0.04* C$_{15}$ | 2.0e-10 |
| 57 | MT-bRO$_2$ + MT-bRO$_2$ → 0.96*C$_{10}$-CBYL + 0.29*C$_{10}$-ROH + 0.67*HO$_2$ + 1.34*HYDRALD + 0.04* C$_{20}$ | 4.0e-11 |
| 58 | MT-cRO$_2$ + MT-cRO$_2$ → 0.96*C$_{10}$-CBYL + 0.29*C$_{10}$-ROH + 0.67*HO$_2$ + 1.34*HYDRALD + 0.04* C$_{20}$ | 2.6e-10 |
| 59 | MT-HOM-RO$_2$ + MT-HOM-RO$_2$ → 0.96*C$_{10}$-CBYL + 0.29*C$_{10}$-ROH + 0.67*HO$_2$ + 1.34*HYDRALD + 0.04* C$_{20}$ | 2.6e-10 |
| 60 | MT-bRO$_2$ + MT-cRO$_2$ → 0.96*C$_{10}$-CBYL + 0.29*C$_{10}$-ROH + 0.67*HO$_2$ + 1.34*HYDRALD + 0.04* C$_{20}$ | 2.6e-10 |
| 61 | MT-bRO$_2$ + MT-HOM-RO$_2$ → 0.96*C$_{10}$-CBYL + 0.29*C$_{10}$-ROH + 0.67*HO$_2$ + 1.34*HYDRALD + 0.04* C$_{20}$ | 2.6e-10 |
| 62 | MT-cRO$_2$ + MT-HOM-RO$_2$ → 0.96*C$_{10}$-CBYL + 0.29*C$_{10}$-ROH + 0.67*HO$_2$ + 1.34*HYDRALD + 0.04* C$_{20}$ | 2.6e-10 |
| 63–68 | MT-bRO$_2$ + ISOP-RO$_2$ → 0.48*C$_{10}$-CBYL + 0.145*C$_{10}$-ROH + 0.145*ROH + 0.67*HO$_2$ + 1.485*HYDRALD + 0.04* C$_{15}$ | 2.0e-11 |
| 69–74 | MT-cRO$_2$ + ISOP-RO$_2$ → 0.48*C$_{10}$-CBYL + 0.145*C$_{10}$-ROH + 0.145*ROH + 0.67*HO$_2$ + 1.485*HYDRALD + 0.04* C$_{15}$ | 4.0e-11 |
| 75–80 | MT-HOM-RO$_2$ + ISOP-RO$_2$ → 0.48*C$_{10}$-CBYL + 0.145*C$_{10}$-ROH + 0.145*ROH + 0.67*HO$_2$ + 1.485*HYDRALD + 0.04* C$_{15}$ | 4.0e-11 |

**2.2.4 C₁₀ HOMs formation**

*When MT-HOM-RO₂ are oxidized by HO₂, NO, or NO₃ (Fig. 3), three types of gas-phase C₁₀ HOMs are formed: two types of C₁₀ non-nitrate HOMs (C₁₀-aNON and C₁₀-bNON) and C₁₀ nitrate HOMs (C₁₀-ON), as shown in Table 6. The rate constants used are the same as those for the MT-RO₂ + HO₂, NO, and NO₃ reactions in Xu et al. (2022).*

[Figure]

**Figure 3**. *Schematic diagram illustrating the oxidation of MT-HOM-RO₂ by HO₂, NO, or NO₃, leading to the formation of three types of gas-phase C₁₀-HOMs.*

**Table 6.** *$C_{10}$ HOMs formation. Detailed descriptions of the intermediate species are provided in Table S12.*

| Index | Reactions | Reaction rate |
|:---:|:---|:---:|
| 109 | MT-HOM-RO₂ + HO₂ → C₁₀-aNON + O₂ | 1.5e-11 |
| 110 | MT-HOM-RO₂ + NO →  0.8*NO₂ + 0.8*HO₂ + 0.4* C₁₀-bNON + 0.8*HYDRALD + 0.2* C₁₀-ON | 4.0e-12 |
| 111 | MT-HOM-RO₂ + NO₃ → HO₂ + NO₂ + 0.5* C₁₀-bNON + HYDRALD | 1.2e-12 |

**3. Justification for Excluding NO₃-Initiated HOMs**

The manuscript excludes NO3-initiated HOMs due to uncertainties. While this is understandable, recent studies suggest this pathway may be more important than previously thought. Therefore, a brief discussion of their potential importance (especially in polluted nighttime conditions) would provide a perspective for this HOM formation pathway.

**Response**: Sorry for ignoring the role of NO₃-initiated oxidation pathways in our previous discussion. We have revised the manuscript in Lines 138 to 150 (the underlined content is newly added or modified):

> *The MT-RO₂ formed by the oxidation of monoterpenes by NO₃ radicals is not considered in this study, as some studies report that the branching ratio remains highly uncertain  (Zhao et al., 2021; Nah et al., 2016; Yan et al., 2016; Roldin et al., 2019).*

Additionally, the following discussion has been added in Section 5 (Conclusion):

*This study investigates the formation of HOMs from monoterpene oxidation in a global simulation, yet significant uncertainties remain in the representation of $NO_3$-initiated pathways. Recent studies suggest that $NO_3$-initiated HOM formation may be more important than previously thought, particularly under polluted nighttime conditions. Chamber experiments on α- and β-phellandrene oxidation by $NO_3$ have shown significant SOA and HOM production, with SOA yields reaching approximately 35% and 60%, respectively, accompanied by abundant HOM monomers and dimers (Harb et al., 2024). Furthermore, field observations from the southeastern United States indicate that $NO_3$ remains the dominant oxidant of monoterpenes at night, accounting for around 60% (observed) to 80% (modeled) of total monoterpene oxidation (Desai et al., 2024). These results highlight the potential importance of $NO_3$-initiated HOM formation in contributing to organic aerosol formation under polluted nighttime conditions. However, due to structural differences in monoterpenes, such as ring strain and double-bond position, HOM yields vary widely among different species (Dam et al., 2022; Draper et al., 2024) and are highly sensitive to ambient $NO_x$ concentrations and humidity (Pasik et al., 2025; Li et al., 2022). The incomplete understanding of these mechanisms limits the accuracy of HOM predictions in models. Future research should combine field observations, laboratory constraints, and updated reaction schemes to reduce these uncertainties and improve global-scale modeling of nighttime organic aerosol formation.*

**Minor Comments**

• Lines 45–46: The introduction discusses the role of HOMs in radiative forcing, but this is not revisited in the results. Including even a qualitative discussion in the results or conclusions on how HOMs-SOA might influence regional or global climate would enhance the impact of the study.

**Response**: Thank you for your suggestion. We have added the following content to the conclusion section (Section 5) to address how HOMs-SOA might influence climate:

*On a global scale, the formation of HOMs-SOA is influenced not only by chemical reaction mechanisms but also by their potential to indirectly affect radiative forcing through changes in cloud condensation nuclei (CCN). In particular, in tropical regions such as the Amazon and central Africa, where HOMs-SOA concentrations are high, the generated CCN could significantly influence the marine low cloud areas on the western side of continents. These changes in CCN may alter cloud droplet size and cloud reflectivity, thereby impacting the regional radiative balance.*

• Lines 89–90: Please provide references for the models mentioned.
• Lines 90–95: Include details on the simulation period used in the study.

**Response**: We have added the reference in lines 89–90 and included details on the simulation period used in the study. The revised text is as follows (the underlined content is newly added or modified):

*The Community Atmosphere Model version 6 with comprehensive tropospheric and stratospheric chemistry (CAM6-Chem) from the Community Earth System Model version 2.1.0 (CESM2.1.0) is used in this study (Danabasoglu et al., 2020). The default configuration of CAM6-Chem employs the four-mode version of the Modal Aerosol Module (MAM4) (Liu et al., 2016) and applies the Volatility Basis Set (VBS) approach (Donahue et al., 2006; Hodzic et al., 2016; Jo et al., 2021; Robinson et al., 2007) to represent the formation of SOA from all volatile organic compounds (VOCs). All simulations are configured with a horizontal resolution of 0.95° in latitude and 1.25° in longitude and a vertical resolution of 32 layers up to approximately 40 km (Emmons et al., 2020). Meteorological fields, including temperature, winds, and surface fluxes, from the Modern-Era Retrospective analysis for Research and Applications (MERRA2) reanalysis data set (Gelaro et al., 2017) are used for offline nudging to minimize uncertainties in meteorology simulation (Jo et al., 2021; Tilmes et al., 2019; Liu et al., 2021). Anthropogenic and biomass burning emissions are from the standard Coupled Model Intercomparison Project 6 (CMIP6) (Eyring et al., 2016). The biogenic emissions are simulated online using the Model of Emissions of Gases and Aerosol from Nature version2.1 (MEGAN2.1) (Guenther et al., 2012).*

*The simulation period spanned from June to July 2013 and from April to June 2014, corresponding to the field campaign periods (SENEX and BAECC, see Section 2.3), and was used to evaluate model improvements (with one month for spin-up). Additionally, to assess the contribution of different chemical reaction pathways in the formation of HOMs and the spatiotemporal distribution of HOMs-SOA, we conducted one-year simulations for 2013 (with one month for spin-up) across different sensitivity experiments (see Section 2.4).*

• Lines 99–100: Specify the total amounts of monoterpene and isoprene emissions considered in the simulations.

**Response**: Thank you for your comment. The spatial distribution of monoterpene and isoprene emissions is shown in Figure S4. The temporal distribution (global and for key regions) is presented in Figure 13.

[Figure]

*Figure S4. 2013 annual averaged surface (a) MTERP emissions (unit: $\mu g/m^3/s$), (b) ISOP emissions (unit: $\mu g/m^3/s$), (c) NO concentration (unit: ppbv) in Control experiment.*

[Figure]

**Figure 13.** *Seasonal variations in monoterpene and isoprene emissions (ng/m³/s) across different regions. The average annual emissions for each species are provided in the caption. Details of each geographic region can be found in Figure S9.*

• Lines 114–115: Please rephrase this sentence, the current wording is difficult to follow.

**Response**: Thank you for your feedback. We have rephrased the sentence to improve clarity and readability. The revised sentence is as follows:

*SOA   are generated when volatile organic compounds (VOCs) undergo oxidation, leading to the formation of low-volatility gases that subsequently condense onto pre-existing aerosols.*

• Line 125: In addition to the vapor pressure equation, please include the gas-particle partitioning equation used in the model.

**Response**: We thank the reviewer for this helpful suggestion. We have added a description of the gas–particle partitioning approach based on the absorptive equilibrium theory (Pankow, 1994). In the revised manuscript, we have added the following content to the main text:

*The SOAG in different volatility bin (SOAG0~4 in Table 1) condenses on the preexisting aerosols to form SOA (soa1~5 in Table 1) based on their saturation vapor pressure calculated following Eqn. (1) (Chung and Seinfeld, 2002):*

$$P_{0,i}(T) = P_{0,i}(T_0) \cdot e^{\left[ \frac{-\Delta H_{vap}}{R} \left( \frac{1}{T} - \frac{1}{T_0} \right) \right]} \tag{1}$$

*where $P_{0,i}(T)$ is the saturation vapor pressure at temperature $T$ and $T_0 = 298$ K; $R$ is the ideal gas constant, and $\Delta H_{vap}$ is the enthalpies of vaporization which represents the energy to transform the liquid substance into gas phase (default parameterized values shown in* **Error! Reference source not found.***).*

*For SOA species $i$ and aerosol mode $m$, the equilibrium gas concentration is expressed as:*

$$g_i^* = \frac{g_{0,i}}{M_{OA}} \times A_i \tag{2}$$

*$g_{0,i}$ is the equilibrium gas mixing ratio derived from the saturation vapor pressure (Eqn. (1) and specifically shown in Eqn. (3)), $A_i$ is the particle-phase concentration of species $i$, and $M_{OA}$ is the total absorbing organic mass (including SOA and oxidized POA).*

$$g_{0,i} = \frac{P_{0,i}(T)}{P} \tag{3}$$

*where $P$ is the atmospheric pressure.*

*The dynamic exchange between gas and particle phases is described by a first-order mass transfer equation:*

$$\frac{dA_i}{dt} = (G_i - g_i^*) \times k_i \qquad (4)$$

*where $G_i$ is the gas-phase concentration and $k_i$ is the transfer coefficient. This formulation ensures mass conservation and is solved using a semi-implicit numerical scheme. Similar approaches are widely used in global climate models to represent SOA gas–particle partitioning within the volatility basis set (VBS) framework (Pankow, 1994; Donahue et al., 2006; Tilmes et al., 2015).*

• Line 189: The field campaigns referenced here have not yet been introduced. Consider moving this sentence or providing context earlier.

**Response**: we have swapped the order of Sections 2.3 (Sensitivity experiments) and 2.4 (Observations) to provide context for the field campaigns earlier in the manuscript. This adjustment ensures that the field campaigns are introduced before they are referenced in Line 189. The detailed description of the observations (Section 2.3) will be provided in the next response.

• Section 2.4: This section should be expanded. Please include more details about the field campaigns, the types of measurements conducted, the instruments used, and a brief summary of key findings relevant to the model evaluation.

**Response**: We thank the reviewer for the helpful suggestion. In response, we have expanded Section 2.4 (which is now Section 2.3) to include more detailed information about the field campaigns. The updated Section 2.3 is as follows:

*Data from two campaigns were used for comparison: the Southern Oxidant and Aerosol Study (SOAS) in the southeastern USA, and the Biogenic Aerosols – Effects on Clouds and Climate (BAECC) in Hyytiälä, Finland (Carlton et al., 2018; Martin et al., 2016; Petäjä et al., 2016) (Table 8). HOM measurements were obtained using high-resolution time-of-flight chemical ionization mass spectrometer (HRToF-CIMS) when available (Lopez-Hilfiker et al., 2014). For HOM measurements, molecular formulas of compounds containing 10 carbon atoms and at least 7 oxygen atoms were selected as HOMs. The compounds with one nitrate and without nitrate were compared to the simulated $C_{10}$-aNON, $C_{10}$-bNON, and $C_{10}$-ON, respectively. In addition to HOMs, related species such as NO, $O_3$, monoterpenes, and isoprene were also compared when the data was available (Figs. S1 and S2). The primary HOM species identified in the SENEX (Southeast Nexus) and BAECC campaigns (Tables S15 and S16).*

*Table 7.* *Field campaigns used in this paper*

| Campaigns | Dates | Locations |
|---|---|---|
| SOAS (Warneke et al., 2016) | 2013.06.01–07.15 | Centreville, Alabama, US (32.93°N, 87.13°W) |
| BAECC (Petäjä et al., 2016) | 2014.04.11–06.03 | Station for Measuring Ecosystem Atmosphere Relations (SMEAR II), Hyytiälä, Finland. (61.85°N,24.28°E) |

*Table S15.* *Molecular formulas of top 5 contributing HOM-ON and HOM-NON species (gas- and particle-phase) at Centreville, Alabama*

| HOM-ON | | HOM-NON | |
|---|---|---|---|
| Gas-phase | Particle-phase | Gas-phase | Particle-phase |
| $C_{10}H_{15}O_7N_1$ | $C_{10}H_{15}O_7N_1$ | $C_{10}H_{14}O_7$ | $C_{10}H_{14}O_7$ |
| $C_{10}H_{17}O_7N_1$ | $C_{10}H_{15}O_8N_1$ | $C_{10}H_{12}O_7$ | $C_{10}H_{12}O_7$ |
| $C_{10}H_{15}O_8N_1$ | $C_{10}H_{17}O_7N_1$ | $C_{10}H_{22}O_8$ | $C_{10}H_{16}O_7$ |
| $C_{10}H_{17}O_8N_1$ | $C_{10}H_{17}O_8N_1$ | $C_{10}H_{22}O_7$ | $C_{10}H_{22}O_8$ |
| $C_{10}H_{13}O_8N_1$ | $C_{10}H_{15}O_9N_1$ | $C_{10}H_{16}O_7$ | $C_{10}H_{22}O_7$ |

*Table S16.* *Molecular formulas of top 5 contributing HOM-ON and HOM-NON species (gas- and particle-phase) at Hyytiälä, Finland*

| HOM-ON | | HOM-NON | |
|---|---|---|---|
| Gas-phase | Particle-phase | Gas-phase | Particle-phase |
| $C_{10}H_{15}O_7N_1$ | $C_{10}H_{15}O_8N_1$ | $C_{10}H_{12}O_{11}$ | $C_{10}H_{14}O_7$ |
| $C_{10}H_{15}O_8N_1$ | $C_{10}H_{15}O_7N_1$ | $C_{10}H_{14}O_8$ | $C_{10}H_{22}O_9$ |
| $C_{10}H_{17}O_7N_1$ | $C_{10}H_{17}O_7N_1$ | $C_{10}H_{16}O_8$ | $C_{10}H_{22}O_7$ |
| $C_{10}H_{13}O_7N_1$ | $C_{10}H_{17}O_8N_1$ | $C_{10}H_{14}O_7$ | $C_{10}H_{22}O_8$ |
| $C_{10}H_{17}O_8N_1$ | $C_{10}H_{15}O_9N_1$ | $C_{10}H_{22}O_7$ | $C_{10}H_{16}O_7$ |

• Lines 225–226: This sentence appears more appropriate for the conclusions section rather than the results.

**Response**: The original sentence in lines 225–226 is as follows:

*Addressing these gaps requires coordinated laboratory measurements and targeted ambient observations to disentangle competing chemical processes.*

We have removed this sentence in the results and, recognizing the relevance of this issue for observational efforts, we have included a directional description in the final paragraph of the conclusion:

*To address persistent gaps between model predictions and observations, field campaigns targeting accretion product speciation and chamber studies that constrain MT-bRO$_2$ branching ratios are needed.*

• Section 3.3: Consider moving the global model results to a new standalone Section 4 to give them greater emphasis and allow for a more structured discussion.

**Response**: We have added an additional section (Section 4) to describe the spatiotemporal variation of HOMs-SOA. For further details, please refer to the response to the first major comment.

• Figures 7b–7c: Consider applying a minimum threshold for MTSOA (7b) and SOA (7c) before calculating the contributions of HOMs. Reporting high percentage contributions in regions with negligible SOA concentrations may be misleading and could skew global averages.
• Figures 8b, 8d, 8f, 8h: The same recommendation applies as for Figures 7b and 7c.

**Response**: Thank you for your suggestion. In response, we have revised the original Figures 7 and 8 and replaced them with the following two updated figures.

[Figure]

***Figure 10.*** *2013 annual averaged surface (a) HOMs-SOA (unit: μg/m³), (b) the contribution of HOMs-SOA to the MTSOA and (c) the contribution of HOMs-SOA to the total SOA (unit: %) in Control experiment. The global averaged value is shown in upper right corner of each figure. Proportions are only shown in regions where MTSOA or total SOA is greater than 10% of the global average.*

[Figure]

***Figure 11.*** *Vertical distribution of 2013 annual averaged (a) HOMs-SOA concentration (μg/m³) and (b) proportion of HOMs-SOA to total SOA (%) in the Control experiment. The global average value is shown in the upper right corner of each panel. Proportions are only shown in regions where total SOA is greater than 10% of the global average.*

• Terminology: Ensure consistent use of terms such as "C10-NON", "C10-ON", and "HOMs-SOA" throughout the manuscript and figures.

**Response**: We have carefully reviewed the manuscript and ensured consistent use of terms such as "$C_{10}$-NON", "$C_{10}$-ON", and "HOMs-SOA" throughout the text and figures.

• Supplement: The supplementary material is well-organized and informative. It would be helpful to reference specific tables and figures more explicitly in the main text to guide the reader.

**Response**: We have made some adjustments to the supplementary material, removing tables and figures that were not cited in the main text.

**Reference**

Berndt, T., Richters, S., Jokinen, T., Hyttinen, N., Kurten, T., Otkjaer, R. V., Kjaergaard, H. G., Stratmann, F., Herrmann, H., Sipila, M., Kulmala, M., and Ehn, M.: Hydroxyl radical-induced formation of highly oxidized organic compounds, Nat Commun, 7, 13677, 10.1038/ncomms13677, 2016.

Chung, S. H. and Seinfeld, J. H.: Global distribution and climate forcing of carbonaceous aerosols, Journal of Geophysical Research: Atmospheres, 107, AAC 14-11-AAC 14-33, https://doi.org/10.1029/2001JD001397, 2002.

Dam, M., Draper, D. C., Marsavin, A., Fry, J. L., and Smith, J. N.: Observations of gas-phase products from the nitrate-radical-initiated oxidation of four monoterpenes, Atmos. Chem. Phys., 22, 9017–9031, https://doi.org/10.5194/acp-22-9017-2022, 2022.

Desai, N. S., Moore, A. C., Mouat, A. P., Liang, Y., Xu, T., Takeuchi, M., Pye, H. O. T., Murphy, B., Bash, J., Pollack, I. B., Peischl, J., Ng, N. L., and Kaiser, J.: Impact of Heatwaves and Declining NOx on Nocturnal Monoterpene Oxidation in the Urban Southeastern United States, Journal of Geophysical Research: Atmospheres, 129, e2024JD041482, https://doi.org/10.1029/2024JD041482, 2024.

Donahue, N. M., Robinson, A. L., Stanier, C. O., and Pandis, S. N.: Coupled partitioning, dilution, and chemical aging of semivolatile organics, Environmental Science & Technology, 40, 2635-2643, 10.1021/es052297c, 2006.

Draper, D., Almeida, T. G., Iyer, S., Smith, J. N., Kurtén, T., and Myllys, N.: Unpacking the diversity of monoterpene oxidation pathways via nitrooxy–alkyl radical ring-opening reactions and nitrooxy–alkoxyl radical bond scissions, J. Aerosol Sci., 179, 106379, https://doi.org/10.1016/j.jaerosci.2024.106379, 2024. Ehn, M., Thornton, J. A., Kleist, E., Sipila, M., Junninen, H., Pullinen, I., Springer, M., Rubach, F., Tillmann, R., Lee, B., Lopez-Hilfiker, F., Andres, S., Acir, I. H., Rissanen, M., Jokinen, T., Schobesberger, S., Kangasluoma, J., Kontkanen, J., Nieminen, T., Kurten, T., Nielsen, L. B., Jorgensen, S., Kjaergaard, H. G., Canagaratna, M., Maso, M. D., Berndt, T., Petaja, T., Wahner, A., Kerminen, V. M., Kulmala, M., Worsnop, D. R., Wildt, J., and Mentel, T. F.: A large source of low-volatility secondary organic aerosol, Nature, 506, 476-479, 10.1038/nature13032, 2014.

Jokinen, T., Berndt, T., Makkonen, R., Kerminen, V. M., Junninen, H., Paasonen, P., Stratmann, F., Herrmann, H., Guenther, A. B., Worsnop, D. R., Kulmala, M., Ehn, M., and Sipila, M.: Production of extremely low volatile organic compounds from biogenic emissions: Measured yields and atmospheric implications, Proc Natl Acad Sci U S A, 112, 7123-7128, 10.1073/pnas.1423977112, 2015. Kurten, T., Rissanen, M. P., Mackeprang, K., Thornton, J. A., Hyttinen, N., Jorgensen, S., Ehn, M., and Kjaergaard, H. G.: Computational Study of Hydrogen Shifts and Ring-Opening Mechanisms in alpha-Pinene Ozonolysis Products, J Phys Chem A, 119, 11366-11375, 10.1021/acs.jpca.5b08948, 2015.

Harb, S., Cirtog, M., Alage, S., Cantrell, C., Cazaunau, M., Michoud, V., Pangui, E., Bergé, A., Giorio, C., Battaglia, F., and Picquet-Varrault, B.: HOMs and SOA formation from the oxidation of α- and β-phellandrenes by NO3 radicals, EGUsphere [preprint], https://doi.org/10.5194/egusphere-2024-3419, 2024.

Lee, B. H., Iyer, S., Kurtén, T., Varelas, J. G., Luo, J., Thomson, R. J., and Thornton, J. A.: Ring-opening yields and auto-oxidation rates of the resulting peroxy radicals from OH-oxidation of α-pinene and β-pinene, Environmental Science: Atmospheres, 3, 399-407, 10.1039/d2ea00133k, 2023.

Li, D.; Huang, W.; Wang, D.; Wang, M.; Thornton, J. A.; Caudillo, L.; Rorup, B.; Marten, R.; Scholz, W.; Finkenzeller, H.; Marie, G.; Baltensperger, U.; Bell, D. M.; Brasseur, Z.; Curtius, J.; Dada, L.; Duplissy, J.; Gong, X.; Hansel, A.; He, X. C.; Hofbauer, V.; Junninen, H.; Krechmer, J. E.; Kurten, A.; Lamkaddam, H.; Lehtipalo, K.; Lopez, B.; Ma, Y.; Mahfouz, N. G. A.; Manninen, H. E.; Mentler, B.; Perrier, S.; Petaja, T.; Pfeifer, J.; Philippov, M.; Schervish, M.; Schobesberger, S.; Shen, J.; Surdu, M.; Tomaz, S.; Volkamer, R.; Wang, X.; Weber, S. K.; Welti, A.; Worsnop, D. R.; Wu, Y.; Yan, C.; Zauner-Wieczorek, M.; Kulmala, M.; Kirkby, J.; Donahue, N. M.; George, C.; El-Haddad, I.; Bianchi, F.; Riva, M. Nitrate Radicals Suppress Biogenic New Particle Formation from Monoterpene Oxidation. Environ. Sci. Technol. 2024, 58 (3), 1601−1614.

Nah, T., Sanchez, J., Boyd, C. M., and Ng, N. L.: Photochemical Aging of α-pinene and β-pinene Secondary Organic Aerosol formed from Nitrate Radical Oxidation, Environ. Sci. Technol., 50, 222-231, 10.1021/acs.est.5b04594, 2016.

Pankow, J.: An absorption model of the gas/aerosol partitioning involved in the formation of secondary organic aerosol, Atmos. Environ., 28, 189, 1994.

Pasik, D., Golin Almeida, T., Ahongshangbam, E., Iyer, S., and Myllys, N.: Monoterpene oxidation pathways initiated by acyl peroxy radical addition, Atmos. Chem. Phys., 25, 4313–4331, https://doi.org/10.5194/acp-25-4313-2025, 2025.

Petäjä, T., O'Connor, E. J., Moisseev, D., Sinclair, V. A., Manninen, A. J., Väänänen, R., von Lerber, A., Thornton, J. A., Nicoll, K., Petersen, W., Chandrasekar, V., Smith, J. N., Winkler, P. M., Krüger, O., Hakola, H., Timonen, H., Brus, D., Laurila, T., Asmi, E., Riekkola, M.-L., Mona, L., Massoli, P., Engelmann, R., Komppula, M., Wang, J., Kuang, C., Bäck, J., Virtanen, A., Levula, J., Ritsche, M., and Hickmon, N.: BAECC: A Field Campaign to Elucidate the Impact of Biogenic Aerosols on Clouds and Climate, Bull. Amer. Meteor. Soc., 97, 1909-1928, https://doi.org/10.1175/BAMS-D-14-00199.1, 2016.

Pye, H. O. T., D'Ambro, E. L., Lee, B. H., Schobesberger, S., Takeuchi, M., Zhao, Y., Lopez-Hilfiker, F., Liu, J., Shilling, J. E., Xing, J., Mathur, R., Middlebrook, A. M., Liao, J., Welti, A., Graus, M., Warneke, C., de Gouw, J. A., Holloway, J. S., Ryerson, T. B., Pollack, I. B., and Thornton, J. A.: Anthropogenic enhancements to production of highly oxygenated molecules from autoxidation, Proc Natl Acad Sci U S A, 116, 6641-6646, 10.1073/pnas.1810774116, 2019.

Roldin, P., Ehn, M., Kurten, T., Olenius, T., Rissanen, M. P., Sarnela, N., Elm, J., Rantala, P., Hao, L., Hyttinen, N., Heikkinen, L., Worsnop, D. R., Pichelstorfer, L., Xavier, C., Clusius, P., Ostrom, E., Petaja, T., Kulmala, M., Vehkamaki, H., Virtanen, A., Riipinen, I., and Boy, M.: The role of highly oxygenated organic molecules in the Boreal aerosol-cloud-climate system, Nat Commun, 10, 4370, 10.1038/s41467-019-12338-8, 2019.

Schervish, M. and Donahue, N. M.: Peroxy radical chemistry and the volatility basis set, Atmos. Chem. Phys., 20, 1183–1199, https://doi.org/10.5194/acp-20-1183-2020, 2020.

Shao, X., Wang, M., Dong, X., Liu, Y., Shen, W., Arnold, S. R., Regayre, L. A., Andreae, M. O., Pöhlker, M. L., Jo, D. S., Yue, M., and Carslaw, K. S.: Global modeling of aerosol nucleation with a semi-explicit chemical mechanism for highly oxygenated organic molecules (HOMs), Atmos. Chem. Phys., 24, 11365–11389, https://doi.org/10.5194/acp-24-11365-2024, 2024.

Shao, X., Wang, M., Dong, X., Liu, Y., Arnold, S. R., Regayre, L. A., Jo, D. S., Shen, W., Wang, H., Yue, M., Wang, J., Zhang, W., and Carslaw, K. S.: The effect of organic nucleation on the indirect radiative forcing with a semi-explicit chemical mechanism for highly oxygenated organic molecules (HOMs), EGUsphere [preprint], https://doi.org/10.5194/egusphere-2024-4135, 2025.

Stolzenburg, D., Fischer, L., Vogel, A. L., Heinritzi, M., Schervish, M., Simon, M., Wagner, A. C., Dada, L., Ahonen, L. R., Amorim, A., Baccarini, A., Bauer, P. S., Baumgartner, B., Bergen, A., Bianchi, F., Breitenlechner, M., Brilke, S., Buenrostro Mazon, S., Chen, D., Dias, A., Draper, D. C., Duplissy, J., El Haddad, I., Finkenzeller, H., Frege, C., Fuchs, C., Garmash, O., Gordon, H., He, X., Helm, J., Hofbauer, V., Hoyle, C. R., Kim, C., Kirkby, J., Kontkanen, J., Kürten, A., Lampilahti, J., Lawler, M., Lehtipalo, K., Leiminger, M., Mai, H., Mathot, S., Mentler, B., Molteni, U., Nie, W., Nieminen, T., Nowak, J. B., Ojdanic, A., Onnela, A., Passananti, M., Petäjä, T., Quéléver, L. L. J., Rissanen, M. P., Sarnela, N., Schallhart, S., Tauber, C., Tomé, A., Wagner, R., Wang, M., Weitz, L., Wimmer, D., Xiao, M., Yan, C., Ye, P., Zha, Q., Baltensperger, U., Curtius, J., Dommen, J., Flagan, R. C., Kulmala, M., Smith, J. N., Worsnop, D. R., Hansel, A., Donahue, N. M., and Winkler, P. M.: Rapid growth of organic aerosol nanoparticles over a wide tropospheric temperature range, P. Natl. Acad. Sci. USA, 115, 9122–9127, https://doi.org/10.1073/pnas.1807604115, 2018.

Tilmes, S., Lamarque, J. F., Emmons, L. K., Kinnison, D. E., Ma, P. L., Liu, X., Ghan, S., Bardeen, C., Arnold, S., Deeter, M., Vitt, F., Ryerson, T., Elkins, J. W., Moore, F., Spackman, J. R., and Val Martin, M.: Description and evaluation of tropospheric chemistry and aerosols in the Community Earth System Model (CESM1.2), Geoscientific Model Development, 8, 1395-1426, 10.5194/gmd-8-1395-2015, 2015.

Warneke, C., Trainer, M., de Gouw, J. A., Parrish, D. D., Fahey, D. W., Ravishankara, A. R., Middlebrook, A. M., Brock, C. A., Roberts, J. M., Brown, S. S., Neuman, J. A., Lerner, B. M., Lack, D., Law, D., Hubler, G., Pollack, I., Sjostedt, S., Ryerson, T. B., Gilman, J. B., Liao, J., Holloway, J., Peischl, J., Nowak, J. B., Aikin, K., Min, K. E., Washenfelder, R. A., Graus, M. G., Richardson, M., Markovic, M. Z., Wagner, N. L., Welti, A., Veres, P. R., Edwards, P., Schwarz, J. P., Gordon, T., Dube, W. P., McKeen, S., Brioude, J., Ahmadov, R., Bougiatioti, A., Lin, J. J., Nenes, A., Wolfe, G. M., Hanisco, T. F., Lee, B. H., Lopez-Hilfiker, F. D., Thornton, J. A., Keutsch, F. N., Kaiser, J., Mao, J., and Hatch, C.: Instrumentation and Measurement Strategy for the NOAA SENEX Aircraft Campaign as Part of the Southeast Atmosphere Study 2013, Atmos Meas Tech, 9, 3063-3093, 10.5194/amt-9-3063-2016, 2016.

Weber, J., Archer-Nicholls, S., Griffiths, P., Berndt, T., Jenkin, M., Gordon, H., Knote, C., and Archibald, A. T.: CRI-HOM: A novel chemical mechanism for simulating highly oxygenated organic molecules

(HOMs) in global chemistry–aerosol–climate models, Atmos. Chem. Phys., 20, 10889-10910, 10.5194/acp-20-10889-2020, 2020.

Xu, L., Moller, K. H., Crounse, J. D., Otkjaer, R. V., Kjaergaard, H. G., and Wennberg, P. O.: Unimolecular Reactions of Peroxy Radicals Formed in the Oxidation of alpha-Pinene and beta-Pinene by Hydroxyl Radicals, J Phys Chem A, 123, 1661-1674, 10.1021/acs.jpca.8b11726, 2019.

Yan, C., Nie, W., Äijälä, M., Rissanen, M. P., Canagaratna, M. R., Massoli, P., Junninen, H., Jokinen, T., Sarnela, N., Häme, S. A. K., Schobesberger, S., Canonaco, F., Yao, L., Prévôt, A. S. H., Petäjä, T., Kulmala, M., Sipilä, M., Worsnop, D. R., and Ehn, M.: Source characterization of highly oxidized multifunctional compounds in a boreal forest environment using positive matrix factorization, Atmos. Chem. Phys., 16, 12715-12731, 10.5194/acp-16-12715-2016, 2016.

Xu, R. C., Thornton, J. A., Lee, B., Zhang, Y. X., Jaegle, L., Lopez-Hilfiker, F. D., Rantala, P., and Petaja, T.: Global simulations of monoterpene-derived peroxy radical fates and the distributions of highly oxygenated organic molecules (HOMs) and accretion products, Atmos. Chem. Phys., 22, 5477-5494, 10.5194/acp-22-5477-2022, 2022.

Zhao, B., Fast, J. D., Donahue, N. M., Shrivastava, M., Schervish, M., Shilling, J. E., Gordon, H., Wang, J., Gao, Y., Zaveri, R. A., Liu, Y., and Gaudet, B.: Impact of Urban Pollution on Organic-Mediated New-Particle Formation and Particle Number Concentration in the Amazon Rainforest, Environ Sci Technol, 55, 4357-4367, 10.1021/acs.est.0c07465, 2021.

Zhao, B., Donahue, N. M., Zhang, K., Mao, L., Shrivastava, M., Ma, P.-L., Shen, J., Wang, S., Sun, J., Gordon, H., Tang, S., Fast, J., Wang, M., Gao, Y., Yan, C., Singh, B., Li, Z., Huang, L., Lou, S., Lin, G., Wang, H., Jiang, J., Ding, A., Nie, W., Qi, X., Chi, X., and Wang, L.: Global variability in atmospheric new particle formation mechanisms, Nature, 631, 98-105, 10.1038/s41586-024-07547-1, 2024.